# Causal effect of video gaming on mental well-being in Japan 2020–2022

Hiroyuki Egami [1,2] ✉, Md. Shafiur Rahman [3,4], Tsuyoshi Yamamoto [5], Chihiro Egami [6] & Takahisa Wakabayashi [7]

The widespread use of video games has raised concerns about their potential negative impact on mental well-being. Nevertheless, the empirical evidence supporting this notion is largely based on correlational studies, warranting further investigation into the causal relationship. Here we identify the causal effect of video gaming on mental well-being in Japan (2020–2022) using game console lotteries as a natural experiment. Employing approaches designed for causal inference on survey data ($n$ = 97,602), we found that game console ownership, along with increased game play, improved mental well-being. The console ownership reduced psychological distress and improved life satisfaction by 0.1–0.6 standard deviations. Furthermore, a causal forest machine learning algorithm revealed divergent impacts between different types of console, with one showing smaller benefits for adolescents and females while the other showed larger benefits for adolescents. These findings highlight the complex impact of digital media on mental well-being and the importance of considering differential screen time effects.

As digital devices and the Internet become integral parts of daily life, concerns about their potential negative impact on human well-being, especially that of prolonged screen time, have become more pronounced. Video games, at the forefront of this debate, increasingly encounter public scepticism[1]. Controversial health policy decisions, such as the latest discussions by the World Health Organization regarding gaming disorder, have exacerbated negative perceptions of video gaming[2]. The addition of gaming disorder to the International Classification of Diseases (ICD-11) has led to stigmatization among many young people and their carers who consider gaming as a normal part of life[3].

Amid escalating concerns about the negative effects of gaming, the coronavirus disease 2019 (COVID-19) pandemic that emerged in 2020 temporarily spotlighted video games as a preferred form of leisure that fit social distancing guidelines. The global number of individuals playing video games has reached nearly three billion[4], accompanied by an increase in gaming time[5]. Yet, this surge in video game engagement has renewed concerns about potential negative health impacts[6]. Policymakers, researchers and public stakeholders are particularly concerned about game addiction and its possible adverse effects on psychological well-being. However, the current evidence on the effects of video game play is insufficient, not necessarily due to a lack of research but rather the focus and approach of existing studies.

There is extensive research on the effects of video games on users, including their impact on addiction, well-being, cognitive function and aggression. Over the past decades, aggression has received considerable academic attention, but no conclusive evidence on the relationship between gaming and aggression exists. Early studies, predominantly from the 2000s, suggested a connection between digital game violence and heightened aggression[7,8]. However, many subsequent studies, including pre-registered ones, have disputed this linkage[9–11]. While the evidence about aggression remains inconclusive, it is noteworthy that the interest among scholars and policymakers has progressively

[1]Research Institute of Economic Science, Nihon University, Tokyo, Japan. [2]Ritsumeikan Center for Game Studies, Ritsumeikan University, Kyoto, Japan. [3]Research Center for Child Mental Development, Hamamatsu University School of Medicine, Hamamatsu, Japan. [4]United Graduate School of Child Development, Osaka University, Kanazawa University, Hamamatsu University School of Medicine, Chiba University, and University of Fukui, Suita, Japan. [5]Department of Policy Studies, National Graduate Institute for Policy Studies, Tokyo, Japan. [6]Office of Audit Support and Innovations, Board of Audit of Japan, Tokyo, Japan. [7]Faculty of Regional Policy, Takasaki City University of Economics, Takasaki, Japan. ✉e-mail: egami.hiroyuki@nihon-u.ac.jp

shifted to the connection between video gaming and mental well-being over the past decade[12].

The examination of the association between gaming and psychological well-being in existing literature has yielded mixed and inconsistent findings[13,14]. First, negative associations are mainly documented in observational studies[6,15–20], while some experimental studies on violent games also indicate adverse relationships[21]. A substantial part of such observational studies consists of the examination of the effect of screen time, including time spent on video games[6,17,19]. Second, positive associations are documented in both observational studies and experimental studies[22–27]. Some experimental studies have specifically employed video games of genres like casual games and exergames as therapeutic tools, indicating an underlying assumption of their beneficial impact on mental well-being before conducting these studies[24,28]. In contrast, some recent observational studies have found neither positive nor negative associations[29,30].

Two primary methodological challenges might underlie the conflicting findings in the literature. First, the scarcity of evidence regarding the causal relationship between playing video games and well-being has been a problem; most observational studies depend on association analysis with either cross-sectional or longitudinal data[31,32]. Second, many experimental studies lack tests for external validity and have faced criticisms[26,30,33]. Typical experimental studies that invite participants to play video games for a limited time fail to replicate the natural gaming environment. For example, examining the effect of 'heavy gaming' (or dysregulated gaming)—a notion lacking consensus[34]—or even 'moderate gaming' on mental well-being in laboratory settings is challenging. Identifying the causal relationship of habitual behaviours like video gaming, much like other lifestyle habits, such as the health effects of moderate alcohol consumption[35,36], presents inherent challenges. The difficulties in conducting randomized controlled trials with habitual behaviours, the multifaceted nature of these behaviours coupled with numerous confounders, and the bidirectional relationship between behaviours and health outcomes collectively complicate the identification of causal relationships.

Given the substantial disagreement within the literature and the methodological challenges, how video gaming may actually impact psychological well-being remains elusive. A promising approach is to apply causal inference to observational data[37] by exploiting a natural experimental design. However, finding a suitable natural experimental situation has been challenging. This study addresses the gap by leveraging a gaming console lottery, which is as close to an ideal natural experiment as possible. By identifying a unique and fitting situation and collecting relevant data in a timely manner, the study moves beyond the constraints of correlation analysis and controlled laboratory settings, offering a more authentic examination of the effect of playing video games on mental well-being in daily life.

To this end, we applied a natural experimental study design to the original survey data containing information on video gaming activity and mental well-being indicators for individuals aged 10–69 in Japan, collected at various points between 2020 and 2022 during the COVID-19 pandemic. Supply chain disruptions and surged demands during this time limited the availability of two major gaming consoles: Nintendo Switch (Switch) and PlayStation 5 (PS5). To address these shortages, Japanese retailers used lotteries to assign these gaming consoles to consumers, inadvertently creating a plausibly random distribution of opportunities to play video games. Winning a lottery became the primary determinant of whether one could purchase these consoles; details of the lotteries and the two game consoles are provided in Supplementary Method 1. Leveraging this unique circumstance and using original survey data, we drew causal inferences grounded in this pandemic context. Additionally, by applying a causal forest machine learning algorithm—an algorithm for estimation of heterogeneous treatment effects—to our diverse sample, we investigated the moderating role of sociodemographic factors in the causal link between

video gaming and well-being. While this specific intersection remains relatively understudied, there is a growing consensus among digital media scholars about the value of a person-specific approach, viewing it as a pathway towards tailored mental health interventions[38].

## Results

### Participant characteristics
Table 1 presents the study participants' background characteristics. Out of the 97,602 included survey respondents, 8,192 took part in the lottery. Approximately one-fourth of the 97,602 respondents were between 10 and 25 years old, while 39% were between 45 and 69. Around 21% were students, 10.7% were unemployed and 39% were full-time employees. Over one-third of the 8,192 lottery participants (35%) were hardcore gamers, and around 20% were core gamers. Time spent playing games was closely related to video gaming preference; hardcore gamers spent more than 1 h 30 min per day (Supplementary Table 1).

### Preliminary association analysis
Before our main analysis, we conducted multivariate regression without leveraging our natural experiment, expecting confounded results. This traditional approach shows associations rather than establishing causal relationships. A statistically significant positive correlation between video gaming and psychological distress (PD) (Kessler Psychological Distress Scale, K6) was found for two out of five estimates by a regression model controlling for a comprehensive set of covariates (Supplementary Table 2, model 3). Conversely, a significant positive association between gaming and life satisfaction (Satisfaction With Life Scale, SWLS) was found. Further details are presented in Supplementary Result 1.

### Multivariate regression and propensity score matching
Moving to our core analysis (Table 2), which aims to establish causal relationships, we present the results derived from our natural experimental framework. The intention-to-treat (ITT) effects of winning game console lotteries, estimated by multivariate regression and propensity score matching approach (PSM), are statistically significant and comparable (Fig. 1). Winning a Switch lottery reduced PD by approximately 0.2 standard deviations (s.d.) (0.18 s.d. with 95% confidence interval (CI) 0.13–0.24, $P < 0.001$ by regression; 0.16 s.d. with 95% CI 0.06–0.25, $P = 0.001$ by PSM). Similarly, winning a PS5 lottery led to a distress reduction of around 0.1 s.d. (0.08 s.d. with 95% CI 0.02–0.13, $P = 0.014$ by regression; 0.07 s.d. with 95% CI 0.01–0.14, $P = 0.032$ by PSM). Furthermore, winning a PS5 lottery enhanced life satisfaction by around 0.2 s.d. (0.15 s.d. with 95% CI 0.10–0.20, $P < 0.001$ by regression; 0.18 s.d. with 95% CI 0.10–0.25, $P < 0.001$ by PSM). Additionally, lottery winners increased their daily video game play time by around 0.5 h (0.53 h with 95% CI 0.43–0.63, $P < 0.001$ by regression; 0.57 h with 95% CI 0.35–0.79, $P < 0.001$ by PSM) but did not increase smartphone game play time (Supplementary Fig. 1).

The lottery winners and non-winners exhibited minor differences in the background characteristics: only one—number of times that the respondents joined the lotteries (as expected, see 'Statistical analysis' section in Methods)—out of 30 variables had standardized differences exceeding 0.10 in absolute value for PS5 lottery participants (Supplementary Table 3) and three for Switch lottery participants (Supplementary Table 4 with more details in Supplementary Result 3). Additionally, the 'ITT effect' on pseudo-outcomes was small and not statistically significant, confirming unconfoundedness (Supplementary Table 5). Moreover, the covariates' balance after matching and common support indicated that our PSM analysis was successful in achieving a balance between the treatment and control groups (Supplementary Figs. 2–5 with further explanations in Supplementary Result 4).

Additional analyses mitigate concerns over an unadjusted potential confounder: non-winning lottery participations (elaborated in 'Assessing natural experiment validity' subsection in Methods), thereby

**Table 1 | Background characteristics of study participants**

| | Lottery participant status | | | Total |
|---|---|---|---|---|
| | **Switch lottery participant** | **PS5 lottery participant** | **Non-participant** | |
| **Age** | | | | |
| 10–25 years (n, %) | 498 (28.1) | 1,565 (24.4) | 22,225 (24.9) | 24,288 (24.9) |
| 25–44 years | 689 (38.9) | 2,639 (41.1) | 31,540 (35.3) | 34,868 (35.7) |
| 45–69 years | 586 (33.1) | 2,215 (34.5) | 35,645 (39.9) | 38,446 (39.4) |
| Male | 945 (53.3) | 3,943 (61.4) | 45,215 (50.6) | 50,103 (51.3) |
| Have children (yes) | 1,055 (59.6) | 3,289 (51.3) | 44,614 (49.9) | 48,958 (50.2) |
| **Marital status** | | | | |
| Married | 1,186 (67.0) | 3,807 (59.4) | 50,872 (56.9) | 55,865 (57.3) |
| Divorced/separated | 80 (4.5) | 330 (5.1) | 5,807 (6.5) | 6,217 (6.4) |
| Not married | 505 (28.5) | 2,275 (35.5) | 32,658 (36.6) | 35,438 (36.3) |
| **Occupation** | | | | |
| Student | 420 (23.7) | 1,342 (20.9) | 19,057 (21.3) | 20,819 (21.3) |
| Stay-at-home wife/husband | 165 (9.3) | 458 (7.1) | 9,409 (10.5) | 10,032 (10.3) |
| Full-time employee | 795 (44.8) | 3,146 (49.0) | 34,186 (38.2) | 38,127 (39.1) |
| Part-time employee | 172 (9.7) | 596 (9.3) | 10,778 (12.1) | 11,546 (11.8) |
| Self-employed/others | 117 (6.6) | 413 (6.4) | 6,143 (6.9) | 6,673 (6.8) |
| Unemployed/not a student | 104 (5.9) | 464 (7.2) | 9,837 (11.0) | 10,405 (10.7) |
| **Gaming preference** | | | | |
| Hardcore gamer | 623 (35.1) | 2,306 (35.9) | 12,709 (14.2) | 15,638 (16.0) |
| Core gamer | 344 (19.4) | 1,625 (25.3) | 17,806 (19.9) | 19,775 (20.3) |
| Middle-core gamer | 346 (19.5) | 1,103 (17.2) | 21,261 (23.8) | 22,710 (23.3) |
| Casual gamer | 185 (10.4) | 596 (9.3) | 16,354 (18.3) | 17,135 (17.6) |
| Non-gamer | 275 (15.5) | 789 (12.3) | 21,280 (23.8) | 22,344 (22.9) |
| **Job: industries** | | | | |
| Engineering and construction[a] | 125 (7.1) | 434 (6.8) | 4,743 (5.3) | 5,302 (5.4) |
| Textile and cosmetics[b] | 104 (5.9) | 435 (6.8) | 4,844 (5.4) | 5,383 (5.5) |
| Manufacturing | 204 (11.5) | 889 (13.8) | 9,009 (10.1) | 10,102 (10.4) |
| Trading and mass media[c] | 65 (3.7) | 227 (3.5) | 2,727 (3.0) | 3,019 (3.1) |
| Distributors, retailers | 79 (4.5) | 285 (4.4) | 4,236 (4.7) | 4,600 (4.7) |
| Carriers[d] | 74 (4.2) | 281 (4.4) | 3,625 (4.1) | 3,980 (4.1) |
| Public works | 97 (5.5) | 356 (5.5) | 4,503 (5.0) | 4,956 (5.1) |
| IT industries[e] | 112 (6.3) | 419 (6.5) | 3,878 (4.3) | 4,409 (4.5) |
| Financial services | 62 (3.5) | 220 (3.4) | 2,485 (2.8) | 2,767 (2.8) |
| Food services and other services[f] | 184 (10.4) | 657 (10.2) | 9,794 (11.0) | 10,635 (10.9) |
| Medical care, welfare | 125 (7.1) | 399 (6.2) | 6,170 (6.9) | 6,694 (6.9) |
| Education | 50 (2.8) | 220 (3.4) | 3,312 (3.7) | 3,582 (3.7) |
| Others[g] | 130 (7.3) | 426 (6.6) | 6,230 (7.0) | 6,786 (7.0) |
| Not applicable[h] | 362 (20.4) | 1,171 (18.2) | 23,854 (26.7) | 25,387 (26.0) |
| **Exposures** | | | | |
| Have a Switch (yes) | 1,098 (61.9) | 3,452 (53.8) | 24,070 (26.9) | 28,620 (29.3) |
| Played Switch this month (yes) | 828 (46.7) | 2,513 (39.1) | 14,549 (16.3) | 17,890 (18.3) |
| Have a PS5 (yes) | 117 (6.6) | 1,341 (20.9) | 771 (0.9) | 2,229 (2.3) |
| Played PS5 this month (yes) | 72 (4.1) | 786 (12.2) | 344 (0.4) | 1,202 (1.2) |
| **Video game play time** | | | | |
| <1 h per day | | 1,500 (43.0) | 27,342 (71.0) | 28,842 (68.6) |
| 1 to 3 h per day | | 1,581 (45.3) | 9,290 (24.1) | 10,871 (25.9) |
| Over 3 h per day | | 410 (11.7) | 1,899 (4.9) | 2,309 (5.5) |

**Table 1 (continued)| Background characteristics of study participants**

| | Lottery participant status | | | Total |
| --- | --- | --- | --- | --- |
| | **Switch lottery participant** | **PS5 lottery participant** | **Non-participant** | |
| Smartphone game play time | | | | |
| <1h per day | | 1,446 (41.4) | 21,918 (56.9) | 23,364 (55.6) |
| 1 to 3h per day | | 1,692 (48.5) | 14,335 (37.2) | 16,027 (38.1) |
| Over 3h per day | | 353 (10.1) | 2,278 (5.9) | 2,631 (6.3) |
| Win a Switch lottery (yes) | 926 (52.2) | | | 926 (52.2) |
| Win a PS5 lottery (yes) | | 1,397 (21.8) | | 1,397 (21.8) |
| Total | 1,773 | 6,419 | 89,410 | 97,602 |

Respondents' characteristics are displayed. Caregivers' characteristics are used where appropriate. [a]Civil engineering, construction, property development and housing services; [b]Daily essentials, textile and apparel, beauty products, and food and drinks; [c]Trading firms, publishing, printing and mass communication; [d]Transportation, storage and logistics; [e]Software and IT services; [f]Food services, hair styling, beauty services and miscellaneous services; [g]Other sectors and business types; [h]Not applicable (including missing responses).

**Table 2 | Overview of statistical analysis**

| Methods | Estimand | Exposure/treatment (assignment) variables | Excluded instrument | Advantages/purposes | Details/equations |
| --- | --- | --- | --- | --- | --- |
| Multivariate regression | | | | Simple and easy to understand, thus suitable to be a baseline specification. This method can be extended to the IV approach. | Following Imbens et al.[57], we used equation (1) in Supplementary Methods. |
| PSM method | ITT effect | Winning game console lottery. | | While linear regression assumes a linear relationship between the independent and dependent variables and extrapolates beyond the range of observed data, PSM does not depend on such parametric assumptions. Instead, PSM creates a balanced comparison by matching units based on propensity scores. | Following Imbens (2015)[43], we utilized the algorithm to choose matching variables and created a subsample from the original dataset. We conducted covariate balance checks and examined the common support of the propensity score distribution. Further details and the PSM design choice are explained in Supplementary Methods. |
| IV method | LATE | Owning Switch/PS5. Playing Switch/PS5 last month. Video game play time. | Winning game console lottery. | This method can estimate the causal effect of standard exposure variables (for example, owning a Switch/PS5) among compliers. Additionally, this method is utilized for a subgroup analysis investigating whether the impact of video gaming varies on the basis of the time spent. | The 2SLS estimation technique is employed. The first-stage regression results are found in Supplementary Table 10. Weak instrument tests were conducted to assess the relevance. Further methodological details are available in Supplementary Methods. |
| Machine learning (IV causal forest) | CLATE | Owning Switch/PS5. | | This method predicts the treatment effects for each individual, considering their specific characteristics. The primary aim was to investigate effect modification (or moderation). Additionally, the correlation between CLATEs on well-being and video game duration was analysed to better understand the underlying mechanism of the impact of video gaming. | The method aligns with key principles of IV regressions. The credibility of the estimation is assessed by comparing estimates with those derived from traditional IV regression analyses (Fig. 3 and Supplementary Fig. 19). Further methodological details are available in Supplementary Methods. |

Methodological particulars of the statistical analyses are displayed. The causal inference and machine learning analysis targeted only those who participated in the console lottery (N=1,773 for Switch, 6,419 for PS5).

reinforcing our primary findings. The imputation method and the short-period subsample analysis yielded results aligning with our primary ITT analysis (Supplementary Figs. 6 and 7). Moreover, the examination of causal diagrams supported our assumption of conditional unconfoundedness while highlighting data limitations (Supplementary Figs 8 and 9), discussed further in the subsequent section. Comprehensive details of these additional analyses are available in Supplementary Method 2.8 and Supplementary Result 5.

Our findings were further supported through sensitivity checks with alternative model selections; these included variations in the selection of regression covariates and different specifications for PSM models (discussed in Supplementary Result 6 and Supplementary Fig. 10). As further robustness checks, regression analysis taking mild-to-serious psychological distress (MSPD)/serious psychological distress (SPD) dummy variables (K6 ≥ 5 and K6 ≥ 13, respectively) as the outcome was conducted and reinforced our findings (details in Supplementary Result 2 and Supplementary Table 6). Additionally, to address potential concerns regarding survey non-responses, we assessed systematic differences in background characteristics between respondents and non-respondents across the entire sample (Supplementary Table 7), though this comparison showed modest differences and is not directly related to our analysis sample.

### Instrumental variable approach

The impacts (local average treatment effect, LATE) of the three types of exposure—owning a Switch/PS5, playing Switch/PS5 within the survey

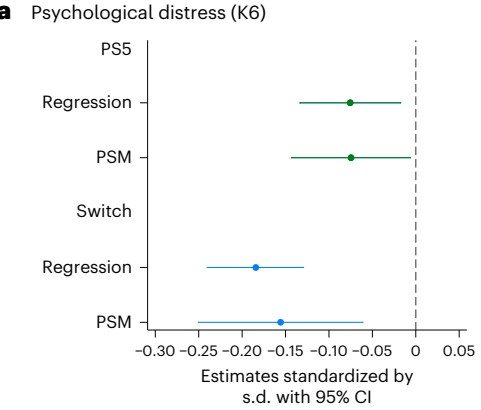

**a** Psychological distress (K6)

**b** Life satisfaction (SWLS)

**Fig. 1 | Causal effects on mental well-being from winning Switch and PS5 lotteries in Japan (_N_ = 8,192).** The causal effect of winning console lotteries on well-being is estimated by multivariate regression and PSM methods. **a,b**, The effects on PD (**a**) and on life satisfaction (**b**). The analysis sample is limited to those who joined game console lotteries. The point estimates (mean values) and the 95% CIs are shown. Regression standard errors are clustered by prefectures. Abadie–Imbens robust standard errors are used for PSM. The regression estimates are derived on the basis of equation (1) (Supplementary Methods). The estimates are standardized by the s.d. A lower K6 score indicates less PD, whereas a higher SWLS score indicates greater life satisfaction.

month, and video gaming time—estimated using the instrumental variable (IV) method, are shown in Fig. 2. Possessing a Switch improved mental health by 0.60 s.d. (95% CI 0.38–0.82, $P < 0.001$), whereas possessing a PS5 improved it by 0.12 s.d. (95% CI 0.03–0.21, $P = 0.013$). Playing Switch within the survey month improved mental health by 0.81 s.d. (95% CI 0.53–1.10, $P < 0.001$), while playing PS5 resulted in a 0.20 s.d. (95% CI 0.05–0.36, $P = 0.012$) improvement. Additionally, possession of a PS5 enhanced life satisfaction by 0.23 s.d. (95% CI 0.16–0.31, $P < 0.001$), and playing PS5 improved life satisfaction by 0.41 s.d. (95% CI 0.27–0.55, $P < 0.001$). Furthermore, an extra hour of daily video game play led to a 0.20 s.d. (95% CI 0.01–0.40, $P = 0.043$) improvement in mental health and a 0.27 s.d. (95% CI 0.06–0.47, $P = 0.014$) increase in life satisfaction. Moreover, the results of IV regression analysis taking MSPD/SPD dummy variables as outcomes supported our findings (Supplementary Table 8). Weak instrument tests detected no issues (Supplementary Table 9; first-stage regression results are found in Supplementary Table 10). In addition, possession of a PS5 increased video gaming time by 0.82 h (95% CI 0.51–1.14, $P < 0.001$) (Supplementary Fig. 11).

Subsequently, subgroup analysis in Supplementary Fig. 12 indicated that the benefits of video gaming diminished as the duration of gaming increased; extending video gaming time beyond 3 h per day was less beneficial than playing video games for a more limited period. Further explanation is found in Supplementary Result 7. Moreover, our quantile regressions revealed a larger effect of gaming among individuals with high distress levels, as shown in Supplementary Fig. 13.

**Machine learning**

Figure 3 demonstrates conditional local average treatment effect (CLATE)[39] estimates, predicted by the IV causal forest algorithm (also called IV forest or instrumental forest), for ownership of a Switch or a PS5, respectively. First, Fig. 3a,b depicts histograms of estimated CLATEs (outcome: K6), indicating that video gaming positively impacts the mental well-being of most individuals. The illustrated histograms of CLATEs align with the LATE derived through the IV method, thereby supporting the credibility of the estimated CLATEs.

Second, Fig. 3c,d portray the magnitude of psychological benefits from video gaming on the vertical axis and that of increased video game play time on the horizontal axis. As the magnitude of estimated CLATEs on game play time increased, the CLATEs on K6 and SWLS also became more pronounced, indicating a more pronounced improvement in well-being.

Third, we assessed the effect of owning a Switch/PS5 on psychological welfare, gauged by K6 (relevant assessment for SWLS available in Supplementary Result 8), by examining effect modification (or moderating effects). Estimated CLATEs were more substantial in absolute value for Switch in younger age groups (Fig. 3e)—remember that a smaller K6 indicates reduced PD. Conversely, the CLATEs were less pronounced for younger individuals with a PS5 (Fig. 3f).

Further, the impact of owning a PS5 was more prominent among males, while the effect of Switch ownership was similar for both genders, possibly slightly favouring females (Fig. 4a,b). Moreover, the effect of PS5 was more pronounced among households without children (or full-time employees), which was not observed in the effect of Switch (Fig. 4c–f). Lastly, the effect of PS5 was more prominent among hardcore gamers, whereas the effect of Switch was stronger for non-gamers (Supplementary Fig. 14).

## Discussion

This study provides an estimation of the causal effects of engagement with video games on mental well-being within a real-world context, grounded in plausible causal assumptions. Through a natural experimental approach, we demonstrated that winning a lottery for a Switch or PS5 positively impacted mental well-being for individuals aged 10–69 in Japan (2020–2022), using multivariate regression and PSM methods. Subsequently, the IV method revealed that engagement with video games—ownership/playing of a Switch or PS5 and time of video gaming—positively impacted mental well-being. Further, the machine learning analysis indicated that socioeconomic factors influenced the magnitude of the gaming effect. Each method offered unique insights into the causal effects of video gaming on mental well-being (an overview of the statistical analysis is found in Table 2, with detailed descriptions of the methodologies provided in the Methods section).

Consistent with previous research, our preliminary association analysis showed positive correlations between video gaming and PD, implying negative associations between gaming and mental health (a higher K6 score represents poorer mental health; Supplementary Table 2). Nevertheless, the modest effect sizes observed in our analysis question their practical importance, particularly when we consider a threshold of 0.2 s.d. units as the smallest effect size of interest[40]. Notably, three out of the five estimates from model 3 were not statistically significant. Conversely, all our estimates indicated positive correlations between video gaming and life satisfaction—a finding that, while counterintuitive given certain public perceptions around gaming, is consistent with recent studies[26,41]. However, it is crucial to note that

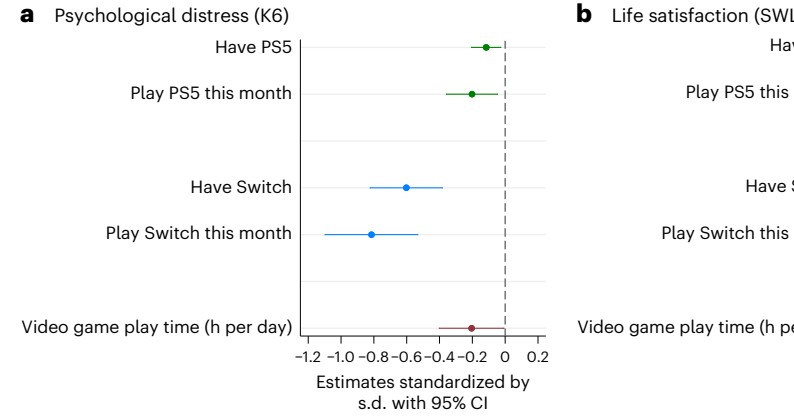

**Fig. 2 | Causal effects of video game engagement on mental well-being in Japan (N = 8,192).** The causal effect of video game engagement on mental well-being is estimated by the IV method. **a,b**, The effects on PD (**a**) and on life satisfaction (**b**). The analysis sample is limited to those who joined game console lotteries. The point estimates (mean values) and the 95% CIs are shown. Standard errors are clustered by prefectures. A lower K6 score indicates less PD,

whereas a higher SWLS score indicates greater life satisfaction. The estimates are standardized by the s.d. The estimates are also presented in Supplementary Table 8. The estimates for possession of video game consoles are preferred as they are more likely to adhere to the exclusion restriction requirement, a nontestable assumption of the IV method. The estimates based on console usage and play duration are more prone to violating this assumption.

these results only reflect associations and do not necessarily elucidate causality. Readers can find further insights on the correlation analysis in Supplementary Method 2.3 and Supplementary Result 1.

Our multivariate regression and PSM results reveal that winning a game device lottery positively impacted psychological health and overall life satisfaction. The beneficial effects were observed for both Switch and PS5, with Switch showing a higher level of psychological improvement. From multiple facets, we assessed whether the results can be considered as causal evidence. Our assessment of the baseline characteristics comparison tables and pseudo-outcome tests supported the unconfoundedness of the lottery results, and the validity of the PSM estimates was confirmed by common support and balance checks. Additional analyses—imputation approach, short-period subsample analysis and consideration of causal diagrams—reinforced the assumption of conditional unconfoundedness. We also found that winning a PS5 lottery increased video game play time but not smartphone game time, suggesting that the rise in engagement with video games was key to enhancing well-being (Supplementary Fig. 1).

The causal inference literature has increasingly acknowledged the importance of not solely relying on ordinary least squares (OLS) regression and comparing OLS estimates with those derived from other methods[42–44]. Following relevant recommendations, we compared regression and PSM estimates across various models (Fig. 1 and Supplementary Fig. 10). The consistency of our results lends credibility to our findings. Notably, we observed that the regression estimates without adjusting for covariates closely resemble estimates from other models, including PSM (Supplementary Fig. 10). This supports the plausibility of assuming the unconfoundedness of the lottery-win variable within our dataset.

The IV method provided causal evidence that increased video game engagement has a positive psychological effect. Specifically, we found that possession of a Switch/PS5 and spending more hours gaming improved well-being. Furthermore, our results were supported by machine learning analysis on the association between CLATEs on gaming time and CLATEs on well-being (Fig. 3). This analysis showed that individuals whose gaming time increased more substantially (due to the lottery wins) experienced, in parallel, greater improvements in their well-being. However, the psychological benefits of video gaming diminished when the gaming duration exceeded 3 h, as demonstrated in the subgroup analysis of the IV method (Supplementary Fig. 12).

Acknowledging effect size is increasingly crucial as large datasets can unveil statistically significant yet minor effects, leading to potential overinterpretation of 'crud' effects—trivial or spurious associations[45].

In our analysis, the IV method identified effect sizes for video game ownership as follows: 0.60 s.d. for Switch on mental health, 0.12 s.d. for PS5 on mental health and 0.23 s.d. for PS5 on life satisfaction. Except for the PS5's impact on mental health, these effect sizes exceed 0.2 s.d.: the smallest effect size of interest for media effects research proposed by Ferguson (2009)[40]. The effect size for Switch ownership is particularly notable, exceeding 0.5 s.d.—a threshold suggested by Norman et al. (2003) as a perceptible improvement to participants in their medical study[46]. Additionally, the effect sizes for game play on Switch and PS5, also estimated by the IV method, range from 0.2 to 0.8 s.d. Therefore, we conclude that the estimated positive effects of video game engagement on mental well-being are not only non-negligible but also probably perceptible to participants.

Previous investigations assessing the relationship between engagement in video gaming and mental well-being have produced inconsistent results[6,15–18,21–27,29,47]. Our findings align with experimental studies and demonstrate causal evidence of the positive gaming effect outside the laboratory environment. The findings contradict previous observational studies; this inconsistency could be due to the lack of causal inference in those studies[6,15–18]. Recall that our preliminary analysis showed a small but negative correlation between gaming and mental health, even after adjusting for covariates. Various confounding factors, such as real-life frustration[48], social connections and lifestyles including long Internet time, skipping meals or late bedtime[49], were likely to remain unadjusted.

The age-dependent influence of digital media has garnered considerable attention from policymakers and scholars[50]. Our machine learning approach unveiled that the pattern of gaming's differential effect depending on age was markedly divergent between Switch and PS5. The psychological benefits were less pronounced among young PS5 users; in contrast, the benefits were less pronounced among adult Switch users. This disparity could be attributed to the differences between the nature of the two devices (elaborated in Supplementary Method 1.2). For instance, Switch is frequently played in-person with family or friends by casual gamers, making it more family member friendly. Conversely, a typical PS5 game is tailored for hardcore gamers and intended to be played alone in a room. Therefore, it is possible that using PS5 could potentially contribute to a relatively increased occurrence of disagreements on video game usage among family members, resulting in reduced psychological benefits for adolescents (supportive evidence in Fig. 4c,d). These findings emphasize the importance of additional research into the diverse impacts of media use on well-being

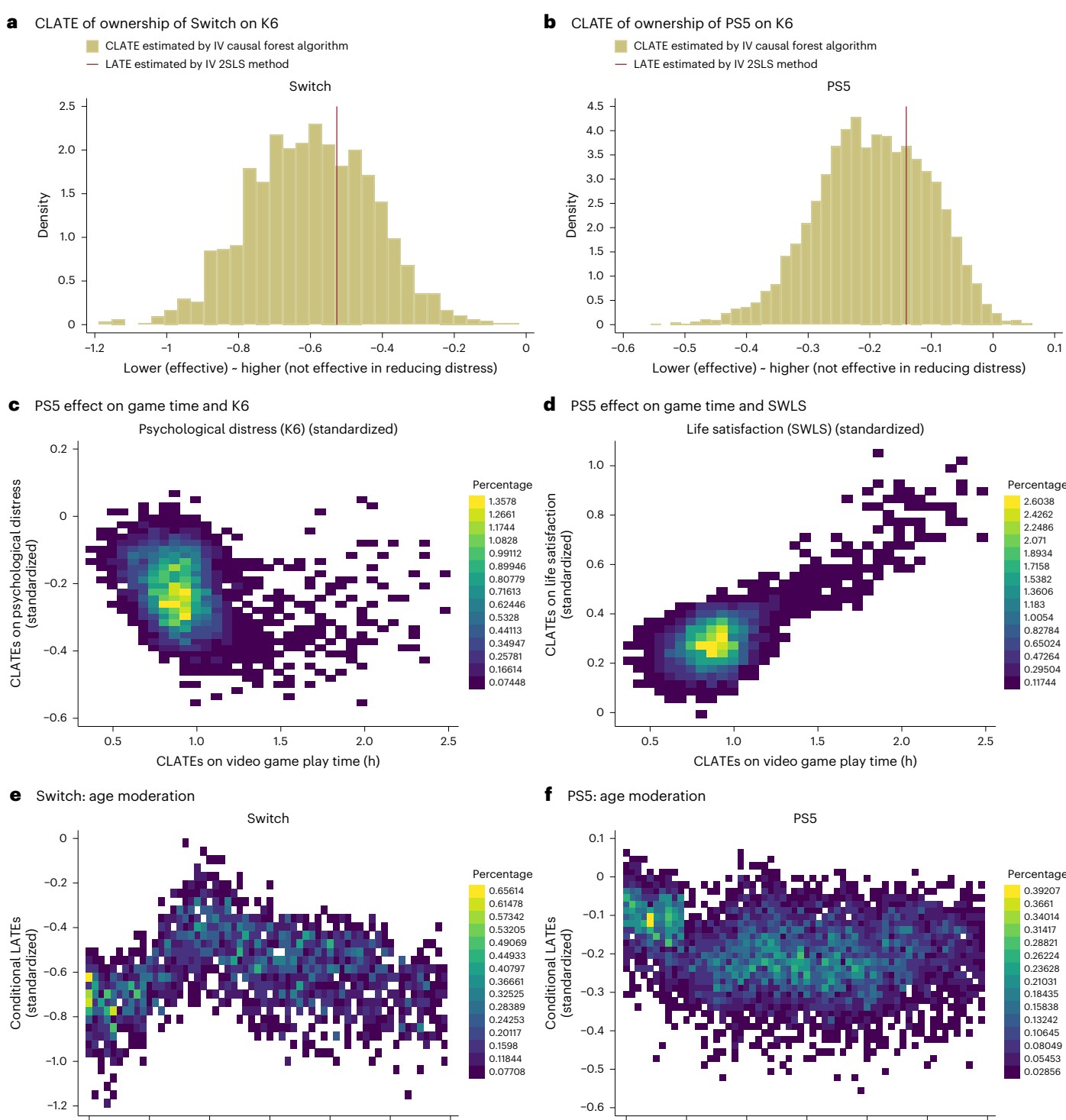

**Fig. 3 | Machine learning results illustrating estimated CLATEs ($N$ = 8,192).** The IV causal forest is used to estimate CLATEs of game console ownership on various outcome variables. **a,b**, Switch (**a**) and PS5 (**b**) histograms of the estimated CLATEs on K6. The vertical red lines in **a** and **b** indicate LATEs computed by the IV method, which uses a set of covariates common to those used in the causal forest (Supplementary Table 16). **c–f**, Two-dimensional histogram heat plots that display the frequencies of binned values on the *y*-axis and *x*-axis variables as rectangular fields using a colour gradient (where a brighter colour signifies a larger number of frequency occurrences in the two-dimensional

histograms): the association between two CLATEs (CLATEs on game time and CLATEs on mental health (**c**), and CLATEs on game time and CLATEs on life satisfaction (**d**)), supporting the positive causal link between video gaming and mental well-being; Switch (**e**) and PS5 (**f**) show age modification effects, with the level of video game benefits (CLATEs) on the vertical axis and age along the horizontal axis, with a larger negative value denoting a greater reduction in PD. The estimates for K6 and SWLS are standardized by the s.d. in all the panels. Machine learning results for other characteristics are shown in Fig. 4 and Supplementary Fig. 18.

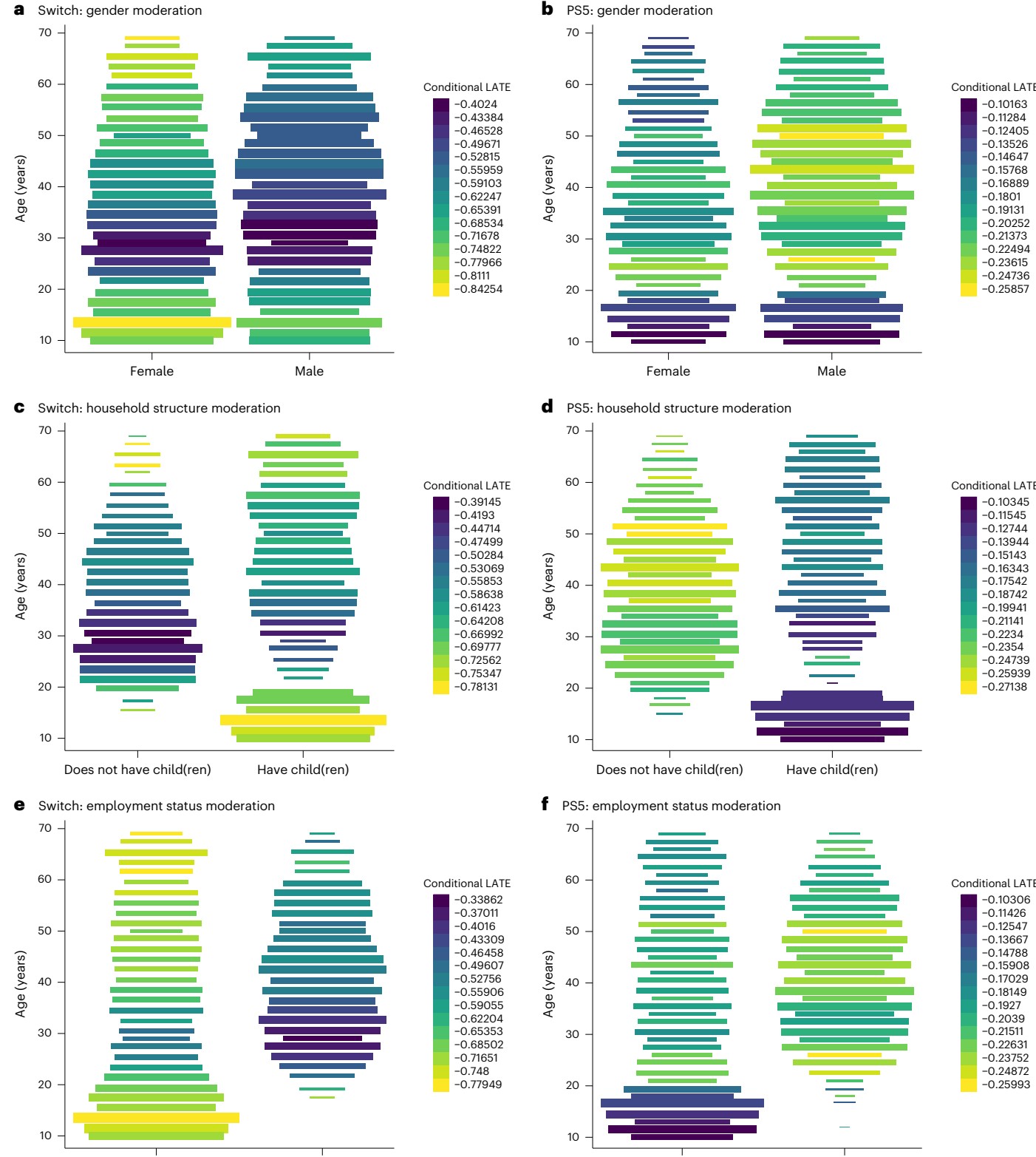

**Fig. 4 | Machine learning results of effect modification contrasting ownership of Switch and PS5 ($N$ = 8,192).** The IV causal forest was used to estimate CLATEs of game console ownership on K6. Each plot is a heat map of a trivariate distribution where the colour gradient visualizes the average value of $z$ within $y$-axis and $x$-axis bins of rectangular fields. The estimated CLATEs of video game console ownership on K6 are illustrated, with age on the $y$ axis and each background characteristic on the $x$ axis. A lighter shade indicates that video game ownership is more advantageous for individuals in that particular bin. The bar width of the heat maps represents the sample size for each age group. The estimates are standardized by the s.d. **a,b**, Gender moderation for Switch (**a**) and PS5 (**b**). **c,d**, Household structure moderation for Switch (**c**) and PS5 (**d**). **e,f**, Employment status moderation for Switch (**e**) and PS5 (**f**).

and the mechanisms behind them. Policymakers should be mindful of designing regulations or interventions that consider the differential effects of various types of screen time and gaming platforms/genres on individuals' well-being.

Previous research has proposed mechanisms for both positive and negative effects of video gaming on mental well-being (summarized in Supplementary Table 11). Our study found that positive effects outweigh negative effects, resulting from both positive and negative pathways. Positive pathways include psychological therapy games, mood management theory, self-determination theory and social connection hypothesis[51–54]. For example, relaxation games can induce a positive mood and improve well-being, as in psychological therapy. An example of an adverse pathway is playing video games to such an extent that it results in insufficient sleep, possibly harming well-being[55,56]. While evaluating the relative impact of each pathway is beyond the scope of this study, understanding potential mechanisms from the literature is essential. Given that data collection occurred during the COVID-19 pandemic, the context may affect how these pathways operate. Thus, interpreting our estimates requires considering the possible effects that the COVID-19 circumstances might exert on these potential pathways.

Our study encountered the prevalent issue of survey non-response, achieving a response rate of 59%. This rate aligns with those observed in other lottery-based natural experiment studies. For example, response rates of 46% were reported by Imbens et al.[57], 55% by Doherty et al.[58] and 32% by Kuhn et al.[59]. Additionally, online surveys like ours generally experience low response rates, with an average of 44% reported[60].

This study dealt with the issue of survey non-responses, primarily leading to biased causal effect estimates and reduced generalizability of the findings. Firstly, employing PSM aids in mitigating potential bias. For example, PSM can enhance covariate balance and ensure common support, as illustrated in Supplementary Figs. 2–5 (further rationale of employing PSM available in Methods). It is also important to note that natural experiments are generally less susceptible to non-response bias than non-experimental observational studies, as highlighted in the previous studies[61].

Secondly, regarding generalizability, we assessed the potential systematic differences in background characteristics across the entire sample between respondents and non-respondents. The comparison showed modest differences; thus, it is plausible to infer that the systematic difference caused by non-responses in our analysis was small, affecting the generalizability of our findings only to a limited extent. Note that using the entire sample for the comparison is the next best alternative. Ideally, a comparison between respondents and non-respondents within the analysis sample—lottery participants—would be more direct, but such data were unavailable for non-respondents in our study.

This study has several limitations. Foremost among them is the concern regarding the external validity of our findings. While our analysis of gaming behaviours in natural settings enhances external validity relative to laboratory studies, the data were collected during the exceptional circumstances of the COVID-19 pandemic. This period was marked by high levels of mental distress and reduced opportunities for physical activity[62,63], which may influence our estimates. For example, our quantile regressions underscored a more pronounced gaming effect among individuals experiencing high distress. Additionally, with fewer opportunities for physical activity during the pandemic, individuals might have been less likely to substitute gaming for physical activities (if there is no exercise to begin with, gaming cannot reduce exercise time), a potential adverse pathway on well-being associated with increased gaming, as documented in Supplementary Table 11. Given the particular context of the pandemic, our current estimates may be higher compared with those obtained in pre- or post-pandemic settings.

Another limitation is the generalizability of our findings. Our estimates reflect the video gaming impact for individuals who bought a Switch/PS5 after winning the lottery, accompanied by 0–4 h longer video game time (Supplementary Fig. 13). Consequently, the study implications might not directly apply to dramatically different contexts. For instance, our study cannot provide insights into the influence on a person who rarely played games but suddenly adopted an extremely long gaming practice (elaborated in Supplementary Result 7). Moreover, the applicability of our results to users of other gaming platforms, including different consoles and smartphones, remains uncertain. Additionally, the previously discussed challenges to our findings' generalizability, stemming from survey non-responses, necessitate careful interpretation in broader contexts. In particular, the potential underrepresentation of certain demographics or gaming behaviour subgroups due to non-responses might conceal varied effects of video gaming, deviating from the patterns we observed among our respondents.

Data limitations in our study necessitate caution, as they may affect the conditional unconfoundedness assumption—specifically, the concern related to a potential confounder: the number of non-winning lottery participations (Supplementary Figs. 8 and 9, displaying causal diagrams, and further elaborated in Supplementary Method 2.8 and Supplementary Result 5). Ideal data collection would occur separately for each 'turn' of the console lottery, facilitating straightforward comparisons between the treatment and control groups within each distinct lottery turn. This strategy would aid in drawing more rigorous causal inferences. While acknowledging the limitations, we posit that the short-period subsample analysis (Supplementary Fig. 7) offers mitigation of the concerns, lending credibility to our findings. Lastly, reliance on self-reported data for gaming behaviour and well-being measures presents a limitation, although using a natural experimental study design alleviates concerns regarding the self-reporting bias; the random variation caused by the lottery is unlikely to be correlated with the self-reporting bias.

Along with the aforementioned limitations, this study possesses several strengths. First, the empirical identification strategy employed a natural experimental design that underwent rigorous robustness checks, ensuring reliable causal evidence. Second, our large sample of respondents, ranging in age from 10 to 69 years, enabled effective investigation into effect modifications using a machine learning approach, leveraging the extensive respondent background information available. Lastly, by capturing actual video game usage in the nationwide survey data, our study enhances external validity compared with laboratory experiments, offering insights more aligned with real-world gaming behaviours.

Our natural experiment showed that video gaming positively impacted mental well-being, but gaming for over 3 h had decreasing psychological benefits. Furthermore, the magnitude of the gaming effect was revealed to be influenced by various socioeconomic factors such as gender, age, job and family structure. Moreover, the effect modification of Switch, based on these sociodemographic characteristics, substantially differed from that of PS5. These findings highlight the necessity for further research into the mechanisms underlying video gaming's effects on mental well-being and point to the importance of policy design that considers the differential effects of various digital media screen time for diverse populations.

## Methods

### Ethics statement
This study complies with all relevant ethical regulations for research involving human participants. The survey was approved by the institutional review board of Takasaki City University of Economics (approval number 245-1). Informed consent was obtained from all participants by the survey agency before the interview. All data were kept confidential and used only for research purposes. The study posed minimal risk to

participants, and the participants' privacy was protected throughout the study. Data were anonymized to protect the participants' privacy. Although the survey agency compensated the participants, the details of this payment were not disclosed to the research team. This study was not pre-registered. A preliminary version of this paper, which contained a partial analysis and preliminary findings, was previously presented at the Proceedings of the Annual Conference, Digital Game Research Association JAPAN, 13th annual conference[64].

### Participants and data collection

The study participants included people aged 10–69 years living in Japan. In partnership with the gaming market research firm gameage R&I (GRI), we conducted five rounds of omnibus online surveys from December 2020 to March 2022 among individuals aged 10–69 ($n = 97,602$) across all 47 prefectures in Japan. A stratified random sampling technique—stratified by age, gender and video gaming preferences—was used to recruit participants from a survey agency's pool of roughly 150,000 respondents. The response rate was 59.3% (details in Supplementary Table 12 and Supplementary Method 1). We collected information on participants' lottery participation, video game ownership, gaming preferences, mental health, life satisfaction and sociodemographic characteristics. The dataset is a blend of panel and repeated cross-sectional observations, with 35.90% of respondents participating multiple times. This data structure facilitates robustness checks, including placebo analysis.

After excluding individuals who did not participate in a Switch lottery in round 1 and those who did not participate in a PS5 lottery in rounds 2–5, the analysis sample for our causal inference comprised 8,192 observations answered by individuals aged 10–69 years (Supplementary Fig. 15). Further details on sampling and data collection are provided in Supplementary Method 1 and Supplementary Figs. 15 and 16.

### Measurements

**Exposures.** The main exposure was video game engagement (additional information in Supplementary Table 13). Respondents were asked if their households owned a Switch and a PS5, respectively, and whether they had played each gaming device in the past 30 days. Additionally, respondents reported the amount of time spent playing video games on weekdays and weekends (asked separately) over the past 30 days. Moreover, we collected data separately on time spent playing (1) video games on a TV or computer (encompassing Switch and PS5) and (2) smartphone games.

**Outcomes.** The primary outcomes of interest were two facets of well-being, specifically mental health and life satisfaction. The Japanese translation of the K6 was used to measure mental health status among participants[65]. This scale features six items that gauge the level of nonspecific PD over the past 30 days, with a total score ranging from 0 (minimum distress) to 24 (maximum level of distress). Given its brevity and high reliability, the K6 is employed in population-based health surveys in Japan as a screening tool for mental disorders[66]. Life satisfaction was measured using the Japanese translation of the SWLS[67,68]. This 5-item scale is scored between 1 (completely disagree) and 7 (completely agree), with a total score ranging from 5 (lowest) to 35 (highest). The SWLS is a reliable and validated instrument for assessing overall life satisfaction in Japan. The outcomes and exposures collected in each round are found in Supplementary Table 14.

**Covariates.** We employed a set of covariates (Supplementary Table 15) including those of respondents: age, gender, employment status, residential prefecture, number of times that respondent households entered game console lotteries, and a five-category video gaming preference scale: (1) hardcore gamer, (2) core gamer, (3) mid-core gamer, (4) casual gamer and (5) non-gamer. Additionally, we utilized

the characteristics of either respondents or caregivers as covariates: marital status, having children (yes, no) and occupation. The research firm GRI assessed video gaming preferences using a clustering algorithm based on factors such as gaming frequency and game software purchases, and classified participants into five groups.

**Statistical analysis.** We used three methods: multivariate regression, PSM approach and IV method. We additionally utilized a machine learning algorithm called causal forest (or generalized random forests, GRFs)[39,69]. Methodological particulars are provided in Table 2, while the underlying causal assumptions are discussed in Supplementary Methods 2.1 and 2.9. The methodological challenges encountered in lottery-based natural experiments and the rationale for employing PSM alongside linear regression are elaborated in a subsequent subsection.

**Multivariate regression and PSM.** Using a dichotomous variable indicating possession of a Switch/PS5 as the variable of interest under study may be affected by confounding bias. Rather, we took advantage of the natural experimental circumstances resulting from the supply constraints of these game consoles.

To estimate the causal effect of winning console lotteries—an ITT analysis or a reduced form analysis, we employed multivariate regression (equation (1) in Supplementary Methods) and PSM used in previous studies evaluating the impact of lottery wins—Imbens et al.[57] and Imbens (2015)[43]. We employed a binary variable indicating whether a participant won the lottery, serving as the treatment assignment variable. We assumed unconfoundedness for the game console lotteries, having controlled for the number of times of lottery participation. All statistical tests, including those in subsequent analyses, were two-tailed.

We also hypothesized that purchasing new video game consoles would increase video game play time and examined the hypothesis. Moreover, we tested the time spent on smartphone gaming as an additional outcome.

It is worth noting that the causal inference literature has acknowledged the limitations of relying exclusively on OLS linear regression in observational studies[42–44]. It is recommended to compare the OLS estimates with those derived from other sophisticated methods, such as matching[43,44]. This approach ensures that the results are not sensitive to the choice of estimators.

**Assessing natural experiment validity.** We evaluated the unconfoundedness of the game console lotteries—natural experiment validity—using two methods[43]. First, we assessed the balance of baseline characteristics through standardized differences. Second, we performed pseudo-outcome tests by utilizing pre-lottery values of well-being as pseudo-outcomes (details in Supplementary Method 2.7).

We further investigated a potential unobserved confounder in the causal relationship between lottery success and well-being: the number of times that a survey respondent participates in the lottery, conditional on not winning it. This variable was not controlled for in our primary analysis owing to its unobservable nature. To assess the potential risk of violating the conditional unconfoundedness assumption due to this unobserved variable, we employed three strategies: (1) an imputation approach, (2) analyses of short-period subsamples and (3) the application of causal diagrams for consideration of covariates. Methodological details are provided in Supplementary Method 2.8.

**Methodological challenge in lottery-based natural experiment.** Natural experiment studies using lotteries, including ours, have advantages regarding internal validity over typical (non-experimental) observational studies as they exploit plausibly random variations. However, it should be noted that lottery studies are not the same as fully randomized experiments. Even if a lottery randomly determines the winners, there may be systematic differences within the subset of

individuals who responded to the survey and are included as the analysis sample. Moreover, most lottery studies are not even 'the ideal lottery study'—argued by Doherty et al.[58]—that examines those who purchased the same number of lottery tickets in a given game. Therefore, we need to address several methodological challenges.

In a randomized experiment, every subject has the same probability of receiving a treatment. Yet, most lottery studies do not meet this criterion and attempt to address the problem. A typical approach is controlling the source of having different probabilities of winning a treatment. In Imbens et al. and Doherty et al.[57,58], two well-cited lottery studies, the number of lottery tickets purchased was controlled in their regression analyses. Note that both papers exploit lotteries in the USA; individuals who won the lotteries obtained prizes (the outcome variables are labour earnings in the former paper and attitude towards government redistribution in the latter paper). The number of tickets bought is included as a covariate because, by the nature of the lottery, individuals purchasing more tickets are more likely to win a lottery; therefore on a priori grounds, researchers expect that the variable affects the probability of winning a lottery. In our study, we also treated the number of times respondents participated in lotteries as a covariate. However, these methods simplify the actual complexities—differential lottery participating behaviour—involved in each lottery scenario.

Natural experiment studies using lotteries are not free from the methodological challenge stemming from differential lottery participating behaviour[43,57,58]. For example, Imbens et al.[57] and Doherty et al.[58] treated different types of lottery (or lottery tickets) as identical in their OLS regression models (for example, the 'season ticket' buyers and 'single-ticket' buyers are treated as comparable in Imbens et al.[57]). In our study's context, we also had to consider various lotteries as equivalent, even though they might offer different odds of winning, such as different Switch lotteries or PS5 lotteries. Despite all targeting a chance to buy a Switch or a PS5, each lottery had its nuances, such as varying retailers (for example, Amazon, Yodobashi Camera and Sony Store) or timings. These differences could affect the likelihood of winning. For instance, as the supply-side challenge for the Switch started in March 2020 and ended in January 2021, winning odds probably shifted over time. Therefore, each lottery could have different probabilities of winning the lottery, and the number of times of lottery participation reflects the differential lottery participating behaviour of the respondents. Further context information is available in Supplementary Method 1.

When a study is not 'the ideal lottery study'[58] examining those who purchased the same number of lottery tickets in a given game, assuming a simple linear relationship for the number of lottery entries in a regression model may become problematic. It is acknowledged that, on the basis of the context information, we a priori know that this variable should affect the propensity score. However, the precise functional form of this variable remains uncertain. It may exhibit a quadratic or more complex polynomial relationship, or involve interaction effects with other variables, among other possibilities. In such scenarios, matching methods help tackle these methodological challenges[43,44].

Imbens (2015)[43]—written on the basis of Imbens and Rubin (2015)[44]—discussed the application of PSM for estimating causal effects and highlighted the 'Imbens–Rubin–Sacerdote lottery' study (Imbens et al.)[57] as a prime example where PSM could effectively complement traditional OLS regression analyses. Applying the recommended PSM procedures to the data from Imbens et al.[57], Imbens demonstrated the robustness of the original findings. The recommended practices include estimating the propensity score in a data-driven manner, improving overlap by trimming units with extreme values of the propensity score and assessing the plausibility of the unconfoundedness assumption.

In both Imbens et al.[57] and Imbens (2015)[43], the authors argued two major methodological challenges in the Imbens–Rubin–Sacerdote

lottery study. The first challenge is the differential ticket buying behaviour, which implies that the inclusion of the number of tickets bought as a covariate in regression analysis might not fully address potential confounders. The second challenge is non-responses to the survey (response rate 46% in Imbens et al.[57]), which could lead to systematic differences between lottery winners and non-winners. Although the lottery mechanism allows randomization, the sample of individuals who respond to the survey would not be random. In addition to those, we may consider self-reported ticket purchasing behaviour as another potential source of bias, a problem that does not arise in randomized experiments with direct access to administrative data.

Therefore, our strategy to tackle those methodological challenges is to incorporate PSM alongside regression analysis, following the procedure recommended by Imbens (2015)[43]. This reduces reliance on the specific assumptions of the OLS regression model, thereby mitigating the bias caused by the differential ticket buying behaviour. Additionally, it enhances the overlap in the covariate distribution—one of the critical concerns of using linear regression. Using PSM, we can better ensure that the matched groups are comparable in their observable characteristics, as shown in Supplementary Figs. 2–5 (ref. [42]). Thereby, our approach helps to lessen potential biases caused by non-response and self-reporting. It should be noted that PSM relies on the propensity score model's functional form and does not address bias due to unobserved confounders as caveats.

**IV regressions.** Employing the two-stage least squares (2SLS) estimation technique, we next estimated the causal impact of possessing a Switch/PS5—LATE. We addressed the unmeasured confounding bias by treating the lottery wins (dummy variables representing whether an individual won a lottery for a Switch or a PS5, respectively) as excluded instruments. As exposure variables, ownership of a Switch/PS5, playing Switch/PS5 last month, and game play time were examined. The exclusion restriction is plausibly satisfied by the lottery-winning instruments for ownership, but reservations exist for game-play-related variables. Further details are in Supplementary Method 2.5.

**Machine learning.** Causal forest (or GRFs)[39,69] predicts treatment effects for individuals using their specific characteristics. This technique, employed in recent research[70,71], allows for flexible visualization and facilitates the exploration of effect modification (or moderating effects). We specifically used IV causal forest[39] evaluated by a recent study[72] to estimate CLATEs, capturing the effect of ownership of a Switch/PS5. While causal forests estimate conditional average treatment effects (CATEs), IV causal forests estimate CLATEs that match CATEs only when the conditional homogeneity assumption holds.

To explore the multidimensional nature of the video gaming effect's heterogeneity, we selected GRFs—specifically, its IV causal forests component—formulated by Athey et al.[39]. Several machine learning algorithms are suited for analysing heterogeneous treatment effects[73,74], yet a few incorporate IV techniques[39,75] essential for causal inference amidst imperfect compliance. The reliability and usefulness of GRFs are further supported by the literature[72,76–80], which includes empirical studies[70,71,81–88] and methodological guides[78,85,89]. Note that heterogeneous treatment effect estimation via IVs using the GRF is called in several ways: IV forests[39], instrumental forests[90] or IV causal forests[72].

GRFs involve generating a series of causal trees[91] from randomly selected data subsets, with each tree comprising a series of decision splits based on specific variables and cut-off values. The algorithm selects splits that maximize the difference in treatment effects, continually partitioning the data until it forms groups, or 'leaves,' with similar effects. For counteracting overfitting, the algorithm uses the selected subset to construct the tree while it estimates treatment effects using the other subgroup. Recognizing the potential for instability in the

estimates from individual trees, GRFs compute many trees, each utilizing different samples and variables, to create a complete forest. This ensemble approach ensures the robustness and consistency of the treatment effect estimates across various subsamples.

In GRFs' unique methodology, a crucial pre-processing step called 'labelling' precedes the standard classification and regression tree (CART) regression splits[39]. This step involves encoding the specific structure of the type of problem to be solved, a process enabling the integration of IVs into the forest framework. By doing so, it tailors the algorithm to the specific characteristics of both the data and the question being addressed. Utilizing its encoded information, the algorithm optimizes its criterion. This facilitates the application of random forests to model a wide array of quantities of interest identified as the solutions to respective local moment equations.

### Reporting summary

Further information on research design is available in the Nature Portfolio Reporting Summary linked to this article.

### Data availability

Data analysed in this study are not openly available due to usage restrictions and licensing agreements with GRI. Certain variables (that is, video game console ownership and video gaming preference measures) are the proprietary information of GRI and are not available for public dissemination. However, the data are available upon request by accredited academic researchers from the corresponding author and with permission from GRI.

### Code availability

In this study, we utilized standard methods and did not rely on custom code or specialized mathematical algorithms. The code associated with the major analyses presented in this manuscript (software: STATA 16.1, R version 4.3.1 with R package 'grf', version 2.3.2.) is found on GitHub (https://anonymous.4open.science/r/game-6B46/).

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

## Acknowledgements

We gratefully acknowledge funding from the following organizations: JSPS KAKENHI (grant numbers JP19K13804, T.W.; JP24K20909, H.E.), the Takasaki City University of Economics Grant-in-Aid for Encouragement of Social Scientists (T.W.), The Telecommunications Advancement Foundation (H.E.) and the Nihon University College of Economics Grant-in-Aid for Encouragement of Social Scientists (H.E.). The funders had no role in study design, data collection and analysis, decision to publish or preparation of the manuscript. We thank T. Matsumoto, R. Goto, S. Litschig, M. Takahashi, M. Alistair, T. Kuroda, S. Yamaguchi, T. Tanaka, J. Goto, M. Matsushima, K. Takahashi, Y. Kijima, K. Kawata and K. Ono for insightful comments and feedback; Y. Yoshinari for technical and graphical support; Y. Yamaki for superb research assistance; H. Kanemitsu and T. Kinoshita for precious feedback on our survey questionnaire; and GRI for conducting omnibus surveys together with us and sharing invaluable data.

## Author contributions

H.E., M.S.R., T.Y., C.E. and T.W. had full access to all of the data in the study and took responsibility for the integrity of the data and the accuracy of the data analysis. Concept and design: H.E., T.Y., C.E. and T.W. Drafting of the manuscript: H.E., M.S.R., T.Y. and C.E. Critical revision of the manuscript for important intellectual content: M.S.R. Statistical analysis: H.E., T.Y. and C.E. Obtained funding: H.E., T.Y., C.E. and T.W. Administrative, technical or material support: H.E., T.Y., C.E. and T.W. Supervision: H.E. and M.S.R.

## Competing interests

The authors have no competing interests as defined by Nature Portfolio, or other interests that might be perceived to influence the results and/or discussion reported in this paper.

## Additional information

**Correspondence and requests for materials** should be addressed to Hiroyuki Egami.

# Reporting Summary

## Statistics

For all statistical analyses, confirm that the following items are present in the figure legend, table legend, main text, or Methods section.

| n/a | Confirmed | |
|---|---|---|
| ☐ | ☒ | The exact sample size (*n*) for each experimental group/condition, given as a discrete number and unit of measurement |
| ☐ | ☒ | A statement on whether measurements were taken from distinct samples or whether the same sample was measured repeatedly |
| ☐ | ☒ | The statistical test(s) used AND whether they are one- or two-sided<br>*Only common tests should be described solely by name; describe more complex techniques in the Methods section.* |
| ☐ | ☒ | A description of all covariates tested |
| ☐ | ☒ | A description of any assumptions or corrections, such as tests of normality and adjustment for multiple comparisons |
| ☐ | ☒ | A full description of the statistical parameters including central tendency (e.g. means) or other basic estimates (e.g. regression coefficient) AND variation (e.g. standard deviation) or associated estimates of uncertainty (e.g. confidence intervals) |
| ☐ | ☒ | For null hypothesis testing, the test statistic (e.g. *F*, *t*, *r*) with confidence intervals, effect sizes, degrees of freedom and *P* value noted<br>*Give P values as exact values whenever suitable.* |
| ☒ | ☐ | For Bayesian analysis, information on the choice of priors and Markov chain Monte Carlo settings |
| ☒ | ☐ | For hierarchical and complex designs, identification of the appropriate level for tests and full reporting of outcomes |
| ☐ | ☒ | Estimates of effect sizes (e.g. Cohen's *d*, Pearson's *r*), indicating how they were calculated |

*Our web collection on statistics for biologists contains articles on many of the points above.*

## Software and code

Policy information about availability of computer code

| Data collection | No specific software was used for data collection. Respondents filled out the questionnaire using their web browser. |
|---|---|
| Data analysis | STATA 16.1; R version 4.3.1; R package "grf", version 2.3.2. |

For manuscripts utilizing custom algorithms or software that are central to the research but not yet described in published literature, software must be made available to editors and reviewers. We strongly encourage code deposition in a community repository (e.g. GitHub). See the Nature Portfolio guidelines for submitting code & software for further information.

## Data

Policy information about availability of data

All manuscripts must include a data availability statement. This statement should provide the following information, where applicable:
- Accession codes, unique identifiers, or web links for publicly available datasets
- A description of any restrictions on data availability
- For clinical datasets or third party data, please ensure that the statement adheres to our policy

Data analyzed in this study are not openly available due to usage restrictions and licensing agreements with gameage R&I. Certain variables (i.e., video game console ownership and video gaming preference measures) are the proprietary information of gameage R&I and are not available for public dissemination. However, the data are available upon request by accredited academic researchers from the corresponding author and with permission from gameage R&I.

# Research involving human participants, their data, or biological material

Policy information about studies with <u>human participants or human data</u>. See also policy information about <u>sex, gender (identity/presentation), and sexual orientation</u> and <u>race, ethnicity and racism</u>.

| | |
|---|---|
| Reporting on sex and gender | Gender was considered in the heterogeneity analysis. Gender was determined based on self-reporting. Informed consent was obtained from all participants by the survey agent prior to the interview. In the case of minor participants, informed consent was taken from parents or legally authorized representatives. |
| Reporting on race, ethnicity, or other socially relevant groupings | N/A |
| Population characteristics | Japanese population aged 10-69, including half male and half female. |
| Recruitment | We conducted online omnibus surveys with gameage R&I, a gaming market research firm. Individuals who have pre-registered at a survey agency Cross Marketing (a pool of roughly 150,000 respondents) were sent survey offers. The stratified random sampling technique (stratified by gender, age, and gaming preference) was used. While voluntary participation in surveys can introduce self-selection bias, our comparison of respondents and non-respondents showed similar characteristics, mitigating this concern. It is acknowledged that individuals registered with the survey agency might differ from those who did not register, presenting an inherent challenge of online surveys. However, by stratifying based on gaming preference, we mitigated the concern of attracting a disproportionate number of avid gamers to our survey. Additionally, our study might have yielded context-based estimates as the data was collected during the COVID-19 period (2020-2022). |
| Ethics oversight | The survey was approved by the institutional review board of Takasaki City University of Economics (approval number 245-1). Informed consent was obtained from all participants by the survey agency prior to the interview. In the case of minor participants, informed consent was taken from parents or legally authorized representatives. All data was kept confidential and used only for research purposes. The study posed minimal risk to participants, and the participants' privacy was protected throughout the study. Data was anonymized to protect the participants' privacy. |

Note that full information on the approval of the study protocol must also be provided in the manuscript.

# Field-specific reporting

Please select the one below that is the best fit for your research. If you are not sure, read the appropriate sections before making your selection.

☐ Life sciences ☒ Behavioural & social sciences ☐ Ecological, evolutionary & environmental sciences

For a reference copy of the document with all sections, see nature.com/documents/nr-reporting-summary-flat.pdf

# Behavioural & social sciences study design

All studies must disclose on these points even when the disclosure is negative.

| | |
|---|---|
| Study description | This is a natural experimental study identifying the causal effects of video gaming on mental well-being. The study utilized a unique context where major gaming consoles, Nintendo Switch and PlayStation 5, were distributed through lotteries due to supply shortages. This setup provided a near-random variation in gaming console ownership, which we leveraged to estimate causal effects. |
| Research sample | Japanese population aged 10-69, including half male and half female. 21.3% were students, 10.7% were unemployed, 39.1% were full-time employees, and 57% were married. 16.0% were Hardcore gamers, 20.3% were Core gamers, 23.3% were Middle-core gamers, 17.6% were Casual gamers, and 22.9% were Non-gamers. Individuals who have pre-registered at the survey agency Cross Marketing were sampled. We intend to study the impact of video gaming on well-being, and thus, the sample is not nation-representative and rather includes more video gamers. |
| Sampling strategy | We conducted five rounds of omnibus online surveys with a market research firm gameage R&I. Due to the nature of the natural experiment, we did not predetermine the minimum sample size before the survey. The number of participants who entered the lottery and the number of winners were not known before the study, as these aspects were managed externally by the lottery organizers. Moreover, before conducting the study, we did not have an expectation of the effect size of winning the lottery on our outcome variables. The survey sample size (shown in the supplementary material) varied each month based on the business objective of gameage R&I, the research firm conducting the monthly survey. The survey was distributed to a large, stratified random sample of individuals pre-registered with the survey agency, ensuring representation across gender, age, and gaming preferences. This approach maximizes the likelihood of capturing a broad and representative subset of the population. The sufficiency of our sample size was assessed post hoc based on the statistical power and the precision of our estimates. |
| Data collection | Gameage R&I (GRI), a research firm, conducts regular monthly surveys of individuals who have pre-registered through the survey agency Cross Marketing. Their survey aims to gather consumer data specifically related to the video game industry. Additional questions of ours were incorporated throughout five surveys. Particularly, mental well-being measures (K6 and SWLS) were collected. Respondents filled out the questionnaire using their web browser. In the case of minor participants, their parents or legally authorized representatives filled out the questionnaire. [Presence of others] The data collection process was conducted online, ensuring that responses were provided in a private setting without the direct presence of researchers.[Blindness for researchers] As |

the respondents answered online surveys, the researchers were initially blinded to the participants' lottery outcomes when collecting the data. However, the researchers knew whether the participants won or lost the lottery when they analyzed the data. The researchers were also aware of the study hypotheses. [Blindness for respondents] Participants were aware of whether they won the lottery or not, as this directly impacted their experience and subsequent behavior. Participants were not informed about the specific hypotheses regarding the effects of winning the lottery on mental well-being.

| | |
|---|---|
| Timing | The survey respondents answered on 2-6 December 2020, 1-4 March 2021, 3-6 May 2021, 1-5 November 2021, and 2-7 March 2022. |
| Data exclusions | No data were exlcluded. |
| Non-participation | The survey response rate was 59.3%. |
| Randomization | The natural experimental study design using video game console lotteries works as randomization. We provide balance tables of a number of covariates. The multivariate regression and propensity score matching approach were used to address potential confounders. |

# Reporting for specific materials, systems and methods

We require information from authors about some types of materials, experimental systems and methods used in many studies. Here, indicate whether each material, system or method listed is relevant to your study. If you are not sure if a list item applies to your research, read the appropriate section before selecting a response.

## Materials & experimental systems

| n/a | Involved in the study |
|---|---|
| ☒ | Antibodies |
| ☒ | Eukaryotic cell lines |
| ☒ | Palaeontology and archaeology |
| ☒ | Animals and other organisms |
| ☒ | Clinical data |
| ☒ | Dual use research of concern |
| ☒ | Plants |

## Methods

| n/a | Involved in the study |
|---|---|
| ☒ | ChIP-seq |
| ☒ | Flow cytometry |
| ☒ | MRI-based neuroimaging |

