## [Peer Review File · Nature Human Behaviour]

Peer Review Information

Journal: Nature Human Behaviour

Manuscript Title: Causal effect of video gaming on mental well-being in Japan 2020-2022

Corresponding author name(s): Hiroyuki Egami

Reviewer Comments & Decisions:

Decision Letter, initial version:

14th August 2023

Dear Dr Egami,

Thank you once again for your manuscript, entitled "Causal effect of video game play on mental well-being: a quasi-experimental study among Japanese population," and for your patience during the peer review process.

Your manuscript has now been evaluated by 3 reviewers, whose comments are included at the end of this letter. Although the reviewers find your work to be of interest, they also raise some important concerns. We are interested in the possibility of publishing your study in Nature Human Behaviour, but would like to consider your response to these concerns in the form of a revised manuscript before we make a decision on publication.

To guide the scope of the revisions, the editors discuss the referee reports in detail within the team, including with the chief editor, with a view to (1) identifying key priorities that should be addressed in revision and (2) overruling referee requests that are deemed beyond the scope of the current study. We hope that you will find the prioritised set of referee points to be useful when revising your study. Please do not hesitate to get in touch if you would like to discuss these issues further.

1. Reviewer 2 raises concerns about the conditional ignorability assumption for causal inference not being satisfied. Please carefully revise your manuscript to address this concern, and conduct additional analyses if necessary.
2. Reviewer 2 also mentions concerns about external validity. We agree and ask that you ensure that the interpretation of your results (including in the Abstract, Introduction and Discussion) considers the issue of reduced external validity due to the data collection occurring during the pandemic, acknowledging that the findings might not generalize to the post-COVID-19 period. In your title, please specify the country and years of data collection; however, please do not mention COVID-19 in the title itself as this may suggest that your data speak to pandemic-specific trends, and in the absence of comparison data this is not the case.
3. Reviewer 1 makes important points about the meaningfulness of effect sizes. Please follow Reviewer 1's suggestion and adopt the smallest effect size of interest.
4. Reviewer 2 comments on grammatical correctness. Where grammatical errors hinder understanding and transparency please do correct these; however, we will accept a revised manuscript containing grammatical errors as long as these do not hinder our (and our reviewers') ability to assess the scientific content.

Finally, your revised manuscript must comply fully with our editorial policies and formatting

requirements. Failure to do so will result in your manuscript being returned to you, which will delay its consideration. To assist you in this process, I have attached a checklist that lists all of our requirements. If you have any questions about any of our policies or formatting, please don't hesitate to contact me.

In sum, we invite you to revise your manuscript taking into account all reviewer and editor comments. We are committed to providing a fair and constructive peer-review process. Do not hesitate to contact us if there are specific requests from the reviewers that you believe are technically impossible or unlikely to yield a meaningful outcome.

We hope to receive your revised manuscript within two months. I would be grateful if you could contact us as soon as possible if you foresee difficulties with meeting this target resubmission date.

- Include a "Response to the editors and reviewers" document detailing, point-by-point, how you addressed each editor and referee comment. If no action was taken to address a point, you must provide a compelling argument. When formatting this document, please respond to each reviewer comment individually, including the full text of the reviewer comment verbatim followed by your response to the individual point. This response will be used by the editors to evaluate your revision and sent back to the reviewers along with the revised manuscript.
- Highlight all changes made to your manuscript or provide us with a version that tracks changes.

[REDACTED]

We look forward to seeing the revised manuscript and thank you for the opportunity to review your work. Please do not hesitate to contact me if you have any questions or would like to discuss these revisions further.

Sincerely,

[REDACTED]

Reviewer expertise:

Reviewer #1: video gaming and well-being

Reviewer #2: video gaming and well-being ; causal inference

Reviewer #3: Random Forest algorithms ; machine learning ; cyberpsychology

REVIEWER COMMENTS:

Reviewer #1:

Remarks to the Author:

I appreciated the opportunity to read this study which I think could be a good contribution to the

literature pending a few revisions. My comments are below.

The authors do a nice job balancing the literature review. As one comment the authors claim there's no conclusive answer to the video game violence question, and they are quite right to note there is debate of course. However, much like other areas in psychology's replication crisis, I think it's also fair to note that *no* (or close to it) preregistered studies have supported links between violent games and aggression. That's probably as conclusive an answer as we're going to get. The authors could note some of the preregistered studies.

Overall, I thought the design and analysis impressive and well done.

The authors are probably going to need to adopt a Sesoi (smallest effect size of interest). The problem is in big studies a lot of "statistically significant" effects are just non-random methodological noise. Some of those sd units of like .10 are too small to interpret as meaningful. I'd suggest an sd unit closer to .2 as the threshold (see Ferguson & Heene, 2021 for discussion of why this is important).

When the authors say "negative/inconsistent", do you just mean "non-significant"? I think the word "negative" here might be misconstrued.

What exactly is a post-analysis registration? I know preregistration...

In the discussion the authors refer to Etable13 and claim "and exposure to violent video games" is a negative indicator. As they (accurately) indicated in the lit review, there's not good evidence for that...and as they cited me on that table, the study they cite is more nuanced than how they're wording it here. Basically girls (on average) didn't like action games and tended to find them mildly stressful. Of course you could say being forced to do anything you don't care for would be stressful (this was a randomized experiment with teens), and the outcome was mild, temporary stress, not mental health disorders. There is in fact, no good evidence playing violent video games is associated with mental health disorders.

I hope these comments are helpful.

Signed,
Chris Ferguson

Reviewer #2:
Remarks to the Author:
See attachment

Reviewer #3:
Remarks to the Author:
abstract should have less of background information. this background information can be instead included in the introduction section.
abstract should focus more on methodology and results.
include mental health or mental wellbeing in the keywords.
this manuscript is not about the covid-19 pandemic. hence, the introduction should not start with the description of the pandemic. it should be restructured.
introduction consists of multiple short paragraphs. it seems the flow or connection between these short paragraphs are missing. authors are advised to restructure the introduction with appropriate content with in-depth discussion. if needed, subsections can be included.
the hyperparameters of the machine learning algorithms used should be included.
results should be depicted graphically.

Reviewer 1

I appreciated the opportunity to read this study which I think could be a good contribution to the literature pending a few revisions. My comments are below.

Overall, I thought the design and analysis impressive and well done.

Reviewer 1, Point 1:

*The authors do a nice job balancing the literature review. As one comment the authors claim there's no conclusive answer to the video game violence question, and they are quite right to note there is debate of course. However, much like other areas in psychology's replication crisis, I think it's also fair to note that *no* (or close to it) preregistered studies have supported links between violent games and aggression. That's probably as conclusive an answer as we're going to get. The authors could note some of the preregistered studies.*

Thank you for highlighting the importance of preregistered studies in the context of video game violence research. In line with your valuable suggestion, we have now revised the relevant paragraph to emphasize that preregistered studies have not supported a connection between violent games and aggression. Additionally, we have cited notable preregistered studies to strengthen this point.

The revised paragraph:

“There is extensive research on the effects of video games on users, including their impact on aggression, addiction, well-being, and cognitive functioning. Aggression has particularly attracted scholarly interest over the past decades, but no conclusive evidence on the relationship between gaming and aggression exists. Early studies, predominantly from the 2000s, suggested a connection between digital game violence and heightened aggression.^{1,2} However, many subsequent studies, including preregistered ones, have disputed this linkage.³⁻⁵ While the evidence about aggression remains inconclusive, it is noteworthy that the interest among scholars and policymakers has gradually shifted to the link between video gaming and mental well-being over the last decade.⁶” Lines 79-86

Reviewer 1, Point 2:

The authors are probably going to need to adopt a Sesoi (smallest effect size of interest). The problem is in big studies a lot of “statistically significant” effects are just non-random methodological noise. Some of those sd units of like .10 are too small to interpret as meaningful. I'd suggest an sd unit closer to .2 as the threshold (see Ferguson & Heene, 2021 for discussion of why this is important).

Thank you for directing our attention to the work of Ferguson & Heene (2021).⁷ Their insight into the importance of considering the effect size, with reference to the controversy illustrated by Orben and Przybylski (2019),⁸ indeed casts a valuable perspective on interpreting effect sizes. We concur with the reviewer's concerns regarding arbitrary model selection and its implications for effect size interpretation. To illustrate, our initial manuscript underscored, in the supplementary material of our preliminary association analysis, how estimated effect sizes were sensitive to the model selection process.

In accordance with your valuable recommendation, we have incorporated in the Discussion section of our manuscript an additional paragraph. We elucidated our consideration and interpretation of the smallest effect size of interest (SESOI):

“Acknowledging effect size is increasingly crucial as large datasets can unveil statistically significant yet minor effects, leading to potential overinterpretation of ‘crud’ effects—trivial or spurious associations.⁷ In our analysis, the IV method identified effect sizes for video game ownership as follows: 0.60 SD for Switch on mental health, 0.12 SD for PS5 on mental health, and 0.23 SD for PS5 on life satisfaction. Except for the PS5’s impact on mental health, these effect sizes exceed 0.2 SD: the smallest effect size of interest for media effects research proposed by Ferguson (2009).⁹ The effect size for Nintendo Switch ownership is particularly notable, exceeding 0.5 SD—a threshold suggested by Norman et al. (2003) as a perceptible improvement to participants in their medical study.¹⁰ Additionally, the effect sizes for gameplay on Switch and PS5, also estimated by the IV method, range from 0.2 to 0.8 SD. Therefore, we conclude that the estimated positive effects of video game engagement on mental well-being are not only non-negligible but also likely perceptible to participants.” Lines 278-289

We hope this revision addresses the reviewer’s concerns and suggestions, improving the contribution of our study to the ongoing discussions on this topic.

Reviewer 1, Point 3:

When the authors say “negative/inconsistent”, do you just mean “non-significant”? I think the word “negative” here might be misconstrued.

Thank you for pointing out the ambiguity. Our initial description in the Discussion might have been confusing regarding the observed correlation between video gaming and psychological distress.

In our original submission, we wrote:

“Consistent with previous research, our preliminary association analysis indicated a negative/insignificant correlation between video gaming and psychological distress (eTable 6). In contrast, we found a positive correlation between video gaming and life satisfaction, which is somewhat counterintuitive but aligns with recent studies. “

Upon reflection and taking into account both your feedback and that of Reviewer 2 (Point 9), we recognized that our description was unclear. We intended to explain that our association analysis showed a negative relationship between video gaming and psychological health, but it was confusing.

To address this, we have revised the paragraph in the main text to improve clarity:

“Consistent with previous research, our preliminary association analysis showed positive correlations between video gaming and psychological distress; a higher K6 score represents poorer mental health (Supplementary Table 2). Yet, the modest effect sizes found question their practical importance, particularly when we consider a threshold of 0.2 SD units as the smallest effect size of interest. Notably, three out of the five estimates from Model 3 were not statistically significant. Conversely, all our estimates indicated positive correlations between video gaming and life satisfaction—a finding that, while counterintuitive given certain public perceptions around gaming, is consistent with recent studies.^{11,12} However, these analyses do not necessarily elucidate causality. Readers can find further insights on the correlation analysis in Supplementary Method 2.3 and Supplementary Result 1.” Lines 246-255

We also revised the relevant part in the supplementary material:

“First, as anticipated, we observed statistically significant positive correlations between psychological distress level measured by K6 and video game engagement, although not consistently across all estimates. It is important to highlight that a higher K6 score indicates poorer mental health; thus, a positive correlation between K6 and video gaming implies a negative association between mental health and video gaming. However, the observed effect sizes were modest, raising questions about their practical significance. For instance, when adopting a threshold of 0.2 SD units as the smallest effect size of interest,¹¹ the observed effects appear non-meaningful. Moreover, consistent with previous literature,¹² models incorporating a larger set of control variables displayed a smaller negative correlation. Notably, when adjusting for gaming preference in the third model, the beta (β)

coefficient estimates were substantially altered; three out of the five estimates from Model 3 were not statistically significant. The marked variability in estimates across different models suggests the potential unreliability of these estimates.

Second, the analysis revealed a consistent positive correlation between SWLS scores and measures of video game engagement (Supplementary Table 2). While these findings might initially seem counterintuitive given certain public perceptions around gaming, they align well with recent studies,^{13,14} which found positive associations between video game participation and well-being. Nevertheless, it is essential to note that those association analyses may not provide insight into the causal relationship.” Lines 331-348

Reviewer 1, Point 4:

What exactly is a post-analysis registration? I know preregistration...

Thank you for highlighting our reference to post-analysis registration in the manuscript, necessitating clarification. Ideally, study registration occurs before data analysis or even before data collection. However, practical constraints, such as resource limitations, sometimes impede researchers from registering their studies at these initial stages.¹⁵

In situations like ours involving unexpected natural experiments that require prompt data collection, pre-analysis registration is challenging. Such registration is inherently more difficult than registering studies conducted in controlled labs or observational studies using existing data.

Some academic discussions suggest that post-analysis registration can still be beneficial,^{16,17} as illustrated by a paper in *Psychological Science*.¹⁸ Even when pre-registration is impractical after analysis has commenced, registering study protocols, including data collection and handling procedures, can still enhance transparency.¹⁶

However, in light of the reviewer’s insightful comment, we have opted to omit references to our post-analysis registration to preclude any potential confusion among readers. We stand ready to reinstate this information in the manuscript should it be deemed necessary or advisable by the editorial team or reviewers. We appreciate the reviewer’s thoughtful feedback and are open to any additional suggestions you might have to improve our manuscript further.

Reviewer 1, Point 5:

In the discussion the authors refer to Etable13 and claim “and exposure to violent video games” is a negative indicator. As they (accurately) indicated in the lit review, there’s not good evidence for that...and as they cited me on that table, the study they cite is more nuanced than how they’re wording it here. Basically girls (on average) didn’t like action games and tended to find them mildly stressful. Of course you could say being forced to do anything you don’t care for would be stressful (this was a randomized experiment with teens), and the outcome was mild, temporary stress, not mental health disorders. There is in fact, no good evidence playing violent video games is associated with mental health disorders.

Thank you for your thoughtful comments and insights. We particularly value your contributions to the field, and your work has been instrumental in guiding us to focus our study on the relationship between well-being and video gaming.

We apologize for the misrepresentation of your research findings in our initial manuscript. Upon revisiting your study, we fully recognize the nuanced conclusions you have drawn, particularly concerning the gender differences in response to action games and the mild, temporary stress involved—rather than mental health disorders.

To address the reviewer's comment, firstly, we have undertaken two revisions to Supplementary Table 10 (formerly eTable 13). Firstly, we have replaced our previous citation with Hasan et al. (2013),¹⁹ which investigated the potential stressfulness of violent video games through measurements of cardiac coherence levels. While the authors of that study termed their measures as 'stress' rather than mental well-being, we acknowledge that cardiac coherence could be interpreted as an indicator of mental well-being. Secondly, we have adjusted the wording in our descriptions to better capture subtle nuances:

Original:

Violent video games can cause psychological harm by inciting stress.

Revised:

"Violent video games can potentially cause psychological harm by inciting stress, though there has been little observational evidence."

We wholeheartedly agree with the reviewer's point that there is no good evidence to indicate that playing violent video games deteriorates mental health or well-being. Our goal was to present a comprehensive yet neutral overview of the potential mechanisms by which gaming could impact well-being, with a view to appealing to a broad audience. If you recommend removing the row concerning violent video games as a negative indicator, we are fully prepared to make this adjustment.

Secondly, we made a revision to a sentence in Discussion.

Original:

Negative pathways include inadequate sleep, sedentary behavior, and exposure to violent video games.

Revised:

"Potential negative pathways include inadequate sleep, sedentary behavior, exposure to violent video games, and reduction of social interactions." Lines 318-320

Reviewer 2

The paper I was asked to referee (NATHUMBEHAV-23061961) studies the causal effects of video-games on mental health and subjective well-being in a quasi-experimental setting. The setting is as follows: during the COVID-19 pandemic, supply chain disruptions limited the availability of gaming consoles such as the Nintendo Switch and the Playstation 5. Japanese retailers responded to the supply shortages by assigning the consoles through lotteries. The authors have access to survey data containing: i) how many times a survey respondent participated in a lottery for the Nintendo Switch and/or the Playstation 5; ii) whether the respondent won the lottery; iii) various measures of well-being (specifically, the Satisfaction With Life Scale) and mental health (specifically the Kessler Psychological Distress Scale K-6). The authors perform various kinds of analysis, but the key idea is to exploit exogenous variation in lottery winnings as an instrument for Nintendo Switch or Playstation 5 ownership. The authors find that owning a Nintendo Switch or a Playstation 5 has a positive causal effect on mental health and subjective well-being.

Overall, I think that studying the impact of video-games on mental health and subjective well-being is very important given: a) the amount of time many people spend playing video-games, and b) recent research suggesting that other popular digital activities (e.g., social media) have negative effect on well-being and mental health. The internal validity of the study also seems pretty good: the quasi-experimental variation exploited by the authors, namely a lottery, is as close to an ideal experiment as possible (though see the first bullet point in the Specific Comments section for a caveat).

Reviewer 2, Point 1:

My main comment is about external validity (which is often a cheap shot, but in this case it seems quite relevant). Specifically, the quasi-experiment exploited by the authors occurred during the COVID-19 pandemic which was a time of: a) severe mental distress, and b) limited opportunities for social interactions. As a pointer on the severity of mental distress during the pandemic, Twenge and Joiner (2020) found that that mental distress – measured on the same scale as the one employed by the authors – increased on average by around 1.5 standard deviation units between 2018 and 2020 on a representative online US sample. Furthermore, the pandemic years were a time in which social interactions, which tend to have positive effects on subjective well-being, were severely curtailed. Therefore, one of the main mechanisms whereby video-games might have a negative effect on mental health and subjective well-being – namely crowding out offline social interactions – was mechanically attenuated. The fact that the quasi-experiment occurred during the COVID-19 pandemic makes it hard to assess the external validity of the findings. If, for instance, the lottery had happened in 2019, would the results have been the same? I have a hard time believing that, especially in light of the fact that some of the results are very large. For instance, the authors find that the causal effect of owning a Nintendo Switch on mental health is 0.6 standard deviation units. As a point of comparison, a meta-analysis on the effectiveness of cognitive behavioral therapy (CBT) estimates that the effects of CBT (in non-pandemic times) is around 0.8 standard deviation units (Cuijpers et al., 2023). It seems unlikely that, in normal times, the effect on mental health of giving people a Nintendo Switch would be around 3/4 of the effects of psychotherapy.

We sincerely appreciate your insightful feedback on the external validity of our study, especially considering the unique context of the COVID-19 pandemic. We agree with the reviewer's concerns and have made revisions to emphasize the unique circumstances of the pandemic and how they might influence our results. We recognize that our estimates, drawn during this period, could be larger than those from pre-pandemic or post-pandemic times. Concurrently, we have expanded on the potential mechanisms through which gaming might affect mental well-being.

To revise our manuscript, we considered the following three points to examine the reduced external validity due to the COVID-19 pandemic.

(I) Severe Mental Distress: We concur with the reviewer's observation regarding the heightened magnitude of gaming's positive impact on well-being due to severe mental distress. Our quantile regressions support this, revealing that the gaming effect was pronounced among those experiencing severe psychological distress (the Nintendo Switch ownership effect in Supplementary Figure 14; former eFigure 14). The mitigation of pandemic-induced mental health deterioration could indeed be a significant contributor to the estimated large benefit.

(II) Physical Exercise and Gaming: In light of limited opportunities for physical exercise during the pandemic, it is conceivable that there was less displacement of exercise by gaming—listed as a potential negative pathway in Supplementary Table 10. This could have further contributed to the pronounced effect size we estimated.

(III) Offline Social Interactions and Gaming: We are grateful for the reviewer's note on gaming's potential to "crowd out offline social interactions." Reflecting upon this, we realized our oversight in omitting the potential negative impact of reduced offline social interactions due to gaming in our initial list (Supplementary Table 10; formerly eTable 13). This oversight has now been amended. Yet, we have also acknowledged perspectives from researchers who posit that gaming can enhance or maintain offline relationships.^{20–22} As a result, our revised table now presents a balanced view, encompassing both the potential negative and positive impacts of gaming on social interactions. Given these arguments, we would be cautious about explaining the negative impact of games on

offline relationships as a reason why gaming effects were more pronounced during pandemics. This is because whether video gaming has a positive or negative impact on offline social connection seems to be still an empirical question. Therefore, video gaming may not have a crowding-out effect on offline social interaction in a normal time. However, if you recommend including limited opportunities for social interactions as a source of reduced external validity, we are fully prepared to make this adjustment.

Given those, we revised the Abstract, Introduction, and Discussion to address the reviewer's concerns.

Abstract

We explicitly stated that we identified “the causal relationship between video gaming and mental well-being in Japan 2020-2022.”

“Here, we identify the causal relationship between video gaming and mental well-being in Japan (2020-2022), using game console lotteries as a quasi-experiment.” Lines 45-47

We share the concerns raised by the editors (about the title) and Reviewer 3 (Reviewer 3, Point 3) regarding the emphasis on the COVID-19 pandemic context at the beginning of the manuscript. To prevent any impressions that our study examines pandemic-specific trends (we do not have comparisons; thus, it is also difficult to say so), we have adopted a more restrained tone in the Abstract. However, should you advise a more explicit articulation in the Abstract, we stand ready to implement further adjustments.

Introduction

We reconstructed the Introduction to address your concern and Reviewer 3's concern (Reviewer 3, Point 3). Particularly, we emphasized that our estimates were drawn by grounding on the COVID-19 pandemic and the game console lotteries. To do so, we moved the description of game console lotteries from the Methods section to the Introduction.

“To this end, we applied a quasi-experimental study design to the original survey data containing information on video gaming activity and mental well-being indicators for individuals aged 10-69 in Japan, collected at various points between 2020 and 2022 during the COVID-19 pandemic. Supply chain disruptions and surged demands during this time limited the availability of two major gaming consoles: Nintendo Switch (Switch) and PlayStation5 (PS5). To address these shortages, Japanese retailers assigned the gaming consoles through lotteries, inadvertently creating a plausibly random variation in access to video games. Winning a lottery became the primary determinant of whether one could purchase these consoles—details of the lotteries and the two game consoles are provided in Supplementary Method 1. Leveraging this unique circumstance and using original survey data, we drew causal inferences grounded in this pandemic context. Additionally, by applying a causal forest machine learning algorithm to our diverse sample, we investigated the moderating role of socio-demographic factors in the relationship between video gaming and well-being. While this specific intersection remains relatively understudied, there is a growing consensus among digital media scholars about the value of a person-specific approach, viewing it as a pathway toward tailored mental health interventions.²³” Lines 123-137

Discussion

We made two revisions considering this issue. Firstly, we revised our explanation about the potential mechanism of the video gaming effect.

“Previous research has proposed mechanisms for both positive and negative effects of video gaming on mental well-being (summarized in Supplementary Table 10). Our study found that positive effects outweigh negative effects, resulting from both positive and negative pathways. Potential negative pathways include inadequate sleep, sedentary behavior, exposure to violent video games, and reduction of social interactions.²⁴⁻²⁶ For instance, playing video games excessively to the point of insufficient sleep can harm well-being. Positive pathways include psychological therapy games, mood management theory, self-determination theory, and social

connection hypothesis.²⁷⁻³⁰ For example, relaxation games can induce a positive mood and improve well-being, as in psychological therapy.” Lines 316-324

Secondly, we carefully explained the external validity as a major study limitation by writing about “severe mental distress and diminished opportunities for physical activity” during the pandemic.

“This study has several limitations. Foremost among them is the concern regarding the external validity of our findings. While our analysis of gaming behaviors in natural settings enhances external validity relative to laboratory studies, the data was collected during the exceptional circumstances of the COVID-19 pandemic. This period was marked by high levels of mental distress and reduced opportunities for physical activity,^{31,32} which may influence our estimates. For example, our quantile regressions underscored a more pronounced gaming effect among individuals experiencing high distress, as depicted in Supplementary Figure 14. Additionally, with fewer opportunities for physical activity during the pandemic, individuals might have been less likely to substitute gaming for physical activities (if there is no exercise to begin with, gaming cannot reduce exercise time), a potential adverse pathway on well-being associated with increased gaming, as documented in Supplementary Table 10. Given the particular context of the pandemic, our current estimates may be higher compared to those obtained in pre- or post-pandemic settings.” Lines 332-343

We hope these modifications adequately address the reviewer’s concerns and enhance the manuscripts’ clarity in presenting the study’s implications and limitations.

Reviewer 2, Point 2:

I am not sure that simply controlling for the number of times in which a person participated in a lottery is sufficient. Specifically, the ideal variable that one would want to control for is the number of times that an individual participates in the lottery, conditional on not winning it. For the control group, that’s something that the authors observe directly. For the treatment group, it is not something that the authors observe, because individuals who do win the lottery stop participating. But then, controlling for number of times in which the lottery is played compares, for instance, a person who won the lottery on her second try (but who would have gone on participate in the lottery three more time had she not won) to a person who participated in the lottery twice and then gave up. Those are not statistically identical individuals, which implies that the conditional ignorability assumption is not satisfied. Unfortunately, I do not have a solution to the problem from the top of my head but maybe the authors do.

– The propensity score matching approach mitigates my concern a bit, but maybe there’s a way of also addressing it in the standard IV framework. Also, if the propensity score matching approach mitigates my concern, it would be good to make a case for it in the paper.

We appreciate the reviewer for raising this important issue. The reviewer highlighted an essential concern regarding the potential confounder in the causal relationship between lottery success and well-being: the number of non-winning lottery participations, hereafter termed “Game-Console-Purchase-Motivation.” This variable represents an individual’s persistent effort and desire to acquire a game console through continuous lottery participation without winning.

We acknowledge that we cannot directly observe the Game-Console-Purchase-Motivation for lottery winners (treatment group). However, for non-winners (control group), this variable aligns directly with their observed number of lottery participations.

(I) Explaining the context further

To clarify the situation and alleviate the concern, we would like to provide additional context that may diminish the perceived discrepancy between the observed number of lottery participation and Game-

Console-Purchase-Motivation. This deeper understanding is also vital for the interpretation of our subsequent analysis.

Contrary to a potentially simplified view of the process—wherein an individual participates in a lottery, awaits the result, and decides on further participation based on the outcome—the reality is somewhat more complex. Participants have the option to enter multiple lotteries concurrently, often with assistance from family members. For example, an individual intensely desiring a game console may participate in lotteries from various retailers while also involving family members—an aspect accounted for in our analysis (our variable representing the number of lottery participation includes those of family members).

Furthermore, the announcement of lottery results does not follow a consistent timeline, with the waiting period fluctuating between two weeks to several months. Retailers also lack a uniform schedule for launching new lotteries. Consequently, non-winning participants cannot instantly join the next lottery and must wait until new ones are announced. Additionally, participants are free to enter new lotteries without waiting for the results of previous ones. With these considerations in mind, the distinction between the observed and true counts of non-winning participations (Game-Console-Purchase-Motivation) might be less pronounced than initially presumed.

(II) Additional analysis 1: Imputation Approach for Game-Console-Purchase-Motivation

In an effort to address the concerns raised, we engaged in two additional analyses, although we acknowledge the inherent challenges in resolving the issue conclusively. The first approach involved predicting Game-Console-Purchase-Motivation. Leveraging the control group's data, we developed a model to predict the Game-Console-Purchase-Motivation for individuals within the treatment group. This issue closely aligns with the attrition problem, leading us to approach it through imputation. It is worth noting that out of 8,192 respondents, 2,323 were successful in the lottery. Using the data from the remaining 5,869 respondents, we imputed Game-Console-Purchase-Motivation values for the lottery winners, addressing the 28.3% (2,323/8,192) missing values.

The prediction utilized various individual characteristics, including gender, age, employment status, and gaming preferences. Given its ensemble learning capabilities and robustness, we opted for the random forest model, drawing from previous studies.³³ The process unfolded: We initiated a random forest model based on non-winning participants, then predicted Game-Console-Purchase-Motivation for lottery winners. Following the prediction phase, we established a new variable to represent Game-Console-Purchase-Motivation, replacing the original participation counts of lottery winners with the predicted values. This new variable—“Imputed-Console-Purchase-Motivation”—incorporates observed and predicted values.

Comparison between predicted Game-Console-Purchase-Motivation and observed number of times that an individual participates in the lottery.

Histograms presented in Letter Figure 1 compare predicted Game-Console-Purchase-Motivation with actual participation counts among lottery winners. Predicted values tended to be higher than actual participation counts for early-stage lottery winners. The median of predicted Game-Console-Purchase-Motivation (3.45) surpassed the median of actual participation (3.00), while the mean of predicted values (3.96) was lower than observed values (5.45), which can be attributed to the presence of extreme values.

We proceeded with our primary ITT analysis using Imputed-Console-Purchase-Motivation instead of actual participation counts, incorporating both multivariate regression and propensity score matching. Supplementary Figure 6, formatted similarly to Figure 1 in the main text, displays the results for easy comparison. The estimates derived from models using Imputed-Console-Purchase-Motivation as a covariate— ‘alternative model’ — mirrored those from our primary model. This underscores the robustness of our results; our findings remain unchanged even after the introduction of Imputed-Console-Purchase-Motivation as a covariate. While we acknowledge the challenges in controlling for Game-Console-Purchase-Motivation, we believe these results provide useful insights that may address Reviewer 2’s concerns.

Intention-to-Treat estimates using Imputed-Console-Purchase-Motivation instead of the number of times of lottery participation (N=8,192).

Notes: CI, confidence intervals; PSM, Propensity Score Matching. The analysis sample is limited to those who joined game console lotteries. The point estimates and the 95 percent confidence intervals are shown. Regression standard errors are clustered by prefectures. Abadie-Imbens robust standard errors are used for PSM. As for the primary model, Equation (1) is used for the regression estimates (Supplementary Methods). The number of times of lottery participation was replaced by the imputed-console-purchase-motivation for the alternative model. A lower K6 means having less psychological distress, while a higher SWLS means greater life satisfaction. The estimates are standardized by the standard deviations. The random forest model, used for the imputation, was constructed using R’s Random Forest package, configured with 500 trees, terminal node sizes of at least five, and considering $\frac{K}{3}$ covariates at each split, where K represents the total number of predictors; a 10-fold cross-validation segmented the dataset into 10 subsets.

(III) Additional analysis 2: Short-period subsample analysis

Another method we used involves analyzing data from shorter time periods. When the data is collected over a short time, and participants cannot enter new lotteries after losing one, the issues we are concerned about decrease. In this scenario, the observed number of lottery participations is almost the same as the Game-Console-Purchase-Motivation, which helps reduce potential biases from not considering this factor.

However, for this method to work, we need to know both when a respondent entered a lottery and if they won, when that happened. Our data for the Nintendo Switch lottery shows when participants entered but not when they won, and unfortunately, our PlayStation5 data does not have either piece of information.

We utilized available Nintendo Switch data for additional analysis, under the assumption that participants ceased entering lotteries upon winning. The full sample encompasses nine months, from March to November 2020. Within this timeframe, we examined shorter intervals: November alone (one month), October through November (two months), September through November (three months), August through November (four months), July through November (five months), and June through November (six months).

Our primary ITT analysis, which includes multivariate regression and the PSM method, was applied to these shorter-interval subsamples. However, for the subsamples of one month (N=352), two months (N=514), and three months (N=681), the PSM estimates turned out unreliable and thus not displayed. These specific subsamples presented propensity score distributions that were not only irregular but also volatile, markedly differing from the distribution observed in the full sample. This significant discrepancy highlights a concerning lack of consistency and comparability, casting doubt on the reliability of the PSM results obtained from these smaller subsamples.

The results of short-period subsample analysis are shown in Supplementary Figure 7. While using smaller samples can lead to unstable estimates with larger errors, the results from these short timeframes were mostly similar to the full sample's estimates, except in the two-month timeframe. Interestingly, the estimates from the four-month period, which had less than half the number of samples as the full period, were still consistent with the full sample's estimates.

Intention-to-Treat estimates using subsample of short period data.

Notes: CI, confidence intervals; PSM, Propensity Score Matching. The analysis sample is limited to those who joined Nintendo Switch lotteries. The point estimates and the 95 percent confidence intervals are shown. Regression standard errors are clustered by prefectures. Abadie-Imbens robust standard errors are used for PSM. As for the primary model, Equation (1) is used for the regression estimates (Supplementary Methods), while the number of times of lottery participation is restricted to the period of interest. The outcome variable is K6; a lower K6 means having less psychological distress. The estimates are standardized by the standard deviations.

Given these findings, we gently propose that our main results are still valid, and the omission of Game-Console-Purchase-Motivation might not have a significant impact on the results. We hope Reviewer 2 finds this additional analysis reassuring.

(IV) Evaluation of PSM in addressing concerns

We aim to evaluate whether the use of PSM method can ease the concerns raised. The application of PSM may provide some relief to the reviewer's concerns since it does not base comparisons on the count of lottery participations. Instead, the method focuses on comparing individuals with similar propensity scores. This approach avoids making direct comparisons between, for example, an individual who won on the second lottery attempt (who might have continued to participate if not for winning) and another who participated twice and then stopped. This method of comparison may help alleviate concerns regarding inappropriate pairing in the analysis.

However, if there is an assumption that controlling for Game-Console-Purchase-Motivation is essential for maintaining conditional unconfoundedness—expressed as $(Y_{0i}, Y_{1i}) \perp D_i \mid G_i$, where G_i represents Game-Console-Purchase-Motivation, then employing PSM without Game-Console-Purchase-Motivation as a covariate might not be useful. This concern arises because achieving conditional unconfoundedness between treatment and outcomes in PSM requires conditioning on $\pi(G_i)$ —the propensity score.

(V) Causal diagram considerations

We aim to explore whether controlling for Game-Console-Purchase-Motivation (GCPM) is better than controlling for the number of times a respondent has participated in the lottery. We approach this by drawing causal diagrams and considering the backdoor criterion.³⁴

Supplementary Figure 8 showcases three causal diagrams. Initially, before acknowledging Reviewer 2's concerns, we operated under the assumptions of causal diagram (1). Upon reflection, diagrams (2) and (3) emerged as viable alternatives. While diagram (2) necessitates controlling for GCPM, diagram (3) only requires control for the number of lottery participation to satisfy the backdoor criterion. Diagram (2) aligns with the concerns raised by Reviewer 2. However, diagram (3) presents a valid alternative, as GCPM does not directly influence lottery success. Rather, the number of lottery participation increases the probability of winning, making control for GCPM potentially unnecessary for achieving conditional ignorability.

GCPM: Game-Console-Purchase-Motivation	X: Covariates
NTLP: Number of times of lottery participation	

Causal Diagrams representing Intention-to-Treat analysis.

(VI) Reflecting on causal diagrams with multiple lottery rounds

Acknowledging that our data might not perfectly capture the information needed for causal inference in the game console lottery scenario, we appreciate the concerns raised by Reviewer 2. These concerns prompted us to reconsider the potential actual causal diagram, especially considering the multiple rounds or ‘turns’ of lottery participation. For clarification, a ‘turn’ represents a distinct round of lottery participation at a particular retailer.

Supplementary Figure 9 presents a revised causal diagram incorporating two ‘turns’ of participation. While this figure illustrates the concept with two turns, actual participation involved multiple, sometimes overlapping, turns. It is worth noting that Supplementary Figure 8 did not account for these multiple turns. However, as previously outlined, participants could enter lotteries at various retailers at different times. Ideally, data collection (of lottery information, gaming behaviour, and well-being) would occur separately for each ‘turn,’ allowing for comparisons within each distinct turn. Under this approach, the concerns raised by Reviewer 2 would be addressed effectively, as the winning or losing of future lotteries would not influence the observed number of lottery participations per turn.

The causal diagram in Supplementary Figure 9 suggests that controlling for the number of lottery participation might not be appropriate when analyzing the causal relationship with data that aggregates all turns—our ITT analysis. The control variable might inadvertently incorporate information only available post hoc, after lottery outcomes have been determined. As a result, some might suggest not using the number of lottery participations as a control, but rather using GCPM instead. However, as demonstrated in Supplementary Figure 6, the alternative models using Imputed-Console-Purchase-Motivation yielded results similar to our primary models, affirming their robustness.

GCPM: Game-Console-Purchase-Motivation
 NTLP₁: Number of times of lottery participation of the first turn
 NTLP₂: Number of times of lottery participation of the second turn
 X: Covariates

Causal diagram with two ‘turns’ representing Intention-to-Treat analysis.

(VII) Data limitations and conclusions

We acknowledge our dataset’s limitations, particularly regarding the granularity of lottery participation across different rounds. The optimal approach would indeed involve collecting detailed participation data for each lottery round, allowing for more precise comparisons. However, our additional analyses and causal diagram considerations aim to transparently and comprehensively address the reviewer’s concerns, underscoring the robustness of our main findings despite the noted limitations.

(VIII) Revisions in the main text and the supplementary material

Given those, we made several revisions to our main text and the supplementary materials. We listed major revisions below.

Main text

Results

Multivariate regression and PSM

“Additional analyses mitigate concerns over an unadjusted potential confounder: non-winning lottery participations (elaborated in ‘Assessing quasi-experiment validity and conditional unconfoundedness’ Sub-subsection in Methods), thereby reinforcing our primary findings. The imputation method and the short-period subsample analysis yielded results aligning with our primary ITT analysis (Supplementary Figures 6-7). Moreover, examination of causal diagrams supported our assumption of conditional unconfoundedness while highlighting data limitations (Supplementary Figures 8-9), discussed further in the subsequent section. Comprehensive details of these additional analyses are available in Supplementary Method 2.8 and Supplementary Result 5.” Lines 177-184

Discussion

“Data limitations in our study necessitate caution, as they may affect the conditional unconfoundedness assumption—specifically, the concern related to a potential confounder: the number of non-winning lottery participations (Supplementary Figures 8-9, displaying causal diagrams, and further elaborated in Supplementary Method 2.8 and Supplementary Result 5). Ideal data collection would occur separately for each ‘turn’ of the

console lottery, facilitating straightforward comparisons between the treatment and control groups within each distinct lottery turn. This strategy would aid in drawing more rigorous causal inferences. While acknowledging the limitations, we posit that the short-period subsample analysis (Supplementary Figure 7) offers mitigation of the concerns, lending credibility to our findings. Lastly, reliance on self-reported data for gaming behavior and well-being measures presents a limitation, although using a quasi-experimental study design alleviates concerns regarding the self-reporting bias; the random variation caused by the lottery is unlikely to be correlated with the self-reporting bias.” Lines 352-363

Methods

Assessing quasi-experiment validity and conditional unconfoundedness

“We further investigated a potential confounder in the causal relationship between lottery success and well-being: the number of times that a survey respondent participates in the lottery, conditional on not winning it. This variable was not controlled for in our primary analysis due to its unobservable nature. To assess the potential risk of violating the conditional unconfoundedness assumption, three strategies were employed: (i) an imputation approach, (ii) analyses of short-period subsamples, and (iii) the application of causal diagrams for consideration of covariates. Methodological details are provided in Supplementary Method 2.8.” Lines 463-469

Supplementary material

Supplementary Methods

Supplementary Method 1.1: Lotteries for Nintendo Switch and PlayStation5.

We made additional explanations about the process of the lottery, as explained above.

“Contrary to a potentially simplified view of the process—wherein an individual participates in a lottery, awaits the result, and decides on further participation based on the outcome—the reality is somewhat more complex. Participants have the option to enter multiple lotteries concurrently, often with assistance from family members. For example, an individual intensely desiring a game console may participate in lotteries from various retailers while also involving family members—an aspect accounted for in our data (our variable representing the number of lottery participation includes those of family members).

Furthermore, the announcement of lottery results does not follow a consistent timeline, with the waiting period fluctuating between two weeks to several months. Retailers also lack a uniform schedule for launching new lotteries. Consequently, non-winning participants cannot instantly join the next lottery and must wait until new ones are announced. Additionally, participants are free to enter new lotteries without waiting for the results of previous ones.” Lines 22-33

Supplementary Method 2.8: Assessment of conditional unconfoundedness assumption.

“A concern arises regarding the assumption of conditional ignorability; controlling solely for the number of lottery participations may not be sufficient. In the control group, we can directly observe the total number of participations without a win. However, the data for the treatment group is limited after an individual wins, as they stop participating. Imagine an individual who wins on the second attempt but would have continued participating; compare this with another who stops after two unsuccessful attempts. These participants are not statistically identical, casting doubt on the conditional ignorability assumption and necessitating further analysis to understand how significantly this caveat might impact our primary findings.

In this supplementary section, we attempt to alleviate the concern regarding the number of non-winning lottery participations, hereafter termed “Game-Console-Purchase-Motivation.” This variable represents an individual’s persistent effort and desire to acquire a game console through continuous lottery participation without winning. We acknowledge that we cannot directly observe the Game-Console-Purchase-Motivation for lottery winners (treatment group). However, for non-winners (control group), this variable aligns directly with their observed number of lottery participations. We use three strategies: (i) an imputation approach, (ii) analyses of short-period subsamples, and (iii) the application of causal diagrams for consideration of covariates.” Lines 256-271

Method 2.8.1 Additional analysis 1: Imputation Approach for Game-Console-Purchase-Motivation.

“The first approach involved predicting Game-Console-Purchase-Motivation. Leveraging the control group’s data, we developed a model to predict the Game-Console-Purchase-Motivation for individuals within the treatment group. This issue closely aligns with the attrition problem, leading us to approach it through imputation. It is worth noting that out of 8,192 respondents, 2,323 were successful in the lottery. Using the data from the remaining 5,869 respondents, we imputed Game-Console-Purchase-Motivation values for the lottery winners, addressing the 28.3% (2,323/8,192) missing values.

The prediction utilized various individual characteristics, including gender, age, employment status, and gaming preferences. Given its ensemble learning capabilities and robustness, we opted for the random forest model, drawing from previous studies.³³ The process unfolded: We initiated a random forest model based on non-winning participants, then predicted Game-Console-Purchase-Motivation for lottery winners. Following the prediction phase, we established a new variable to represent Game-Console-Purchase-Motivation, replacing the original participation counts of lottery winners with the predicted values. This new variable—“Imputed-Console-Purchase-Motivation”—incorporates observed and predicted values.

Subsequently, using Imputed-Console-Purchase-Motivation instead of the actual participation counts, we developed alternative models. The models mirror our primary Intent-To-Treat (ITT) models—multivariate regression and propensity score matching. This alteration facilitated a comparison between the primary and alternative model results, enabling an examination of how the potential confounder influences the outcomes.”
Lines 275-294

Method 2.8.2 Additional analysis 2: Short-period subsample analysis.

“Another method we used involves analyzing data from shorter time periods. When the data is collected over a short time, and participants cannot enter new lotteries after losing one, the issues we are concerned about decrease. In this scenario, the observed number of lottery participations is almost the same as the Game-Console-Purchase-Motivation, which helps reduce potential biases from not considering this factor.

However, for this method to work, we need to know both when a respondent entered a lottery and if they won, when that happened. Our data for the Nintendo Switch lottery shows when participants entered but not when they won, and unfortunately, our PlayStation5 data does not have either piece of information.

We utilized available Nintendo Switch data for additional analysis, under the assumption that participants ceased entering lotteries upon winning. The full sample encompasses nine months, from March to November 2020. Within this timeframe, we examined shorter intervals: November alone (one month), October through November (two months), September through November (three months), August through November (four months), July through November (five months), and June through November (six months).

Our primary ITT analysis, which includes multivariate regression and the PSM method, was applied to these shorter interval subsamples. However, for the subsamples of one month (N=352), two months (N=514), and three months (N=681), the PSM estimates turned out unreliable and thus not displayed. These specific subsamples presented propensity score distributions that were not only irregular but also volatile, markedly differing from the distribution observed in the full sample. This significant discrepancy highlights a concerning lack of consistency and comparability, casting doubt on the reliability of the PSM results obtained from these smaller subsamples.” Lines 296-317

Supplementary Results

Result 5: Assessment of conditional unconfoundedness assumption for ITT analysis.

Result 5.1. Additional analysis 1: Imputation Approach for Game-Console-Purchase-Motivation.

“We proceeded with our primary ITT analysis using Imputed-Console-Purchase-Motivation instead of actual participation counts, incorporating both multivariate regression and propensity score matching. Supplementary Figure 6, formatted similarly to Figure 1 in the main text, displays the results for easy comparison. Findings derived from models using Imputed-Console-Purchase-Motivation as a covariate—‘alternative model’—closely mirrored those presented in Figure 1, underscoring the robustness of our results even after the introduction of Imputed-Console-Purchase-Motivation as a covariate.” Lines 389-395

Result 5.2. Additional analysis 2: Short-period subsample analysis.

“While using smaller samples can lead to unstable estimates with larger errors, the results from these short timeframes (Supplementary Figure 7) were mostly similar to the full sample’s estimates, except in the two-month timeframe. Interestingly, the estimates from the four-month period, which had less than half the number of samples as the full period, were still consistent with the full sample’s estimates. Given these findings, we gently propose that our main results are still valid, and the omission of Game-Console-Purchase-Motivation might not have a significant impact on the results.” Lines 397-403

Result 5.3. Causal diagram considerations.

“In this subsection, we evaluate whether it is more appropriate to control for Game-Console-Purchase-Motivation (GCPM) or the actual number of lottery participation in our study. This assessment is conducted using causal diagrams, with the backdoor criterion guiding the analysis.

Supplementary Figure 8 presents three distinct causal diagrams for consideration. The first diagram (1) outlines a typical study design that employs lotteries as a natural experiment, devoid of considerations related to GCPM. Upon reassessment, two alternative diagrams, labeled as (2) and (3), surface as plausible alternatives to the initial design.

Diagram (2) underscores the need to control for GCPM, as it is a crucial variable influencing the study outcome. In contrast, Diagram (3) indicates that simply controlling for the number of times a respondent participates in the lottery is adequate to satisfy the backdoor criterion. This approach is grounded on the observation that GCPM does not directly influence lottery success. Rather, a higher number of lottery participations increases the chances of winning. Hence, for the purpose of achieving conditional ignorability, it may not be necessary to control for GCPM.” Lines 405-417

Result 5.4. Reflecting on causal diagrams with multiple lottery rounds.

“The concerns related to GCPM as a potential confounder prompt us to reconsider the causal diagram, especially considering the multiple rounds or “turns” of lottery participation. For clarification, a “turn” represents a distinct round of lottery participation at a particular retailer.

Supplementary Figure 9 presents a revised causal diagram incorporating two “turns” of participation. While this figure illustrates the concept with two turns, actual participation involved multiple, sometimes overlapping, turns. It is worth noting that Supplementary Figure 8 did not account for these multiple turns.

As previously outlined, participants could enter lotteries at various retailers at different times. Ideally, data collection (of lottery information, gaming behaviour, and well-being) would occur separately for each “turn,” allowing for comparisons within each distinct turn. Under this ideal approach, the concerns related to GCPM as a potential confounder would be addressed effectively, as the winning or losing of future lotteries would not influence the observed number of lottery participations per turn.

Additionally, the causal diagram in Supplementary Figure 9 suggests that controlling for the number of lottery participation might not be appropriate when analyzing the causal relationship with data that aggregates all turns—our ITT analysis. The control variable might inadvertently incorporate information only available post hoc, after lottery outcomes have been determined. As a result, some might suggest not using the number of lottery participations as a control, but rather using GCPM instead. However, as demonstrated in Supplementary Figure 6, the alternative models using Imputed-Console-Purchase-Motivation yielded results similar to our primary models, affirming their robustness.” Lines 419-439

We hope our clarifications and additional analyses alleviate the reviewer’s concerns regarding the control variables. We agree that further research with more detailed data collection is ideal, and we believe our current approach, limitations acknowledged, provides valuable insights into the causal relationship in question.

Reviewer 2, Point 3:

I am not sure I agree with the authors' terminology in terms of what they are estimating. Suppose the question of interest is: what is the effect of owning a Playstation 5 on mental health? Then, the IV results would measure the local average treatment effect (LATE) on individuals who are induced to own a Playstation 5 as a result of winning the lottery. The reduced form would measure the intent-to-treat (ITT) effect. To the best of my knowledge, the authors cannot estimate the average treatment effect (ATE) using an IV strategy.

We sincerely appreciate your insightful feedback regarding our terminology. Upon reflection, we concur with the reviewer's observation about the appropriateness of the terms "intent-to-treat" (ITT) effect and "local average treatment effect" (LATE) in our context.

To address this, we have made necessary revisions to adopt the recommended terminology in our manuscript. Specifically, we made revisions in Table 2 (formerly Table 1) and explicitly mentioned that the IV method estimates the causal effect among compliers. Further, "sensitivity checks with alternative model selections" section in the supplementary material has been revised to ensure the accurate use of these terms. We have attached the modified parts for the reviewer's convenience.

Revised Table 2 (formerly Table 1):

Overview of statistical analysis

Methods	Estimand	Exposure/ Treatment (assignment) variables	Excluded instrument	Advantages/purposes	Details/Equations
Multivariate regression				Simple and easy to understand, thus suitable to be a baseline specification. This method can be extended to the instrumental variable approach.	Following Imbens et al. (2001) we used Equation 1 in Supplementary Methods.
Propensity score matching method	ITT effect	Winning game console lottery.		While the linear regression approach relies on assumptions of linearity and extrapolations beyond observed variable combinations to correct for differences between the treatment and control groups, the PSM approach does not rely on such parametric assumptions.	Following Imbens (2015), we used the algorithm to select matching variables and also constructed a subsample of the original data set. We conducted covariate balance checks and examined the common support of the propensity score distribution. Further details and the PSM design choice are explained in Supplementary Methods.
Instrumental variable method	LATE	Possession of Nintendo Switch/PS5. Played Nintendo Switch/PS5 last month. Video gameplay time.		This method can estimate the causal effect of standard exposure variables (for example, owning a Nintendo Switch/PS5) among compliers. Additionally, this method was utilized for a subgroup analysis investigating whether the impact of video gaming varies based on the time spent.	The first-stage regression results are found in Supplementary Table 9. Weak instrument tests were conducted to assess the relevance. Further methodological details are available in Supplementary Methods.
Machine learning (instrumental variable causal forest)	CATE	Possession of Nintendo Switch/PS5.	Winning game console lottery.	This method predicts the treatment effects for each individual, considering their specific characteristics. The primary aim was to investigate effect modification (or moderation). Additionally, the correlation between CATEs on well-being and video game duration was analyzed to better understand the underlying mechanism of the impact of video gaming.	Further methodological details are available in Supplementary Methods.

Methodological particulars of the statistical analyses are displayed. The causal inference and machine learning analysis targeted only those who participated in the video game console lottery (N = 1,773 for Nintendo Switch, 6,419 for PS5). ITT, Intention-to-Treat; LATE, Local average treatment effect; CATE, Conditional average treatment effect; PSM, Propensity score matching; PS5, PlayStation5.

The revised section of “Sensitivity checks with alternative model selections” in the supplementary material:

“In Supplementary Figure 10, we presented several estimates with alternative model selections of regression and Propensity Score Matching (PSM). Initially, we demonstrated the estimates of regression without any adjustments. The estimates were observed to be very similar to those with adjustment, which is our principal model. This supports the idea that there was no confounding of the treatment effect.

Next, we exhibited estimates using several alternative design choices of PSM, commonly used for estimating either Average Treatment Effect on Treated (ATT) or Average Treatment Effect (ATE). Some might advocate for estimating the ATT—or, more precisely, in our study, Intention-to-Treat (ITT) effect on the assigned treatment group—using PSM without replacement. Nonetheless, our primary focus was the ATE (or the ITT effect) due to its consistency with our regression that also aimed to estimate the ATE (or the ITT effect). As a sensitivity check, we estimated ATT and found that the estimates are comparable to ATE. We further assessed the ATE using an alternate PSM design, one-to-one matching with replacement. The estimates were in line with prior results, though, as anticipated, the standard errors increased.” Lines 442-455

Reviewer 2, Point 4:

Besides allowing people to play video-games, what else do the Nintendo Switch and Playstation 5 allow people to do? How can we be sure the effects go through video-gaming rather than other activities on those consoles (the IV estimates on game times help a bit there, but game time is likely to be measured quite noisily).

Thank you for highlighting additional functionalities of the Nintendo Switch and PlayStation5 beyond gaming. Indeed, these consoles allow users to engage in other activities like watching videos on platforms such as YouTube. Additionally, PlayStation5 (excluding the digital edition) supports Blu-ray playback. In response to your insightful suggestion, we have updated our supplementary material to acknowledge this.

The revised text in Supplementary Information:

“Additionally, Nintendo Switch and PlayStation5 offer functionalities such as online video streaming (exemplified by platforms like YouTube) and Blu-ray playback (exclusive to PlayStation5), though these are not presumed to be the primary utilities sought by users of these gaming consoles.” Lines 162-165

It is reasonable to assume that the main use of these consoles is for gaming, rather than for watching videos or Blu-rays. While we acknowledge the lack of data to fully support this assertion, it is important to consider that watching videos or Blu-rays can be done on various other devices, which are accessible without entering lotteries. Thus, we find it unlikely that the effects on console ownership identified in our study are driven primarily by these non-gaming activities.

We deeply value your insights and feedback, which enhance the robustness and depth of our work. The reviewer’s point on non-gaming activities on the consoles has provided a pivotal perspective that enriches our discussion, and we remain open to further dialogues to refine our study further.

Reviewer 2, Point 5:

The paper might benefit from a bit of additional editing. E.g., page 3, line 68 says “Second, experimental studies have not been tested the external validity and exposed to criticisms” which does not strike me as grammatically correct.

Thank you for drawing our attention to the areas that need further refinement. In response to the reviewer’s feedback, we consulted with an academic writing expert to improve the manuscript. Efforts have been made to rectify grammatical errors and enhance the overall coherence of the paper, particularly in the part you highlighted and throughout the document.

Here are some examples of the revisions:

#1

Original:

Title: Causal effect of video game play on mental well-being in Japan during 2020-2022

Revised:

“Title: Causal effect of video game play on mental well-being in Japan 2020-2022”

#2

Original:

As digital devices and the Internet become integral parts of daily life, concerns about their potential negative impact on human well-being, especially prolonged screen time, have become more pronounced.

Revised:

“As digital devices and the Internet become integral parts of daily life, concerns about their potential negative impact on human well-being, especially that of prolonged screen time, have become more pronounced.” Lines 61-63

#3

Original:

Second, experimental studies have not been tested the external validity and exposed to criticisms.

Revised:

“Second, many experimental studies lack tests for external validity and have faced criticisms.” Lines 102-103

We believe that these edits, among others, have significantly improved the quality of our manuscript. Your feedback has been invaluable in this refining process, and we sincerely appreciate your attention to detail.

Reviewer 2, Point 6:

Since the survey targets gamers and the main sample involves individuals who want to purchase a Nintendo Switch or a Playstation 5, it seems reasonable that survey respondents already have access to other gaming platforms. Does the survey ask about that? Whether it does or does not, the effects that the authors measure are likely to be the marginal effects of updating one’s gaming console (e.g., going from some other Nintendo to the Nintendo Switch). This makes it all the more surprising that the effects on the Nintendo Switch are so large.

Thank you for your insightful comment regarding the availability of other gaming platforms to lottery participants, the potential marginal effects of upgrading gaming consoles, and how these relate to the large effect observed with Nintendo Switch ownership. The reviewer’s feedback brings up important points about what our study is measuring.

Initially, the reviewer suggests that participants in the game console lottery likely have older consoles and are actively using them. The reviewer also highlights concerns about the transition from older consoles and software to newer versions, expecting only marginal effects to occur in this process. The reviewer probably assumes these changes might be less impactful than when a participant shifts from a different hobby (or any other activities) to video gaming, leading to questions about the large Nintendo Switch effect we estimated.

In this response, we would like to investigate whether the large Nintendo Switch effects should be interpreted as marginal effects. Firstly, it is necessary to elucidate the ownership status among our sample. The assumption the reviewer provided about participants' access to other gaming platforms does not entirely align with our collected data. In Letter Table 1, Information from the Nintendo Switch Lottery shows that 34.18% of participants owned a Nintendo 3DS (the model prior to the Nintendo Switch, a foldable handheld game console), while 25.49% had a PlayStation4. This data implies that most did not have other (old) gaming consoles.

Video game ownership among Nintendo Switch lottery participants.

	Mean	SD	N
Have Nintendo 3DS (=1)	0.3418	0.4744	1773
Have PlayStation4 (=1)	0.2549	0.4359	1773

Notes: The table provides information from Round one survey.

To provide further clarity, we describe the gaming habits of those who did not win the lottery. This would give us an idea about the habits of lottery participants before they won. Letter Table 2 reveals that only 12.87% (approximately a third of owners of Nintendo 3DS) of lottery non-winners were actively playing Nintendo 3DS. There was slightly higher active engagement for PlayStation4 at 15.82%. Thus, the majority of non-winning participants were not actively using older gaming consoles.

Video game ownership and video game engagement among Nintendo Switch lottery non-winners.

	Mean	SD	N
Have Nintendo 3DS (=1)	0.3483	0.4767	847
Played Nintendo 3DS this month (=1)	0.1287	0.3351	847
Have PlayStation4 (=1)	0.2574	0.4374	847
Played PlayStation4 this month (=1)	0.1582	0.3651	847

Notes: The table provides information from Round one survey, about the game consoles owned by respondents who entered but did not win the Nintendo Switch Console lottery.

Given those, it is possible that the changes we saw after people won the Nintendo Switch lottery are more than just minor updates or marginal effects typically seen with console upgrades. For a more comprehensive understanding, regression analysis was employed. A variable was introduced to indicate ownership of any current gaming consoles (including Nintendo Switch, Nintendo 3DS, PlayStation 5, PlayStation 4, PlayStation 3, and Xbox series). Another variable was created to represent active use of these consoles in the month when the survey was conducted. These variables were designed to examine the impact of winning the Nintendo Switch lottery on console ownership and gameplay. The transition from non-ownership to ownership of gaming consoles, and from non-use to use due to winning the lottery, likely represents more than just minor effects.

Letter Table 3 offers additional insights. The estimates in row 1 and row 2 suggest a 30.5 percentage point increase in Nintendo Switch ownership and a 22.6 percentage point increase in playing Nintendo Switch within the survey month.

Nintendo Switch lottery success, video game console ownership, and video game engagement.

Explanatory variables	Number of observations	β coefficient (95% CI)	Mean of non-winners
Outcome variable: Have Nintendo Switch			
(1) Win Nintendo Switch lottery	1,773	0.305 (0.254 to 0.357)***	0.451
Outcome variable: Played Nintendo Switch this month			
(2) Win Nintendo Switch lottery	1,773	0.226 (0.175 to 0.276)***	0.342
Outcome variable: Have video game consoles			
(3) Win Nintendo Switch lottery	1,773	0.158 (0.119 to 0.196)***	0.667
Outcome variable: Played video game consoles this month			
(4) Win Nintendo Switch lottery	1,773	0.141 (0.093 to 0.189)***	0.492

Notes. CI, confidence intervals. The analysis sample was limited to those who joined the Nintendo Switch lottery in Round one. Equation (1) of the supplementary material, which controls for a comprehensive set of covariates, was used for the regression estimates. Standard errors were clustered by prefectures. *** $p < 0.01$, ** $p < 0.05$, * $p < 0.1$.

Moreover, rows 3 and 4 in the table indicate that winning the lottery increased overall console ownership by 15.8 percentage points and the likelihood of playing on any console in the surveyed month by 14.1 percentage points. If the lottery-win only led to marginal effects in access to consoles, we would not expect to see these increases. This suggests that the significant effects documented in our study are not mostly due to the marginal impacts associated with upgrading gaming consoles.

Lastly, the term ‘updating a gaming console’ may inadvertently simplify the process, likening it to upgrading from one smartphone model to another, or switching between similar laptop brands. However, acquiring a new gaming console may represent more than a mere upgrade. It could provide an opportunity for individuals to start gaming again after some time away, a scenario that is both anticipated and valued by gaming companies.³⁵ From another perspective, it offers users the experience of engaging with new gaming software and technology.

These considerations are important for understanding the strong effects we observed with Nintendo Switch ownership in our study. We hope this explanation addresses the reviewer’s concerns, and we appreciate the opportunity to clarify these points for Reviewer 2.

Reviewer 2, Point 7:

It would be good to present results with up to two decimal points in the main text.

We appreciate the reviewer’s suggestion and have revised the main text to present results with up to two decimal points as recommended.

Reviewer 2, Point 8:

From what I understand, the structure of the data is a panel rather than a repeated cross-section (though neither term is used in the paper). It would be good to make the structure of the data clear early on.

Thank you for this insightful suggestion. We have now explained the data structure in the subsection of Participants and data collection in the Method section.

Participants and data collection:

“The study participants included people aged 10-69 years living in Japan. We partnered with a gaming market research firm, gameage R&I (GRI), to conduct five rounds of omnibus online surveys from December 2020 to March 2022 among the 10-65 age group (n = 97,602) across all 47 prefectures in Japan. A stratified random sampling technique—stratified by age, gender, and video gaming preferences—was used to recruit participants from the pool of approximately 150,000 respondents from a survey agency. The response rate was 59.3% (details in Supplementary Table 11). We collected information on participants’ lottery participation, video game ownership, gaming preferences, mental health, life satisfaction, and sociodemographic characteristics. The dataset is a blend of panel and repeated cross-sectional observations, with 35.90% of respondents participating multiple times. This data structure facilitates robustness checks, including placebo analysis.” Lines 386-395

Reviewer 2, Point 9:

On page 13 of the appendix, line 310, the authors refer to table e6 and state: “there were negative associations between psychological distress measured by K6 and video game engagement.” In table e6, however, the correlation between between psychological distress and having a Playstation 5 is positive. Maybe the authors meant to say that there was a negative association between psychological well-being and video game engagement?

Thank you for highlighting the ambiguity. Given that a similar concern was raised by Reviewer 1 (Point 3), we realize that our explanation was unclear. We aimed to communicate that while there was a positive correlation between K6 scores and video gaming, this indicates a negative association with mental health, given that higher K6 scores represent poorer mental health.

Considering that Reviewer 1 (Point 3) raised a similar issue, we recognized that our description was ambiguous. We intended to explain that our association analysis showed a negative relationship between video gaming and psychological health, but it was confusing. To ensure clarity, we have made necessary revisions in both the main manuscript (see Reviewer 1, Point 3) and the supplementary material.

The revised part of the supplementary material:

“First, as anticipated, we observed statistically significant positive correlations between psychological distress level measured by K6 and video game engagement, although not consistently across all estimates. It is important to highlight that a higher K6 score indicates poorer mental health; thus, a positive correlation between K6 and video gaming implies a negative association between mental health and video gaming. However, the observed effect sizes were modest, raising questions about their practical significance. For instance, when adopting a threshold of 0.2 SD units as the smallest effect size of interest,¹³ the observed effects appear non-meaningful. Moreover, consistent with previous literature,¹⁴ models incorporating a larger set of control variables displayed a smaller negative correlation. Notably, when adjusting for gaming preference in the third model, the beta (β) coefficient estimates were substantially altered; three out of the five estimates from Model 3 were not statistically significant. The marked variability in estimates across different models suggests the potential unreliability of these estimates.

Second, the analysis revealed a consistent positive correlation between SWLS scores and measures of video game engagement (Supplementary Table 2). While these findings might initially seem counterintuitive given certain public perceptions around gaming, they align well with recent studies,^{11,12} which found positive associations between video game participation and well-being. Nevertheless, it is essential to note that those association analyses may not provide insight into the causal relationship.” Lines 331-348

We hope that these changes effectively address the issues raised.

Reviewer 2, Point 10:

The first stage for the Nintendo Switch in table e14 is pretty weak: if I understand the table correctly, only about a third of individuals who win the lottery for the Nintendo Switch end up buying it. Any idea why that might be?

Thank you for raising this insightful question regarding the weak first stage in the Nintendo Switch analysis. We appreciate your meticulous review and would like to provide further clarification on this matter.

The first stage’s relative weakness in the Nintendo Switch analysis is predominantly due to the high take-up rate amongst lottery non-winners. Letter Table 4 illustrates that while a minor 6.55% of PlayStation5 lottery non-winners own a PlayStation5, a substantial 45.10% of non-winners in the Nintendo Switch lottery do have the console. This discrepancy in ownership rates significantly influences our first stage.

It is crucial to note that about 75% of the lottery winners for both consoles do possess their respective devices. Thus, the observed weak first stage is not resulting from winners opting not to purchase the consoles. This pattern suggests the presence of a notable number of ‘always takers’, assuming there are no defiers in the sample.

Take-up among lottery participants.

(a) Nintendo Switch Lottery

	Have Nintendo Switch	
	Yes	No
Winner	716 77.32%	210 22.68%
Non-winner	382 45.10%	465 54.90%
Total	1,098 61.93%	675 38.07%

(b) PlayStation5 Lottery

	Have PlayStation5	
	Yes	No
Winner	1,012 72.44%	385 27.56%
Non-winner	329 6.55%	4,693 93.45%
Total	1,341 20.89%	5,078 79.11%

Notes: The table provides information from Round one survey, about the game consoles owned by respondents who entered but did not win the Nintendo Switch Console lottery.

Several reasons can account for the elevated ownership among Nintendo Switch lottery non-winners:

1. Second-Hand Market Accessibility:

With its release in 2017, Nintendo Switch has been accessible far longer than PlayStation5. Its availability in the second-hand market is hence plausible, offering non-winners an opportunity to acquire the console outside the lottery, albeit at extra costs.

2. Multiple Console Ownership within Households:

Nintendo Switch offers flexibility by providing both TV-connected and handheld gaming options. This feature makes it appealing for households to own more than one unit to accommodate various members. According to Nintendo, around 20% of their sales in the fiscal year 2020 were generated by customers purchasing an additional Switch.³⁶ While we do not have data on prior ownership for our

lottery participants, the general market trend allows us to reasonably expect similar behavior within our sample.

We hope this explanation resolves the reviewer's concerns.

Reviewer 3

Reviewer 3, Point 1:

Remarks to the Author: abstract should have less of background information. this background information can be instead included in the introduction section. abstract should focus more on methodology and results.

We thank you for your constructive feedback. In response, we have restructured the Abstract to trim down the background information and give more emphasis to our methodology and results, following the Nature abstract format. Below is our revised abstract:

“The widespread use of video games has raised concerns about their potential negative impact on mental well-being. Nevertheless, the empirical evidence supporting this notion is largely based on association studies, warranting further investigation. Here, we identify the causal relationship between video gaming and mental well-being in Japan (2020-2022), using game console lotteries as a quasi-experiment. Applying causal inference methods to survey data (n=97,602), we found that game console ownership (Nintendo Switch and PlayStation5), along with increased gameplay, improved mental well-being. The console ownership reduced psychological distress and improved life satisfaction by 0.1-0.6 standard deviations. Further, a causal forest machine learning algorithm revealed divergent impacts between the two consoles, with PlayStation5 showing lower benefits for females and adolescents, while no similar pattern emerged for Nintendo Switch. These findings highlight the complex impact of digital media on mental well-being and underscore the need for policy design that considers differential screen time effects.” Lines 42-53

Reviewer 3, Point 2:

include mental health or mental wellbeing in the keywords.

Thank you for the suggestion. We have now added ‘mental well-being’ to the keywords, as recommended.

Reviewer 3, Point 3:

this manuscript is not about the covid-19 pandemic. hence, the introduction should not start with the description of the pandemic. it should be restructured. introduction consists of multiple short paragraphs. it seems the flow or connection between these short paragraphs are missing. authors are advised to restructure the introduction with appropriate content with in-depth discussion. if needed, subsections can be included.

We value and appreciate your insightful comments, which have been crucial in refining the introduction section of our manuscript. In response, we have revised the introduction to enhance its structure, flow, clarity, and depth.

Firstly, we have begun our introduction by discussing the broader debates on the potential impacts of digital technologies on human well-being, focusing on video games. This initial context emphasizes that the primary focus of our research is not the COVID-19 pandemic but the broader questions surrounding the effects of video gaming.

Next, we delved deeper into the literature, discussing both the aggression (with consideration of the comment from Reviewer 1, Point 1) and mental well-being angles. We emphasized the methodological challenges that previous research has faced.

Recognizing the need for clarity in our unique methodological approach, we elaborated on how our quasi-experimental design, set within the unique circumstances of the pandemic (with consideration of the comments from Editors, Point 2 and Reviewer 2, Point 1), provides new findings on the causal relationship between video gaming and mental well-being.

Lastly, regarding the suggestion of using subsections, it's worth noting that subsections are not permitted in the formatting guidelines of Nature Human Behaviour. However, we have made efforts to ensure that the revised introduction has a clear and logical flow. We trust that the modifications align with the reviewer's feedback and enhance the manuscript's presentation.

Reviewer 3, Point 4:

the hyperparameters of the machine learning algorithms used should be included.

Thank you for your insightful comment. In response, we have expanded the details on the parameters of the causal forest algorithm used in our study in accordance with relevant empirical literature.³⁷⁻⁴⁰

The updated paragraph in the supplementary material is as follows:

“The instrumental variable causal forest was implemented using the "grf" package in R. Predicted conditional treatment effects were generated with selected parameters as follows: (i) the minimum leaf size was set to five (the default of the package), (ii) the number of trees in the forest was set to 10,000 (larger than the default: 2,000), (iii) the fraction of bootstrapped subsample used to build each tree was set to 50% (the default of the package), (iv) the number of covariates considered at each split was set to $\frac{K}{3}$, with K being the total number of predictors, following the empirical literature.^{37,40} Additionally, the algorithm was run for 1,000 simulations on different test samples. This was done to ensure that the results were robust and not dependent on a particular sample. The list of covariates can be found in Supplementary Table 15.” Lines 235-243

We trust this clarification addresses the reviewer's concerns adequately. We are open to and ready for any further recommendations or suggestions you might have.

Reviewer 3, Point 5:

results should be depicted graphically.

Thank you for your valuable feedback. In response to your suggestion for an enhanced graphical representation of results, we have relocated an important figure—“Machine learning results of effect modification contrasting ownership of Nintendo Switch and PlayStation5”—from the supplementary material (formerly eFigure 11) to the main text as Figure 4. This relocation facilitates easy reference, given its recurrent mention in the text, and enhances accessibility and comprehension for readers. Furthermore, we have graphically presented the results of additional analyses—conducted to address feedback from Reviewer 2—in Supplementary Figures 6-9.

We trust that the adjustments made align well with the reviewer's recommendations, further refining the manuscript's clarity and accessibility. We remain open to any further guidance or suggestions you might have to improve the presentation of our results.

Summary

We thank the editors and the reviewers for the time to consider our manuscript. We are convinced the work has improved substantially, and we hope you agree that it will make an important contribution to the literature around this topic.

Yours Sincerely,

Hiroyuki Egami, PhD
Assistant Professor
Research Institute of Economic Science, Nihon University, Tokyo, Japan

References

1. Prescott, A. T., Sargent, J. D. & Hull, J. G. Metaanalysis of the relationship between violent video game play and physical aggression over time. *Proc. Natl. Acad. Sci.* **115**, 9882–9888 (2018).
2. Anderson, C. A. & Dill, K. E. Video games and aggressive thoughts, feelings, and behavior in the laboratory and in life. *J. Pers. Soc. Psychol.* **78**, 772–790 (2000).
3. Drummond, A., Sauer, J. D. & Ferguson, C. J. Do longitudinal studies support long-term relationships between aggressive game play and youth aggressive behaviour? A meta-analytic examination. *R. Soc. Open Sci.* **7**, (2020).
4. Przybylski, A. K. & Weinstein, N. Violent video game engagement is not associated with adolescents' aggressive behaviour: Evidence from a registered report. *R. Soc. Open Sci.* **6**, (2019).
5. Elson, M. & Ferguson, C. J. Twenty-five years of research on violence in digital games and aggression: Empirical evidence, perspectives, and a debate gone astray. *Eur. Psychol.* **19**, 33–46 (2014).
6. Turner, N. E. *et al.* Prevalence of Problematic Video Gaming among Ontario Adolescents. *Int. J. Ment. Heal. Addict.* **10**, 877–889 (2012).
7. Ferguson, C. J. & Heene, M. Providing a Lower-Bound Estimate for Psychology's "Crud Factor": The Case of Aggression. *Prof. Psychol. Res. Pract.* **52**, 620–626 (2021).
8. Orben, A. & Przybylski, A. K. The association between adolescent well-being and digital technology use. *Nat. Hum. Behav.* **3**, 173–182 (2019).
9. Ferguson, C. J. An Effect Size Primer: A Guide for Clinicians and Researchers. *Prof. Psychol. Res. Pract.* **40**, 532–538 (2009).
10. Norman, G. R., Sloan, J. A. & Wyrwich, K. W. Interpretation of Changes in Health-related Quality of Life: The Remarkable Universality of Half a Standard Deviation. *Med. Care* **41**, 582–592 (2003).
11. Johannes, N., Vuorre, M. & Przybylski, A. K. Video game play is positively correlated with well-being. *R. Soc. Open Sci.* **8**, (2021).
12. Kelly, S., Magor, T. & Wright, A. The Pros and Cons of Online Competitive Gaming: An Evidence-Based Approach to Assessing Young Players' Well-Being. *Front. Psychol.* **12**, 1–9

- (2021).
13. Ferguson, C. J. Is Psychological Research Really as Good as Medical Research ? Effect Size Comparisons Between Psychology and Medicine. *Rev. Gen. Psychol.* **13**, 130–136 (2009).
 14. Ferguson, C. J. Do Angry Birds Make for Angry Children? A Meta-Analysis of Video Game Influences on Children’s and Adolescents’ Aggression, Mental Health, Prosocial Behavior, and Academic Performance. *Perspect. Psychol. Sci.* **10**, 646–666 (2015).
 15. Humphreys, M., Sanchez de la sierra, R. & Van der windt, P. Fishing, Commitment, and Communication: A Proposal for Comprehensive Nonbinding Research Registration. *Polit. Anal.* **21**, 1–20 (2013).
 16. Tackett, J. L., Brandes, C. M. & Reardon, K. W. Leveraging the open science framework in clinical psychological assessment research. *Psychol. Assess.* **31**, 1386–1394 (2019).
 17. World Bank. Study Registration: When should I register? DIME Wiki. https://dimewiki.worldbank.org/Study_Registration#When_should_I_register? (2023).
 18. Brandes, C. M., Herzhoff, K., Smack, A. J. & Tackett, J. L. The p Factor and the n Factor: Associations Between the General Factors of Psychopathology and Neuroticism in Children. *Clin. Psychol. Sci.* **7**, 1266–1284 (2019).
 19. Hasan, Y., Laurent, B. & Bushman, B. J. Violent Video Games Stress People Out and Make Them More Aggressive. *Aggress. Behav.* **39**, 64–70 (2013).
 20. Molyneux, L., Vasudevan, K. & Gil de Zúñiga, H. Gaming Social Capital: Exploring Civic Value in Multiplayer Video Games. *J. Comput. Commun.* **20**, 381–399 (2015).
 21. Trepte, S., Reinecke, L. & Juechems, K. The social side of gaming : How playing online computer games creates online and offline social support. *Comput. Human Behav.* **28**, 832–839 (2012).
 22. Domahidi, E., Festl, R. & Quandt, T. To dwell among gamers : Investigating the relationship between social online game use and gaming-related friendships. *Comput. Human Behav.* **35**, 107–115 (2014).
 23. Valkenburg, P. M., Meier, A. & Beyens, I. Social media use and its impact on adolescent mental health: An umbrella review of the evidence. *Curr. Opin. Psychol.* **44**, 58–68 (2022).
 24. Peracchia, S. & Curcio, G. Exposure to video games: Effects on sleep and on post-sleep cognitive abilities. A systematic review of experimental evidences. *Sleep Sci.* **11**, 302–314 (2018).
 25. Suchert, V., Hanewinkel, R. & Isensee, B. Sedentary behavior and indicators of mental health in school-aged children and adolescents: A systematic review. *Prev. Med. (Baltim).* **76**, 48–57 (2015).
 26. Ferguson, C. J. *et al.* Violent Video Games Don’t Increase Hostility in Teens, but They Do Stress Girls Out. *Psychiatr. Q.* **87**, 49–56 (2016).
 27. Ryan, R. M., Rigby, C. S. & Przybylski, A. The motivational pull of video games: A self-determination theory approach. *Motiv. Emot.* **30**, 347–363 (2006).
 28. Whitaker, J. L. & Bushman, B. J. ‘Remain calm. Be kind.’ Effects of relaxing video games on aggressive and prosocial behavior. *Soc. Psychol. Personal. Sci.* **3**, 88–92 (2012).
 29. Primack, B. A. *et al.* Role of video games in improving health-related outcomes: A systematic review. *Am. J. Prev. Med.* **42**, 630–638 (2012).
 30. Pallavicini, F., Pepe, A. & Mantovani, F. Commercial off-the-shelf video games for reducing

- stress and anxiety: Systematic review. *JMIR Ment. Heal.* **8**, 1–19 (2021).
31. Ai, X., Yang, J., Lin, Z. & Wan, X. Mental Health and the Role of Physical Activity During the COVID-19 Pandemic. *Front. Psychol.* **12**, 1–8 (2021).
 32. Twenge, J. M. & Joiner, T. E. Mental distress among U.S. adults during the COVID-19 pandemic. *J. Clin. Psychol.* **76**, 2170–2182 (2020).
 33. Stekhoven, D. J. & Bühlmann, P. Missforest-Non-parametric missing value imputation for mixed-type data. *Bioinformatics* **28**, 112–118 (2012).
 34. Cunningham, S. *Causal Inference*. (Yale University Press, 2021). doi:10.2307/j.ctv1c29t27.
 35. Sony Corporation. The best PS4 and PS5 games for new and returning players. *Sony Official Website*. <https://www.playstation.com/en-us/editorial/this-month-on-playstation/perfect-games-for-new-and-returning-players/> (2023).
 36. Nintendo Co. *Nintendo Financial Results Briefing for Fiscal Year Ended March 2021*. <https://www.nintendo.co.jp/ir/pdf/2021/210507.pdf> (2021).
 37. Bertrand, M., Crepon, B., Marguerie, A. & Premand, P. *Contemporaneous and Post-Program Impacts of a Public Works Program: Evidence from Côte d'Ivoire Marianne*. *World Bank Policy Research Working Papers* <http://hdl.handle.net/10986/28460> (2017).
 38. Davis, J. M. V. & Heller, S. B. Rethinking the benefits of youth employment programs: The heterogeneous effects of summer jobs. *Rev. Econ. Stat.* **102**, 664–677 (2020).
 39. Cockx, B., Lechner, M. & Bollens, J. Priority to unemployed immigrants? A causal machine learning evaluation of training in Belgium. *Labour Econ.* **80**, 102306 (2023).
 40. Iyengar, R., Park, Y.-H. & Yu, Q. The Impact of Subscription Programs on Customer Purchases. *J. Mark. Res.* **59**, 1101–1119 (2022).

Decision Letter, first revision:

28th February 2024

Dear Dr Egami,

Thank you once again for your revised manuscript, entitled "Causal effect of video game play on mental well-being in Japan 2020-2022," and for your patience during the re-review process.

Your manuscript has now been evaluated by Reviewers 1 and 2 from the original round of review, as well as a new Reviewer (Reviewer 4) with expertise in causal inference and machine learning, as well as epidemiology. Reviewer 4 replaces Reviewer 3 from the previous round of review. All reviewer feedback is included at the end of this letter. Although the reviewers found your manuscript to have improved during revision, they also raise some important outstanding concerns. We remain very interested in the possibility of publishing your study in *Nature Human Behaviour*, but would like to consider your response to these outstanding concerns in the form of a revised manuscript before we make a decision on publication.

1. Reviewer 4 notes that PSM is not necessary when the treatment is randomly assigned, as appears to be the case here. Please either justify the use of PSM or run and present the results of alternative analyses and move your PSM analyses to the SI.

2. Reviewer 4 is concerned about the validity of your approach to estimate CATEs for the IV analysis, noting that this is not appropriate for this analysis and suffers from confounding. Please address this concern and if necessary, remove the analyses of heterogeneous effects

In sum, we invite you to revise your manuscript taking into account all reviewer and editor comments. We are committed to providing a fair and constructive peer-review process. Do not hesitate to contact us if there are specific requests from the reviewers that you believe are technically impossible or unlikely to yield a meaningful outcome.

We hope to receive your revised manuscript within 4-8 weeks. I would be grateful if you could contact us as soon as possible if you foresee difficulties with meeting this target resubmission date.

- Include a "Response to the editors and reviewers" document detailing, point-by-point, how you addressed each editor and referee comment. If no action was taken to address a point, you must provide a compelling argument. This response will be used by the editors and reviewers to evaluate your revision.
- Highlight all changes made to your manuscript or provide us with a version that tracks changes.

[REDACTED]

We look forward to seeing the revised manuscript and thank you for the opportunity to review your work. Please do not hesitate to contact me if you have any questions or would like to discuss these revisions further.

Sincerely,

[REDACTED]

Reviewer expertise:

Reviewer #1: video gaming and well-being

Reviewer #2: video gaming and well-being ; causal inference

Reviewer #4: causal inference ; machine learning ; epidemiology

REVIEWER COMMENTS:

Reviewer #3:

Remarks to the Author:

I think this manuscript is much improved in its present form.

I still think the two casual references to violent video games, however, are problematic given lack of evidence to support them.

So I would simply delete these two sentences:

“Violent video games can potentially cause psychological harm by inciting stress, though there has been little observational evidence.”

“Potential negative pathways include inadequate sleep, sedentary behavior, exposure to violent video games, and reduction of social interactions.” Lines 318-320

Neither are supportable and will confuse the reader so I'd just get rid of them.

Also, I was a bit concerned, mostly in the reviews and cover letter regarding the implication that there are significant impacts of social media on mental health, and thus that might implicate video games as well. In fact, Twenge's work has been criticized as poor quality...see rebuttals by Przybylski & Orben. Meta-analyses also have not supported these claims about social media (e.g., Ferguson, Kaye et al., 2022). Point being: I'd be careful not to go down that road too far in any further revisions. It really didn't get into the manuscript thus far...and that's good!

Otherwise, I think this is good to go!

Signed,
Chris Ferguson

Reviewer #4:

Remarks to the Author:

Referee Report for NATHUMBEHAV-23061961

I am very pleased to report that the authors carefully and thoroughly addressed my comments. In my previous referee report I raised two main concerns: first, a concern related to the external validity of the findings given that the natural experiment occurred during the COVID-19 pandemic. Second, a concern related to the conditional ignorability assumption in the empirical strategy. I find the authors' improved discussion of the external validity of the findings satisfactory and I find the imputation analysis in response to my concern about conditional ignorability sufficiently credible.

I also appreciate the author's care in responding to my other comments. I believe and hope that the additional analyses and clarifications improved the paper.

Reviewer #6:

Remarks to the Author:

The authors seek to estimate the effect of playing video games on two measures related to mental health, among a sample of survey respondents in Japan during the covid pandemic. They take advantage of a lottery system that was implemented during a period of short supply of video game consoles to identify a causal, unconfounded effect. I have several questions about the authors' approach.

- I do not understand the reason for having PSM to estimate the ITT effect of winning the lottery. Winning the lottery was randomly assigned conditional on the number of times participated, correct? In that case, there is no need for PSM. One could do multivariate regression to improve

statistical power. I would recommend deleting the PSM analyses.

- The authors say that they used random forests to estimate the CATEs of the IV method. I do not think this is accurate. They would get CATEs (conditional average treatment effects) from the ITT analysis but not the IV analysis. In the IV analysis, I think they would be estimating the complier conditional average treatment effects, which I think would be called *complier* CATEs — not CATEs.

Looking in Method 2.6 in the supplement, they indeed notate that they estimate CATEs (conditional average treatment effects), but this has nothing to do with an IV analysis, so I remain confused. Based on the supplement details and the R package they used, I think they just estimated CATEs. If this is true, then these have nothing to do with an IV analysis and, moreover, are confounded. I would recommend that they delete everything related to heterogeneous treatment effects.

- The authors discuss potential lack of generalizability/ external validity, but do not discuss the poor survey response rate and how this contributes. Also, do the authors have information on the nonrespondents that they can use to reweight their sample?

- Relatedly, in several places (e.g, the figure captions) the authors interpret their estimates as applying to the "Japanese population". They are not licensed to make such interpretations given the survey design.

- I looked in the supplement and cannot find information on the actual estimator used for estimating the LATE.

Minor

- Line 434: There is no such thing as "causal inference methods". One can be interested in a "causal effect". One can then enumerate the assumptions under which this causal effect is *identified* from the observed data. If the assumptions are believed to be met, then, one can interpret the estimate of the statistical estimand in a causal way.

Author Rebuttal, first revision:

5 April 2024

Manuscript NATHUMBEHAV-23061961

Response to Reviewers (Round 2)

We extend our deepest appreciation to the editors and reviewers for their invaluable contributions to the refinement of our manuscript. In this document, we carefully outline the modifications made to address the insightful feedback received, demonstrating our dedication to enhancing the quality and impact of our work.

Reviewer 1

Reviewer 1, Point 1:

I think this manuscript is much improved in its present form.

I still think the two casual references to violent video games, however, are problematic given lack of evidence to support them.

So I would simply delete these two sentences:

“Violent video games can potentially cause psychological harm by inciting stress, though there has been little observational evidence.”

“Potential negative pathways include inadequate sleep, sedentary behavior, exposure to violent video games, and reduction of social interactions.” Lines 318-320

Neither are supportable and will confuse the reader so I'd just get rid of them.

Also, I was a bit concerned, mostly in the reviews and cover letter regarding the implication that there are significant impacts of social media on mental health, and thus that might implicate video games as well. In fact, Twenge's work has been criticized as poor quality...see rebuttals by Przybylski & Orben. Meta-analyses also have not supported these claims about social media (e.g., Ferguson, Kaye et al., 2022). Point being: I'd be careful not to go down that road too far in any further revisions. It really didn't get into the manuscript thus far...and that's good!

Otherwise, I think this is good to go!

We greatly appreciate your thorough review and guidance on refining our manuscript. We have carefully revised the manuscript to ensure clarity and adherence to the evidence-based discussion regarding the impact of video games on mental health.

Following your recommendation, we first removed the row of violent video games, which included the sentence (“Violent video games can potentially cause psychological harm by inciting stress, though there has been little observational evidence”) from Supplementary Table 10.

Secondly, we removed the sentence (“Potential negative pathways include inadequate sleep, sedentary behavior, exposure to violent video games, and reduction of social interactions”) from the main text. This was done alongside minor modifications to enhance the readability of the paragraph:

“Previous research has proposed mechanisms for both positive and negative effects of video gaming on mental well-being (summarized in Supplementary Table 10). Our study found that positive effects outweigh negative effects, resulting from both positive and negative pathways. Positive pathways include psychological therapy games, mood management theory, self-determination theory, and social connection hypothesis.¹⁻⁴ For example, relaxation games can induce a positive mood and improve well-being, as in psychological therapy. An example of an adverse pathway is playing video games to such an extent that it results in insufficient sleep, possibly harming well-being.^{5,6} While evaluating the relative impact of each pathway is beyond the scope of this study, understanding potential mechanisms from the literature is essential. Conducted during the COVID-19 pandemic, the study’s context may affect how these pathways operate. Thus, interpreting our estimates requires considering the possible effects that the COVID-19 circumstances might exert on these potential pathways.” Lines 330-341

Reviewer 2

I am very pleased to report that the authors carefully and thoroughly addressed my comments. In my previous referee report I raised two main concerns: first, a concern related to the external validity of the findings given that the natural experiment occurred during the COVID-19 pandemic. Second, a concern related to the conditional ignorability assumption in the empirical strategy. I find the authors' improved discussion of the external validity of the findings satisfactory and I find the imputation analysis in response to my concern about conditional ignorability sufficiently credible.

I also appreciate the author's care in responding to my other comments. I believe and hope that the additional analyses and clarifications improved the paper.

Thank you very much for your positive feedback. We appreciate your comments profoundly enhancing the quality of our work.

Reviewer 4

The authors seek to estimate the effect of playing video games on two measures related to mental health, among a sample of survey respondents in Japan during the covid pandemic. They take advantage of a lottery system that was implemented during a period of short supply of video game consoles to identify a causal, unconfounded effect. I have several questions about the authors' approach.

Reviewer 4, Point 1:

- I do not understand the reason for having PSM to estimate the ITT effect of winning the lottery. Winning the lottery was randomly assigned conditional on the number of times participated, correct? In that case, there is no need for PSM. One could do multivariate regression to improve statistical power. I would recommend deleting the PSM analyses.

Thank you for highlighting PSM (propensity score matching)'s role in our study. We would like to share why we find PSM valuable here. We have two key reasons: (i) aspects unique to lottery-based studies and (ii) considerations common to natural experiment studies (or observational studies).

(i) Aspects unique to lottery-based studies

Natural experiment studies using lotteries, including ours, have advantages regarding internal validity over typical (non-experimental) observational studies as they exploit plausibly random variations. However, it should be noted that lottery studies are not the same as fully randomized experiments. Even if a lottery randomly determines the winners, there may be systematic differences within the subset of individuals who responded to the survey and are included as the analysis sample. Moreover, most lottery studies are not even 'the ideal lottery study'—argued by Doherty et al. (2006)⁷—that examines those who purchased the same number of lottery tickets in a given game. Therefore, we need to address several methodological challenges.

In a randomized experiment, every subject has the same probability of receiving a treatment. Yet, most lottery studies do not meet this criterion and attempt to address the problem. A typical approach is controlling the source of having different probabilities of winning a treatment. In Imbens et al. (2001) and Doherty et al. (2006),^{7,8} two well-cited lottery studies, the number of lottery tickets purchased was controlled in their regression analyses. Note that both papers exploit lotteries in the U.S.; individuals who won the lotteries obtained prizes (the outcome variables are labor earnings in the former paper and attitude toward government redistribution in the latter paper). The number of tickets bought is included as a covariate because, by the nature of the lottery, individuals purchasing more tickets are more likely to win a lottery; therefore on a priori grounds, researchers expect that the

variable affects the probability of winning a lottery. In our study, we also treated the number of times respondents participated in lotteries as a covariate. However, these methods simplify the actual complexities—differential lottery participating behaviour—involved in each lottery scenario.

Natural experiment studies using lotteries are not free from the methodological challenge stemming from differential lottery participating behaviour.⁷⁻⁹ For example, Imbens et al. (2001)⁸ and Doherty et al. (2006)⁷ treated different types of lotteries (or lottery tickets) as identical in their OLS regression models (for example, the ‘season ticket’ buyers and ‘single-ticket’ buyers are treated as comparable in Imbens et al. 2001). In our study’s context, we also had to consider various lotteries as equivalent, even though they might offer different odds of winning, such as different Nintendo Switch lotteries or PS5 lotteries. Despite all targeting a chance to buy a Nintendo Switch or a PS5, each lottery had its nuances, like varying retailers (for example, Amazon, Yodobashi Camera, and Sony Store) or timings. These differences could affect the likelihood of winning. For instance, as the supply-side challenge for the Nintendo Switch started in March 2020 and ended in January 2021, winning odds likely shifted over time. Therefore, each lottery could have different probabilities of winning the lottery, and the number of times of lottery participation reflects the differential lottery participating behaviour of the respondents.

When a study is not ‘the ideal lottery study’⁷ examining those who purchased the same number of lottery tickets in a given game, assuming a simple linear relationship for the number of lottery entries in a regression model may become problematic. It is acknowledged that based on the context information, we a priori know that this variable should affect the propensity score. However, the precise functional form of this variable remains uncertain. It may exhibit a quadratic or more complex polynomial relationship, or involve interaction effects with other variables, among other possibilities. In such scenarios, matching methods help tackle these methodological challenges.^{9,10}

Imbens (2015)⁹ discussed the application of PSM for estimating causal effects and highlighted the ‘Imbens-Rubin-Sacerdote lottery’ study⁸ as a prime example where PSM could effectively complement traditional OLS regression analyses. Applying the recommended PSM procedures to the data from Imbens et al. (2001),⁸ Imbens demonstrated the robustness of the original findings. The recommended practices include estimating the propensity score in a data-driven manner, improving overlap by trimming units with extreme values of the propensity score, and assessing the plausibility of the unconfoundedness assumption.^{9,10}

In both Imbens et al. (2001)⁸ and Imbens (2015),⁹ the authors argued two major methodological challenges in the ‘Imbens-Rubin-Sacerdote lottery’ study. The first challenge is the differential ticket buying behavior, which implies that the inclusion of the number of tickets bought as a covariate in regression analysis might not fully address potential confounders. The second challenge is nonresponses to the survey (response rate 46% in Imbens et al. 2001⁸), which could lead to systematic differences between lottery winners and non-winners. Although the lottery mechanism allows randomization, the sample of individuals who respond to the survey would not be random. In addition to those, we may consider self-reported ticket purchasing behavior as another potential source of bias, a problem that does not arise in randomized experiments with direct access to administrative data.

Therefore, our strategy to tackle those methodological challenges is to incorporate PSM alongside regression analysis, following the procedure recommended by Imbens (2015).⁹ This reduces reliance on the specific assumptions of the OLS regression model, thereby mitigating the bias caused by the differential ticket buying behavior. Additionally, it enhances the overlap in the covariate distribution—one of the critical concerns of using linear regression. Using PSM, we can better ensure that the matched groups are comparable in their observable characteristics, as shown in Supplementary Figures 2-5.¹¹ Thereby, our approach helps to lessen potential biases caused by nonresponse and self-reporting. It should be noted that PSM relies on the propensity score model’s functional form and does not address bias due to unobserved confounders as caveats.

(ii) Considerations common to natural experiment studies (or observational studies)

The causal literature has acknowledged the limitations of relying exclusively on OLS linear regression for estimating causal effects in observational studies.⁹⁻¹¹ Although the OLS estimator and matching and propensity score estimators depend on the same unconfoundedness assumption, the former incorporates this assumption alongside strong functional form restrictions.⁹ OLS estimates rely heavily on the assumption that the true model is known and correctly specified by the researcher, which is optimal under such conditions. Yet in practice, in observational studies, it is often unrealistic to adhere to this presumption (for example, a researcher may not have a priori views on whether a covariate should enter linearly or quadratically in the regression). Therefore, it is recommended that violations of the assumptions be anticipated.⁹⁻¹¹ Moreover, when the covariate distribution is different between the comparison groups, OLS estimates become particularly sensitive to slight changes in model specification due to their reliance on extrapolation. Ensuring a balanced distribution of covariates is essential, but OLS adjustments may not achieve this across all relevant variables. The extent to which OLS adjustments improve balance is often unclear, and rigorous diagnostic checks are frequently missing in studies that depend on OLS.¹¹

Therefore, in observational studies, it is recommended not to solely rely on OLS estimates and instead compare these estimates with those derived from other sophisticated methods, such as matching.⁸⁻¹⁰ This approach serves as a robustness check, ensuring that the results are not sensitive to the choice of estimators. Such a practice is also suitable for understanding the factors driving differences between the estimates.⁹ Furthermore, rather than assuming the ability to precisely define the true model through regression, focusing on building good contrasts and approximating an experiment as closely as possible is recommended.¹¹

Our study compared regression and PSM estimates across various models, as illustrated in Figure 1 and Supplementary Figure 10. The consistency of our results lends support to their reliability. Notably, we observed that the regression estimates without adjusting for covariates closely resemble estimates from other models, including PSM (Supplementary Figure 10). This consistency supports the plausibility of assuming the unconfoundedness of the lottery win variable within our dataset. However, it is crucial to underline that our confidence in the unconfoundedness assumption was reinforced only after extensive analysis involving multiple models, including those employing PSM.

Building on the above discussion, we have updated our manuscript to more thoroughly articulate the relevance of PSM to our study. Similar to how other lottery-based studies detail the inherent constraints of their natural experiments,^{7,8} we realize the importance of doing the same in our work. Specifically, we revised the sections of Discussion and Statistical analysis.

Discussion

“The causal inference literature has increasingly acknowledged the importance of not solely relying on OLS (Ordinary Least Squares) regression and comparing OLS estimates with those derived from other methods.⁹⁻¹¹ Following relevant recommendations, we compared regression and PSM estimates across various models (Figure 1 and Supplementary Figure 10). The consistency of our results lends credibility to our findings. Notably, we observed that the regression estimates without adjusting for covariates closely resemble estimates from other models, including PSM (Supplementary Figure 10). This supports the plausibility of assuming the unconfoundedness of the lottery-win variable within our dataset.” Lines 274-281

Statistical analysis

“We used three methods: multivariate regression, propensity score matching approach (PSM), and instrumental variable (IV) method. We additionally utilized a machine learning algorithm called causal forest (or generalized random forests).^{12,13} Methodological particulars are provided in Table 2, while the underlying causal assumptions are discussed in Supplementary Method 2.1 and 2.9. The methodological challenges encountered in lottery-based natural experiments and the rationale for employing PSM alongside linear regression are elaborated in a subsequent subsection.” Lines 486-491

Multivariate regression and PSM

“It is worth noting that the causal inference literature has acknowledged the limitations of relying exclusively on OLS linear regression in observational studies.⁹⁻¹¹ It is recommended to compare the OLS estimates with those derived from other sophisticated methods, such as matching.^{9,10} This approach ensures that the results are not sensitive to the choice of estimators.” Lines 510-513

Methodological challenges in lottery-based natural experiments

“Natural experiment studies using lotteries, including ours, have advantages regarding internal validity over typical (non-experimental) observational studies as they exploit plausibly random variations. However, it should be noted that lottery studies are not the same as fully randomized experiments. Even if a lottery randomly determines the winners, there may be systematic differences within the subset of individuals who responded to the survey and are included as the analysis sample. Moreover, most lottery studies are not even ‘the ideal lottery study’—argued by Doherty et al. (2006)⁷—that examines those who purchased the same number of lottery tickets in a given game. Therefore, we need to address several methodological challenges.

In a randomized experiment, every subject has the same probability of receiving a treatment. Yet, most lottery studies do not meet this criterion and attempt to address the problem. A typical approach is controlling the source of having different probabilities of winning a treatment. In Imbens et al. (2001) and Doherty et al. (2006),^{7,8} two well-cited lottery studies, the number of lottery tickets purchased was controlled in their regression analyses. Note that both papers exploit lotteries in the U.S.; individuals who won the lotteries obtained prizes (the outcome variables are labor earnings in the former paper and attitude toward government redistribution in the latter paper). The number of tickets bought is included as a covariate because, by the nature of the lottery, individuals purchasing more tickets are more likely to win a lottery; therefore on a priori grounds, researchers expect that the variable affects the probability of winning a lottery. In our study, we also treated the number of times respondents participated in lotteries as a covariate. However, these methods simplify the actual complexities—differential lottery participating behavior—involved in each lottery scenario.

Natural experiment studies using lotteries are not free from the methodological challenge stemming from differential lottery participating behavior.⁷⁻⁹ For example, Imbens et al. (2001)⁸ and Doherty et al. (2006)⁷ treated different types of lotteries (or lottery tickets) as identical in their OLS regression models (for example, the ‘season ticket’ buyers and ‘single-ticket’ buyers are treated as comparable in Imbens et al. 2001). In our study’s context, we also had to consider various lotteries as equivalent, even though they might offer different odds of winning, such as different Nintendo Switch lotteries or PS5 lotteries. Despite all targeting a chance to buy a Nintendo Switch or a PS5, each lottery had its nuances, like varying retailers (for example, Amazon, Yodobashi Camera, and Sony Store) or timings. These differences could affect the likelihood of winning. For instance, as the supply-side challenge for the Nintendo Switch started in March 2020 and ended in January 2021, winning odds likely shifted over time. Therefore, each lottery could have different probabilities of winning the lottery, and the number of times of lottery participation reflects the differential lottery participating behavior of the respondents. Further context information is available in Supplementary Method 1.

When a study is not ‘the ideal lottery study’⁷ examining those who purchased the same number of lottery tickets in a given game, assuming a simple linear relationship for the number of lottery entries in a regression model may become problematic. It is acknowledged that based on the context information, we a priori know that this variable should affect the propensity score. However, the precise functional form of this variable remains

uncertain. It may exhibit a quadratic or more complex polynomial relationship, or involve interaction effects with other variables, among other possibilities. In such scenarios, matching methods help tackle these methodological challenges.^{9,10}

Imbens (2015)⁹—written based on Imbens and Rubin (2015)¹⁰—discussed the application of PSM for estimating causal effects and highlighted the ‘Imbens-Rubin-Sacerdote lottery’ study⁸ as a prime example where PSM could effectively complement traditional OLS regression analyses. Applying the recommended PSM procedures to the data from Imbens et al. (2001),⁸ Imbens demonstrated the robustness of the original findings. The recommended practices include estimating the propensity score in a data-driven manner, improving overlap by trimming units with extreme values of the propensity score, and assessing the plausibility of the unconfoundedness assumption.

In both Imbens et al. (2001)⁸ and Imbens (2015),⁹ the authors argued two major methodological challenges in the ‘Imbens-Rubin-Sacerdote lottery’ study. The first challenge is the differential ticket buying behavior, which implies that the inclusion of the number of tickets bought as a covariate in regression analysis might not fully address potential confounders. The second challenge is nonresponses to the survey (response rate 46% in Imbens et al. 2001⁸), which could lead to systematic differences between lottery winners and non-winners. Although the lottery mechanism allows randomization, the sample of individuals who respond to the survey would not be random. In addition to those, we may consider self-reported ticket purchasing behavior as another potential source of bias, a problem that does not arise in randomized experiments with direct access to administrative data.

Therefore, our strategy to tackle those methodological challenges is to incorporate PSM alongside regression analysis, following the procedure recommended by Imbens (2015).⁹ This reduces reliance on the specific assumptions of the OLS regression model, thereby mitigating the bias caused by the differential ticket buying behavior. Additionally, it enhances the overlap in the covariate distribution—one of the critical concerns of using linear regression. Using PSM, we can better ensure that the matched groups are comparable in their observable characteristics, as shown in Supplementary Figures 2-5.¹¹ Thereby, our approach helps to lessen potential biases caused by nonresponse and self-reporting. It should be noted that PSM relies on the propensity score model’s functional form and does not address bias due to unobserved confounders as caveats.” Lines 530-602

We hope the explanation has clarified our approach. If you find it insufficient, we are open to relocating the PSM analysis to the supplementary material in the next revision.

Reviewer 4, Point 2:

*- The authors say that they used random forests to estimate the CATEs of the IV method. I do not think this is accurate. They would get CATEs (conditional average treatment effects) from the ITT analysis but not the IV analysis. In the IV analysis, I think they would be estimating the complier conditional average treatment effects, which I think would be called *complier* CATEs — not CATEs.*

Looking in Method 2.6 in the supplement, they indeed notate that they estimate CATEs (conditional average treatment effects), but this has nothing to do with an IV analysis, so I remain confused. Based on the supplement details and the R package they used, I think they just estimated CATEs. If this is true, then these have nothing to do with an IV analysis and, moreover, are confounded. I would recommend that they delete everything related to heterogeneous treatment effects.

We appreciate the opportunity to refine our manuscript based on the valuable feedback. In response to your comment, we wish to clarify and address the issues through the following key points: (i) Revisiting terminology: CATE versus CLATE in instrumental forest analysis, (ii) Applicability of

instrumental variable causal forests in our study, (iii) Assessment of potential confounding in instrumental forest estimates.

(i) Revisiting terminology: CATE versus CLATE in instrumental forest analysis

Our initial use of terminology might have led to confusion. This issue not only pertains to the interpretation of our estimates but is also of conceptual significance. In our discussion of the estimand for instrumental variable causal forests, we realize our descriptions were not as precise as they should have been.

Typically, causal forests aim to estimate the Conditional Average Treatment Effect (CATE), assuming perfect compliance. However, akin to how traditional instrumental variable methods yield the Local Average Treatment Effect (LATE), instrumental variable causal forests are designed to estimate the Conditional Local Average Treatment Effect (CLATE). Therefore, our manuscript should have described our estimand as CLATE, rather than CATE. Alternatively, we could have adopted a conditional homogeneity assumption, making CLATE and CATE equivalent.¹⁴ This indicates that the effect of the treatment is consistent across all subgroups defined by covariates, implying the effect does not vary between compliers and non-compliers.

In the literature on heterogeneous treatment effects, the usage of CLATE and CATE varies. For instance, Athey et al. (2019)¹³ and Wang et al. (2022)¹⁵ directly use the term CLATE. In their simulation work with instrumental forests, Athey & Wager (2021)¹⁴ refer to their CLATE estimates as CATE, assuming conditional homogeneity. Conversely, Brooks et al. (2022)¹⁶ avoid specifying their estimand as either CLATE or CATE, leaving the distinction unclear.

Following your feedback, we have revised our manuscript to consistently use the term CLATE. We are convinced that this clarification significantly improves the clarity and precision of our analysis, leading to a more precise interpretation of our findings.

Given the frequent mention of the term CLATE in our manuscript, we have outlined the key revisions below.

Main text

Results

“Figure 3 demonstrates conditional local average treatment effect (CLATE)¹³ estimates, predicted by the instrumental variable causal forest algorithm (also called instrumental variable forest or instrumental forest), for ownership of Switch or PS5, respectively. First, panels a and b depict histograms of estimated CLATEs (outcome: K6), indicating that video gaming positively impacts the mental well-being of most individuals. The illustrated histograms of CLATEs align with the LATE derived through the IV method, thereby supporting the credibility of the estimated CLATEs.” Lines 215-220

Supplementary material

Supplementary Method 2.6. Machine learning (causal forest) details.

“We used the instrumental variable causal forest.^{13,14} We predicted conditional LATEs (CLATEs, conditional local average treatment effects) (given treatment W_i and instrument Z_i , treatment effect $\tau(x)$ is identified via $\tau(x) = Cov[Y_i, Z_i | X_i = x] / Cov[W_i, Z_i | X_i = x]$) capturing the causal effect of ownership of video game consoles (Nintendo Switch and PlayStation5, respectively for each estimation). For the identification of treatment effects, a set of assumptions for the instrumental variable approach must hold (discussed in

Supplementary Method 2.9). Note that given a conditional homogeneity assumption, the CLATE is simply the conditional ATE (CATE; $\tau(x) = E(Y_{1i} - Y_{0i} | X_i = x)$)." Lines 338-345

(ii) Applicability of instrumental variable causal forests in our study

(a) Overview of Generalized Random Forests, Causal Forests, and Instrumental Variable Causal Forests

In our analysis of heterogeneous treatment effects within the context of video gaming, we selected Generalized Random Forests (GRF),¹³ specifically its instrumental variable causal forests component. Several machine learning algorithms are suited for analyzing heterogeneous treatment effects,^{17,18} yet a few incorporate instrumental variable techniques^{13,15} essential for causal inference amidst imperfect compliance. The reliability and usefulness of GRF are further supported by the literature^{16,19-22} including empirical studies²³⁻³² and methodological guides.^{21,29,33} Particularly, the literature includes a detailed assessment of the utility of instrumental variable causal forests¹⁶ and an application paper using instrumental forests.¹⁴ Additionally, the active user community and resourceful discussions on the GRF R package forum³⁴ provide practical insights and support, further validating our choice. These elements, taken together, underscore the appropriateness of GRF for exploring the heterogeneity in video gaming effects.

GRF, formulated by Athey et al. (2019),¹³ involve generating a series of causal trees³⁵ from randomly selected data subsets, with each tree comprising a series of decision splits based on specific variables and cutoff values. The algorithm selects splits that maximize the difference in treatment effects, continually partitioning the data until it forms groups, or “leaves,” with similar effects. For counteracting overfitting, the algorithm uses the selected subset to construct the tree while it estimates treatment effects using the other subgroup. Recognizing the potential for instability in the estimates from individual trees, GRF compute many trees, each utilizing different samples and variables, to create a complete forest. This ensemble approach ensures the robustness and consistency of the treatment effect estimates across various subsamples.

Delving deeper into GRF’s unique methodology, a crucial preprocessing step called ‘labeling’ precedes the standard CART (Classification and Regression Tree) regression splits.¹³ This step involves encoding the specific structure of the type of problem to be solved, a process enabling the integration of instrumental variables into the forest framework. By doing so, it tailors the algorithm to the specific characteristics of both the data and the question being addressed. Utilizing its encoded information, the algorithm optimizes its criterion. This facilitates the application of random forests to model a wide array of quantities of interest identified as the solutions to respective local moment equations.

Causal forests¹² laid the groundwork for GRF, which broadened the scope to include causal forest techniques and more, making it a vital tool for examining treatment effect heterogeneity. The causal forest algorithm, developed by Wager & Athey (2018),¹² quickly gained scholarly interest, and applications were already evident by 2017.^{36,37} This rapid adoption highlights its utility, now extended by GRF, making it one of the preferred methods in heterogeneous treatment effect studies.¹⁹ While causal forest assumes a setup with a randomized treatment allocation with perfect compliance, GRF, which include causal forests as a part of the algorithm, addresses scenarios with imperfect compliance, employing instrumental variables.¹³

GRF are versatile and fits well with various research designs, including regression discontinuity design and panel data analysis.¹³ One of its primary uses, as outlined by Athey et al. (2019),¹³ is the estimation of heterogeneous treatment effects with instruments. They demonstrate this application using the Angrist & Evans³⁸ dataset, a well-known case in the field, providing detailed results,

interpretations, and R code for replication. Note that heterogeneous treatment effect estimation via instrumental variables using the GRF is called in several ways: instrumental variables forests,¹³ instrumental forests,³⁴ or instrumental variable causal forests.¹⁶

The key assumption of GRF¹³ for interpreting the estimates causal is the standard unconfoundedness assumption given covariates, formally expressed as $(Y_{0i}, Y_{1i}) \perp D_i \mid X_i$. Likewise, the fundamental assumption underlying instrumental forests^{13,16} aligns with the traditional instrumental variable approach (detailed in Supplementary Methods 2.1 and 2.5; for example, conditional independence of instrument Z_i). Thus, in observational studies employing GRF, researchers should support assuming the unconfoundedness of the source of quasi-randomness. Our paper conducts such evaluations, offering detailed discussions that validate the lottery as our instrument.

(b) Examples of Instrumental Variable Causal Forest applications

To demonstrate the appropriateness of instrumental forests for our study, we reference two examples where their application within natural experimental setups closely aligns with the identification strategy of our study.

Athey et al. (2019)¹³ use data from Angrist & Evans (1998),³⁸ a popular study employing the instrumental variable method, to provide an example of the application of instrumental forests. Angrist & Evans (1998)³⁸ explored the impact of family size on mothers' labor market participation by utilizing an instrument: the sex of the first two children. Their study is based on the idea that families may be more inclined to have additional children if the first two are of the same sex, suggesting that some parents might desire a gender mix among their offspring. This situation creates a natural experiment, as the sex of the children is essentially random and unrelated to the parents' inherent characteristics that could influence labor market outcomes. The unconfoundedness regarding the sex of the children is one of the key assumptions for applying instrumental forests.

Brooks et al. (2022)¹⁶ assess the ability of instrumental variable causal forests by employing Floyd et al. (2020)³⁹ data, which provided a natural experimental situation. Using the data, Brooks et al. (2022)¹⁶ explored early surgery's personalized effects on shoulder fractures as an example of the application of the algorithm. Specifically, Brooks et al. (2022)¹⁶ investigated the effects of early surgery on outcomes for shoulder fracture patients using an instrumental variable: the local area surgery rates (Area Surgery Ratio; ASR) indicative of regional practice styles. This hinges on the observation that the propensity for early surgery might vary significantly between regions, influenced more by localized medical practices than by patient-specific factors. This variation creates a quasi-experimental setup, allowing the researchers to infer causal effects by exploiting the randomness in ASR attributed to geographical differences in practice styles. Thus, using ASR as an instrumental variable is analogous to employing the randomness of the first two children's sex in Angrist & Evans (1998).³⁸ The authors, therefore, assume that the ASR provides an exogenous variation to isolate the impact of early surgery from confounding factors.

These examples parallel the design of our study, where we use the lottery as a quasi-random instrument. The methodological similarity lends support to the use of instrumental forests in our study.

(c) Presentation of Results from Generalized Random Forests

The manner in which we present GRF results, particularly through figures, aligns with emerging practices in visualizing heterogeneous treatment effect analysis in empirical research. Specifically, we utilize heatmaps to visually represent the heterogeneous treatment effects across various dimensions. Recent studies³⁰ have effectively employed this method to articulate complex data patterns clearly.

Additionally, our depiction of the distribution of estimated C(L)ATEs alongside the (L)ATE in Figure 3 (panels a and b) is a straightforward and common method aimed at making complex results understandable.^{24,30} We also employ scatter plots to outline predicted heterogeneous treatment effects, a method widely recognized for its simplicity and effectiveness.^{30,32,36} Thus, our presentation style, though yet to be widespread due to the novelty of the technique, is consistent with the reporting style found in other scholarly works and effectively visualizes the heterogeneous treatment effects across multiple dimensions among diverse subsamples.

(iii) Assessment of potential confounding in instrumental forest estimates

The C(L)ATEs estimated by GRF can be considered causal under the standard unconfoundedness assumption,¹² formally expressed as $(Y_{0i}, Y_{1i}) \perp D_i \mid X_i$. As explained above, the fundamental assumption underlying instrumental forests^{13,16} aligns with the traditional instrumental variable approach (for example, conditional independence of instrument Z_i). The assumptions including the unconfoundedness assumption, central to our study, are extensively discussed in Supplementary Methods 2.1 and 2.5.

Hence, the plausibility of assuming unconfoundedness for the variation introduced by our natural experiment—the lottery—is essential. While unconfoundedness is not testable, it is recommended to assess whether the assumption is plausible.⁹ Accordingly, our paper assesses the unconfoundedness thoroughly from multiple perspectives (particularly, Supplementary Tables 3-6 and Supplementary Figure 10).

Furthermore, in Figure 3 (panels a and b), we compare the estimates obtained from instrumental forests with those derived from traditional instrumental variable regressions. The estimated CLATEs are distributed around the LATEs, supporting the reliability of our instrumental forest estimates.

In response to your concern, we provide an additional robustness check in this letter to assess whether our instrumental forest estimates suffer from confounding (Supplementary Figure 19). We compare the LATEs estimated by the traditional instrumental variable regressions and those estimated by instrumental forests.

GRF extend their utility beyond estimating subgroup treatment effects by also enabling the estimation of LATE across the sample.³⁴ The LATE estimated using the GRF is referred to as the Average Conditional Local Average Treatment Effect (ACLATE), reflecting the estimation methodology.³⁴ The GRF employs Augmented Inverse-Probability Weighting (AIPW) for this purpose.^{34,40} In our study, we found that simply averaging the CLATEs across all samples produces a point estimate that closely mirrors the AIPW estimate while using the AIPW estimates is recommended.³⁴

Supplementary Figure 19 compares the LATE estimates derived from traditional instrumental variable regressions and instrumental forests. To facilitate clear understanding, we designed Supplementary Figure 19 to reflect the layout of Figure 1 of the main text. The consistency between estimates from instrumental forests and those obtained through instrumental variable regressions serves as another piece of supportive evidence for the reliability of our approach, particularly ensuring the plausibility of the unconfoundedness of the heterogeneous treatment effect estimates. Although this figure and Figure 3 (panels a and b) present similar information, the unique visual layout of Supplementary Figure 19 may clarify how closely the estimates from instrumental forests and instrumental variable regressions align.

Comparison of estimates between instrumental variable regression and instrumental variable causal forest on the impact of video game console possession on well-being in Japan (N=8,192).

The causal effect of possession of video game consoles on well-being (LATE) is estimated by instrumental variable regression and machine learning (instrumental variable causal forest algorithm). The analysis sample is limited to those who joined game console lotteries. The point estimates and the 95 percent confidence intervals are shown. Regression standard errors are clustered by prefectures. GRF standard errors estimated via bootstrapping. Exposure variables are possession of video game consoles (Nintendo Switch and PlayStation5 for each). The instrumental variable regression and the instrumental forests use a set of covariates shown in Supplementary Table 15. A lower K6 means having less psychological distress, while a higher SWLS means greater life satisfaction. The estimates are standardized by the standard deviations. LATE, Local Average Treatment Effect; CI, confidence intervals; IV, Instrumental Variable, 2SLS, Two-stage least squares; GRF, Generalized Random Forests.

Following the discussion above, we have made the necessary revisions throughout our manuscript including Table 2. Considering your comments, we have expanded our explanations of GRF and instrumental forests. Although the brevity of our descriptions may have initially led to some confusion, we acknowledge that there was indeed scope to enhance our exposition of the machine learning techniques employed. We are confident that these additional clarifications have improved the quality of our paper.

Please find the key revisions listed below.

Main text

Statistical analysis

Machine learning

“Causal forest (or generalized random forests)^{12,13} predicts treatment effects on each individual based on their characteristics. This method, used in recent studies,^{27,28} allows flexible visualization and helps explore effect modification (or moderating effects). We specifically used instrumental variable causal forest¹³ evaluated by a recent study¹⁶ to estimate conditional **local** average treatment effects (CLATEs), capturing the effect of ownership of Switch/PS5. **While causal forests estimate conditional average treatment effects (CATEs),**

instrumental variable causal forests estimate CLATEs that match CATEs only when the conditional homogeneity assumption holds.

To explore the multidimensional nature of the video gaming effect's heterogeneity, we selected Generalized Random forests (GRF)—specifically its instrumental variable causal forests component—formulated by Athey et al. (2019).¹³ Several machine learning algorithms are suited for analyzing heterogeneous treatment effects,^{17,18} yet a few incorporate instrumental variable techniques^{13,15} essential for causal inference amidst imperfect compliance. The reliability and usefulness of GRF are further supported by the literature,^{14,16,19–22} which includes empirical studies^{23–32} and methodological guides.^{21,29,33} Note that heterogeneous treatment effect estimation via instrumental variables using the GRF is called in several ways: instrumental variables forests,¹³ instrumental forests,³⁴ or instrumental variable causal forests.¹⁶

GRF involve generating a series of causal trees³⁵ from randomly selected data subsets, with each tree comprising a series of decision splits based on specific variables and cutoff values. The algorithm selects splits that maximize the difference in treatment effects, continually partitioning the data until it forms groups, or “leaves,” with similar effects. For counteracting overfitting, the algorithm uses the selected subset to construct the tree while it estimates treatment effects using the other subgroup. Recognizing the potential for instability in the estimates from individual trees, GRF compute many trees, each utilizing different samples and variables, to create a complete forest. This ensemble approach ensures the robustness and consistency of the treatment effect estimates across various subsamples.

Delving deeper into GRF's unique methodology, a crucial preprocessing step called ‘labeling’ precedes the standard CART (Classification and Regression Tree) regression splits.¹³ This step involves encoding the specific structure of the type of problem to be solved, a process enabling the integration of instrumental variables into the forest framework. By doing so, it tailors the algorithm to the specific characteristics of both the data and the question being addressed. Utilizing its encoded information, the algorithm optimizes its criterion. This facilitates the application of random forests to model a wide array of quantities of interest identified as the solutions to respective local moment equations.” Lines 615-650

Supplementary material

Supplementary Method 2.1. Causal assumptions.

“In addition to OLS linear regression models, PSM, and instrumental variable methods, we employed instrumental variable causal forests. The estimates drawn by this technique also need assumptions to be considered causal. Further details are discussed in Method 2.6 and 2.9.” Lines 271-273

Supplementary Method 2.9. Causal assumptions and reliability in instrumental variable causal forest estimation.

“The C(L)ATEs estimated by generalized random forests (GRF) can be considered causal under the standard unconfoundedness assumption,¹² formally expressed as $(Y_{0i}, Y_{1i}) \perp D_i \mid X_i$. This foundational assumption, central to our study, has been extensively discussed in Supplementary Method 2 (this section). Likewise, the assumption underlying instrumental forests^{13,16} aligns with the traditional instrumental variable approach (discussed in Supplementary method 2.1 and 2.5), including conditional independence of the instrument.

Hence, the plausibility of assuming unconfoundedness for the variation introduced by our natural experiment—the lottery—is essential for interpreting the instrumental variable causal forest estimates as causal. While unconfoundedness is not testable, it is recommended to assess whether the assumption is plausible.⁹ Accordingly, we assessed the unconfoundedness thoroughly from multiple perspectives (particularly, Supplementary Tables 3-6 and Supplementary Figure 10).

Regarding the reliability of our instrumental variable causal forest estimation, in Figure 3 (panels a and b), we compare the estimates obtained from instrumental forests with those derived from traditional instrumental variable regressions. The estimated CLATEs are distributed around the LATEs, supporting the reliability of our

instrumental forest estimates. In Supplementary Result 9, we provide further information on a concise additional check on whether our instrumental forest estimates suffer from confounding.” Lines 431-447

Supplementary Result 9. Examination of confounding in instrumental variable causal forest estimation.

“Here, we provide an additional robustness check for assessing whether our instrumental forest estimates suffer from confounding. We compare the LATEs estimated by the traditional instrumental variable regressions and those estimated by instrumental forests.

GRF extend their utility beyond estimating subgroup treatment effects by also enabling the estimation of LATE across the sample.³⁴ The LATE estimated using the GRF is referred to as the Average Conditional Local Average Treatment Effect (ACLATE), reflecting the estimation methodology.³⁴ The GRF employ Augmented Inverse-Probability Weighting (AIPW) for this purpose.^{34,40} In our study, we found that simply averaging the CLATEs across all samples produces a point estimate that closely mirrors the AIPW estimate while using the AIPW estimates is recommended.³⁴

Supplementary Figure 19 compares the LATE estimates derived from instrumental variable regressions (Two-Stage Least Squares) and instrumental forests. To facilitate clear understanding, we designed Supplementary Figure 19 to reflect the layout of Figure 1 of the main text. The consistency between estimates from instrumental forests and those obtained through instrumental variable regressions serves as another piece of supportive evidence for the reliability of our approach, particularly ensuring the plausibility of the unconfoundedness of the heterogeneous treatment effect estimates. Although this figure and Figure 3 (panels a and b) present similar information, the unique visual layout of Supplementary Figure 19 may clarify how closely the estimates from instrumental forests and instrumental variable regressions align.” Lines 634-652

Overview of statistical analysis

Methods	Estimand	Exposure/ Treatment (assignment) variables	Excluded instrument	Advantages/purposes	Details/Equations
Multivariate regression				Simple and easy to understand, thus suitable to be a baseline specification. This method can be extended to the instrumental variable approach.	Following Imbens et al. (2001) we used Equation 1 in Supplementary Methods.
Propensity score matching method	ITT effect	Winning game console lottery.		While the linear regression approach relies on assumptions of linearity and extrapolations beyond observed variable combinations to correct for differences between the treatment and control groups, the PSM approach does not rely on such parametric assumptions.	Following Imbens (2015), we used the algorithm to select matching variables and also constructed a subsample of the original data set. We conducted covariate balance checks and examined the common support of the propensity score distribution. Further details and the PSM design choice are explained in Supplementary Methods.
Instrumental variable method	LATE	Possession of Nintendo Switch/PS5. Played Nintendo Switch/PS5 last month. Video gameplay time.	Winning game console lottery.	This method can estimate the causal effect of standard exposure variables (for example, owning a Nintendo Switch/PS5) among compliers. Additionally, this method is utilized for a subgroup analysis investigating whether the impact of video gaming varies based on the time spent.	The Two-stage least squares (2SLS) estimation technique is employed. The first-stage regression results are found in Supplementary Table 9. Weak instrument tests were conducted to assess the relevance. Further methodological details are available in Supplementary Methods.
Machine learning (instrumental variable causal forest)	CLATE	Possession of Nintendo Switch/PS5.	Winning game console lottery.	This method predicts the treatment effects for each individual, considering their specific characteristics. The primary aim was to investigate effect modification (or moderation). Additionally, the correlation between CLATEs on well-being and video game duration was analyzed to better understand the underlying mechanism of the impact of video gaming.	The method aligns with key principles of instrumental variable regressions. The credibility of the estimation is assessed by comparing estimates to those derived from traditional instrumental variable regression analyses (Figure 3 and Supplementary Figure 19). Further methodological details are available in Supplementary Methods.

Methodological particulars of the statistical analyses are displayed. The causal inference and machine learning analysis targeted only those who participated in the video game console lottery (N = 1,773 for Nintendo Switch, 6,419 for PS5). ITT, Intention-to-Treat; LATE, Local average treatment effect; CLATE, Conditional local average treatment effect; PSM, Propensity score matching; PS5, PlayStation5.

Reviewer 4, Point 3:

- The authors discuss potential lack of generalizability/ external validity, but do not discuss the poor survey response rate and how this contributes. Also, do the authors have information on the nonrespondents that they can use to reweight their sample?

Thank you for highlighting the importance of considering survey response rates. We acknowledge the challenges posed by nonresponse in our study, which achieved a response rate of 59%. While this rate is not high, it aligns with those found in other lottery-based natural experiment studies, such as 46% in Imbens et al. (2001),⁸ 55% in Doherty et al. (2006),⁷ and 32% in Kuhn et al. (2011).⁴¹ Additionally, online surveys generally experience low response rates, with an average of 44% reported in a meta-analysis.⁴²

We recognize the significance of nonresponse as a factor that primarily leads to two issues: (i) biased estimates and (ii) reduced generalizability of the findings. We discuss those concerns as follows.

(i) Survey nonresponses may cause biased estimates.

Firstly, nonresponses may distort the randomness, potentially leading to biased estimates. For example, in a randomized experiment, systematic attrition post-randomization can skew results, despite the initial random assignment of treatments. This issue is similarly pertinent in a lottery-based natural experiment, where nonresponses can interfere with the quasi-randomness inherent in the study design. This is one of the reasons that using PSM is recommended by Imbens (2015).⁹ As detailed in Point 1, we provided an in-depth explanation of how PSM aids in mitigating potential biases. For example, PSM can enhance covariate balance and ensure common support, as illustrated in Supplementary Figures 2-4. It is worth highlighting that natural experiments are generally less susceptible to nonresponse bias, as discussed in previous studies.⁴³ This advantage was noticeable even before applying PSM, as we observed only modest differences in the background characteristics between lottery winners and non-winners (Supplementary Tables 3-4). In contrast, observational studies that do not leverage a natural experimental setup are at a greater risk of encountering nonresponse bias, where nonresponses are more likely to have a systematic relationship with both the outcome and exposure variables, increasing the potential for biased estimates. Given those, we posit that the potential for nonresponse-induced bias in our study is limited.

(ii) Survey nonresponses may reduce generalizability of the findings.

Secondly, nonresponses may diminish the generalizability of the findings. The technique of reweighting, based on the background characteristics of nonrespondents, could alleviate this issue. We possess some basic information on nonrespondents, including age, gender, and gaming preferences, which could be utilized for reweighting. However, our study faced a specific constraint: the lack of essential information about nonrespondents, notably their lottery participation behavior (necessary information to know whether an individual is included in the quasi-experiment and thereby the analysis sample), making reweighting unfeasible.

We would like to carefully explain the difficulty in applying reweighting to our study. Our analysis sample is those who participated in the Nintendo Switch lottery and the PlayStation5 lottery. For those who did not respond to our survey, we do not know whether they participated in the lotteries. In Letter Table 1, we show the comparison between the entire sample and the analysis sample (the sample included in our quasi-experiment) in terms of their nonresponses.

This problem makes it difficult to implement three common strategies to address non-responses. One is providing attrition rates for the treatment and control groups, an essential practice in field experiments.⁴⁴ The second is comparing the characteristics of respondents and nonrespondents among the analysis sample. The third is reweighting using the pre-treatment characteristics of nonrespondents.

Lack of information on nonrespondents who participated in the lottery.

# of Entire Sample: those survey offers were sent.	
Nonrespondents	Respondents
66,987	97,602
# of Analysis Sample: those who participated in the lottery.	
Nonrespondents	Respondents
?	8,192

To address your concern regarding nonresponses, we assessed potential systematic differences in background characteristics across our entire sample, comparing respondents with nonrespondents, as shown in Supplementary Table 17. Recognizing that direct measures to fully address nonresponses within our analysis sample would be beyond our current capabilities (due to insufficient information), this approach may serve as the next best alternative. The comparison of characteristics showed modest differences; thus, our respondent pool adequately reflects the diversity of the original survey recipients. Therefore, it is plausible to infer that the systematic difference caused by nonresponses in our analysis sample was small, affecting the generalizability of our findings only to a limited extent.

Considering these points, we expanded the Discussion section and carefully discussed the issue of nonresponse, which is also highlighted in the paragraphs elaborating study limitations. The revisions are listed below.

Comparison of baseline characteristics between respondents and nonrespondents.

Variable	Nonrespondents (N=66,987)	Respondents (N=97,602)	Normalized difference
	Mean/SD	Mean/SD	
Age	35.031 [15.489]	38.391 [16.518]	-0.210
Gender (Male)	0.424 [0.494]	0.513 [0.500]	-0.179
Married	0.518 [0.500]	0.572 [0.495]	-0.109
Divorced/separated	0.062 [0.242]	0.064 [0.244]	-0.006
Have child(ren)	0.449 [0.497]	0.502 [0.500]	-0.106
Gaming Preference			
Hardcore gamer	0.215 [0.410]	0.160 [0.367]	0.139
Core gamer	0.234 [0.423]	0.203 [0.402]	0.076
Middle-core gamer	0.214 [0.410]	0.233 [0.423]	-0.045
Casual gamer	0.160 [0.367]	0.176 [0.380]	-0.042
Non-gamer	0.178 [0.382]	0.229 [0.420]	-0.128
Job: industries			
Engineering and construction ¹	0.049 [0.216]	0.054 [0.227]	-0.024
Textile and Cosmetics ²	0.055 [0.229]	0.055 [0.228]	0.001
Manufacturing	0.088 [0.283]	0.104 [0.305]	-0.053
Trading and Mass media ³	0.027 [0.161]	0.031 [0.173]	-0.025
Distributors, Retailers	0.048 [0.214]	0.047 [0.212]	0.004
Carriers ⁴	0.038 [0.191]	0.041 [0.198]	-0.015
Public works	0.041 [0.199]	0.051 [0.220]	-0.045
IT industries ⁵	0.038 [0.192]	0.045 [0.208]	-0.034
Banks and Financial services	0.027 [0.161]	0.028 [0.166]	-0.011
Food and Other Services ⁶	0.112 [0.315]	0.109 [0.312]	0.008
Medical care, Welfare	0.080 [0.271]	0.069 [0.253]	0.042
Education	0.034 [0.182]	0.037 [0.188]	-0.013
Others ⁷	0.068 [0.252]	0.070 [0.254]	-0.005

Notes. Respondents' characteristics are displayed. Caregivers' characteristics are used where appropriate. The characteristics that were available before sending survey offers are shown.

¹Civil engineering, Construction, Real estate, Housing and building services; ²Daily necessities, Textile and apparel, Cosmetics, Food and Beverages; ³Trading companies, Publishing, Printing, Mass media; ⁴Carriers, Warehousing, Logistics; ⁵Software and Information services; ⁶Food services, Hairdressing, Cosmetology, Other Services; ⁷Other industries and types of business.

Discussion

“Our study encountered the prevalent issue of survey nonresponse, achieving a response rate of 59%. This rate aligns with those observed in other lottery-based natural experiment studies. For example, response rates of 46% were reported by Imbens et al. (2001),⁸ 55% by Doherty et al. (2006),⁷ and 32% by Kuhn et al. (2011).⁴¹ Additionally, online surveys like ours generally experience low response rates, with an average of 44% reported.⁴²

This study dealt with the issue of survey nonresponses, primarily leading to biased causal effect estimates and reduced generalizability of the findings. Firstly, employing PSM aids in mitigating potential bias. For example, PSM can enhance covariate balance and ensure common support, as illustrated in Supplementary Figures 2-4 (further rationale of employing PSM available in ‘Methods’ section). It is also important to note that natural experiments are

generally less susceptible to nonresponse bias than non-experimental observational studies, as highlighted in the previous studies.⁴³

Secondly, regarding generalizability, we assessed the potential systematic differences in background characteristics across the entire sample between respondents and nonrespondents (Supplementary Table 17). The comparison showed modest differences; thus, it is plausible to infer that the systematic difference caused by nonresponses in our analysis was small, affecting the generalizability of our findings only to a limited extent. Note that using the entire sample for the comparison is the next best alternative. Ideally, a comparison between respondents and nonrespondents within the analysis sample—lottery participants—would be more direct, but such data were unavailable for nonrespondents in our study.” Lines 343-364

“Another limitation is the generalizability of our findings. Our estimates reflect the video gaming impact for individuals who bought a Switch/PS5 after winning the lottery, accompanied by 0-4 hours longer video game time (Supplementary Figure 14). Consequently, the study implications might not directly apply to dramatically different contexts. For instance, our study cannot provide insights into the influence on a person who rarely played games but suddenly adopted an extremely long gaming practice (elaborated in Supplementary Result 7). Moreover, the applicability of our results to users of other gaming platforms, including different consoles and smartphones, remains uncertain. Additionally, the previously discussed challenges to our findings’ generalizability, stemming from survey nonresponses, necessitate careful interpretation in broader contexts. In particular, the potential underrepresentation of certain demographics or gaming behavior subgroups due to nonresponses might conceal varied effects of video gaming, deviating from the patterns we observed among our respondents.” Lines 379-390

Reviewer 4, Point 4:

- Relatedly, in several places (e.g, the figure captions) the authors interpret their estimates as applying to the “Japanese population”. They are not licensed to make such interpretations given the survey design.

Thank you for your valuable feedback. Upon carefully reviewing our manuscript and the highlighted instances, we recognize that our language implied a broader applicability of our findings than is supported by the study design. We wish to clarify that we did not intend to extend our findings to the entire Japanese population. Our study specifically examined a subset of individuals in Japan during the COVID-19 period, and our findings should be understood within this context. We acknowledge the inherent nature of natural experiment studies as context-specific, often limiting their external validity. We agree that our findings should be interpreted with caution regarding their applicability beyond our study’s context.

In light of these considerations, we have meticulously revised the language throughout our manuscript, including figure captions, to more accurately reflect our study’s specific scope and limitations. The adjustments ensure that our interpretations remain faithful to the evidence and context presented. The relevant revisions are listed below.

Figure captions

Figure 1: Causal impact of winning Nintendo Switch and PS5 lottery on well-being in Japan (N=8,192).

Figure 2: Causal impact of video game engagement on well-being in Japan (N=8,192).

Discussion

“Another limitation is the generalizability of our findings. Our estimates reflect the video gaming impact for individuals who bought a Switch/PS5 after winning the lottery, accompanied by 0-4 hours longer video game time (Supplementary Figure 14). Consequently, the study implications might not directly apply to dramatically different contexts. For instance, our study cannot provide insights into the influence on a person who rarely played games but suddenly adopted an extremely long gaming practice (elaborated in Supplementary Result 7). Moreover, the applicability of our results to users of other gaming platforms, including different consoles and smartphones, remains uncertain. Additionally, the previously discussed challenges to our findings’ generalizability, stemming from survey nonresponses, necessitate careful interpretation in broader contexts. In particular, the potential underrepresentation of certain demographics or gaming behavior subgroups due to nonresponses might conceal varied effects of video gaming, deviating from the patterns we observed among our respondents.” Lines 379-390

Reviewer 4, Point 5:

- I looked in the supplement and cannot find information on the actual estimator used for estimating the LATE.

Thank you for your valuable feedback and for drawing our attention to the lack of information regarding the estimator for estimating the LATE. We recognize the oversight and appreciate the opportunity to clarify it upon review.

In response to your comment, we have revised our manuscript’s Methods section and Table 2 (referenced earlier in this letter) to explicitly state that we estimated the LATE using the Two-Stage Least Squares (2SLS) estimator.⁴⁵ This modification ensures that the methodology used is transparent and straightforward for our readers. We revised our manuscript as follows.

Main text

Methods

“Employing the Two-stage least squares (2SLS) estimation technique, we subsequently estimated the causal effect of owning a Switch/PS5—Local Average Treatment Effects (LATE). We addressed the unmeasured confounding bias by treating the lottery data (dummy variables indicating whether one won a lottery for Switch and PS5, respectively) as excluded instruments. As exposure variables, possession of Switch/PS5, played Switch/PS5 last month, and gameplay time were examined. The exclusion restriction is plausibly satisfied by the lottery-winning instruments for ownership, but reservations exist for gameplay-related variables. Further details are in the Supplementary Method 2.5.” Lines 605-612

Reviewer 4, Point 6:

*- Line 434: There is no such thing as “causal inference methods”. One can be interested in a “causal effect”. One can then enumerate the assumptions under which this causal effect is *identified* from the observed data. If the assumptions are believed to be met, then, one can interpret the estimate of the statistical estimand in a causal way.*

Thank you for highlighting the importance of recognizing a nuanced understanding of the terms “causal inference”, “causal effect,” and the problems with using the term “causal inference methods.” Following your advice, we removed the term “causal inference methods” from our manuscript.

We revised the relevant part in the methods section:

“We used three methods: multivariate regression, propensity score matching approach (PSM), and instrumental variable (IV) method.” Lines 486-487

We revised the abstract:

“Employing approaches designed for causal inference on survey data (n=97,602), we found that game console ownership (Nintendo Switch and PlayStation5), along with increased gameplay, improved mental well-being.” Lines 46-48

We revised the relevant part in our supplementary material:

“When using methods such as multivariate regression or propensity score matching (PSM), it is essential to consider the underlying assumptions for the estimates to be interpreted as causal.” Lines 197-199

These revisions have not only addressed your concerns but also enhanced the overall clarity and rigor of our manuscript. Thank you again for your constructive feedback, which has undoubtedly strengthened our work.

Summary

We thank the editors and reviewers for taking the time to consider our manuscript. We are convinced that the work has improved substantially, and we hope you agree that it will make an important contribution to the literature on this topic.

Yours Sincerely,

Hiroyuki Egami, PhD
Assistant Professor
Research Institute of Economic Science, Nihon University, Tokyo, Japan

References

1. Ryan, R. M., Rigby, C. S. & Przybylski, A. The motivational pull of video games: A self-determination theory approach. *Motiv. Emot.* **30**, 347–363 (2006).
2. Whitaker, J. L. & Bushman, B. J. ‘Remain calm. Be kind.’ Effects of relaxing video games on aggressive and prosocial behavior. *Soc. Psychol. Personal. Sci.* **3**, 88–92 (2012).
3. Primack, B. A. *et al.* Role of video games in improving health-related outcomes: A systematic review. *Am. J. Prev. Med.* **42**, 630–638 (2012).
4. Pallavicini, F., Pepe, A. & Mantovani, F. Commercial off-the-shelf video games for reducing stress and anxiety: Systematic review. *JMIR Ment. Heal.* **8**, 1–19 (2021).
5. Peracchia, S. & Curcio, G. Exposure to video games: Effects on sleep and on post-sleep cognitive abilities. A systematic review of experimental evidences. *Sleep Sci.* **11**, 302–314 (2018).
6. Suchert, V., Hanewinkel, R. & Isensee, B. Sedentary behavior and indicators of mental health in

- school-aged children and adolescents: A systematic review. *Prev. Med. (Baltim)*. **76**, 48–57 (2015).
7. Doherty, D., Gerber, A. S. & Green, D. P. Personal income and attitudes toward redistribution: A study of lottery winners. *Polit. Psychol.* **27**, 441–458 (2006).
 8. Imbens, G. W., Rubin, D. B. & Sacerdote, B. Estimating the Effect of Unearned Income on Labor Earnings, Savings, and Consumption: Evidence from a Survey of Lottery Players. *Am. Econ. Rev.* **91**, (2001).
 9. Imbens, G. W. Matching methods in practice: Three examples. *J. Hum. Resour.* **50**, 373–419 (2015).
 10. Imbens, G. W. & Rubin, D. B. *Causal inference for statistics, social, and biomedical sciences: An introduction. Causal inference for statistics, social, and biomedical sciences: An introduction.* (Cambridge University Press, 2015). doi:10.1017/CBO9781139025751.
 11. Chattopadhyay, A. & Zubizarreta, J. R. Causation , Comparison , and Regression. *Harvard Data Sci. Rev.* **6**, (2024).
 12. Wager, S. & Athey, S. Estimation and Inference of Heterogeneous Treatment Effects using Random Forests. *J. Am. Stat. Assoc.* **113**, 1228–1242 (2018).
 13. Athey, S., Tibshirani, J. & Wager, S. Generalized random forests. *Ann. Stat.* **47**, 1179–1203 (2019).
 14. Athey, S. & Wager, S. Policy Learning With Observational Data. *Econometrica* **89**, 133–161 (2021).
 15. Wang, G., Li, J. & Hopp, W. J. An Instrumental Variable Forest Approach for Detecting Heterogeneous Treatment Effects in Observational Studies. *Manage. Sci.* **68**, 3399–3418 (2022).
 16. Brooks, J. M. *et al.* Assessing the ability of an instrumental variable causal forest algorithm to personalize treatment evidence using observational data: the case of early surgery for shoulder fracture. *BMC Med. Res. Methodol.* **22**, 1–16 (2022).
 17. Ling, Y., Upadhyaya, P., Chen, L., Jiang, X. & Kim, Y. Emulate randomized clinical trials using heterogeneous treatment effect estimation for personalized treatments: Methodology review and benchmark. *J. Biomed. Inform.* **137**, 104256 (2023).
 18. Salditt, M., Eckes, T. & Nestler, S. A Tutorial Introduction to Heterogeneous Treatment Effect Estimation with Meta-learners. *Adm. Policy Ment. Heal. Ment. Heal. Serv. Res.* (2023) doi:10.1007/s10488-023-01303-9.
 19. Chernozhukov, V., Hansen, C., Kallus, N., Spindler, M. & Syrgkanis, V. *Applied Causal Inference Powered by ML and AI.* (2024).
 20. Athey, S. & Imbens, G. W. Machine Learning Methods That Economists Should Know about. *Annu. Rev. Econom.* **11**, 685–725 (2019).
 21. Gong, X., Hu, M., Basu, M. & Zhao, L. Heterogeneous treatment effect analysis based on machine-learning methodology. *CPT Pharmacometrics Syst. Pharmacol.* **10**, 1433–1443 (2021).
 22. Zhang, Y., Li, H. & Ren, G. Estimating heterogeneous treatment effects in road safety analysis using generalized random forests. *Accid. Anal. Prev.* **165**, 106507 (2022).
 23. Miller, S. Causal forest estimation of heterogeneous and time-varying environmental policy

- effects. *J. Environ. Econ. Manage.* **103**, 102337 (2020).
24. Langenberger, B. *et al.* Exploring treatment effect heterogeneity of a PROMs alert intervention in knee and hip arthroplasty patients: A causal forest application. *Comput. Biol. Med.* **163**, (2023).
 25. Li, Z. fei, Zhou, Q., Chen, M. & Liu, Q. The impact of COVID-19 on industry-related characteristics and risk contagion. *Financ. Res. Lett.* **39**, 101931 (2021).
 26. Scarpa, J. *et al.* Assessment of Risk of Harm Associated with Intensive Blood Pressure Management among Patients with Hypertension Who Smoke: A Secondary Analysis of the Systolic Blood Pressure Intervention Trial. *JAMA Netw. Open* **2**, 1–11 (2019).
 27. Davis, J. M. V. & Heller, S. B. Rethinking the benefits of youth employment programs: The heterogeneous effects of summer jobs. *Rev. Econ. Stat.* **102**, 664–677 (2020).
 28. Hoffman, I. & Mast, E. Heterogeneity in the effect of federal spending on local crime: Evidence from causal forests. *Reg. Sci. Urban Econ.* **78**, 103463 (2019).
 29. Athey, S. & Wager, S. Estimating Treatment Effects with Causal Forests: An Application. *Obs. Stud.* **5**, 37–51 (2019).
 30. Athey, S., Simon, L. K., Skans, O. N., Vikstrom, J. & Yakymovych, Y. *The Heterogeneous Earnings Impact of Job Loss Across Workers, Establishments, and Markets*. <http://arxiv.org/abs/2307.06684> (2024).
 31. Shiba, K. *et al.* Uncovering heterogeneous associations of disaster-related traumatic experiences with subsequent mental health problems: A machine learning approach. *Psychiatry Clin. Neurosci.* **76**, 97–105 (2022).
 32. Iyengar, R., Park, Y.-H. H. & Yu, Q. The Impact of Subscription Programs on Customer Purchases. *J. Mark. Res.* **59**, 1101–1119 (2022).
 33. Jawadekar, N. *et al.* Practical Guide to Honest Causal Forests for Identifying Heterogeneous Treatment Effects. *Am. J. Epidemiol.* **192**, 1155–1165 (2023).
 34. Tibshirani, J. *et al.* R Package ‘grf’. at <https://grf-labs.github.io/grf/> (2024).
 35. Athey, S. & Imbens, G. Recursive partitioning for heterogeneous causal effects. *Proceedings of the National Academy of Sciences of the United States of America* vol. 113 7353–7360 at <https://doi.org/10.1073/pnas.1510489113> (2016).
 36. Bertrand, M., Crepon, B., Marguerie, A. & Premand, P. *Contemporaneous and Post-Program Impacts of a Public Works Program: Evidence from Côte d’Ivoire Marianne*. *World Bank Policy Research Working Papers* <http://hdl.handle.net/10986/28460> (2017).
 37. Baum, A. *et al.* Targeting weight loss interventions to reduce cardiovascular complications of type 2 diabetes: a machine learning-based post-hoc analysis of heterogeneous treatment effects in the Look AHEAD trial. *lancet. Diabetes Endocrinol.* **5**, 808–815 (2017).
 38. Angrist, J. D. & Evans, W. N. Children and Their Parents’ Labor Supply: Evidence from Exogenous Variation in Family Size. *Am. Econ. Rev.* **88**, 450–477 (1998).
 39. Floyd, S. B., Thigpen, C., Kissenberth, M. & Brooks, J. M. Association of Surgical Treatment with Adverse Events and Mortality among Medicare Beneficiaries with Proximal Humerus Fracture. *JAMA Netw. Open* **3**, 1–14 (2020).
 40. Robins, J. M., Rotnitzky, A. & Zhao, L. P. Estimation of Regression Coefficients When Some

- Regressors Are Not Always Observed. *J. Am. Stat. Assoc.* **89**, 846–866 (1994).
41. Kuhn, P., Kooreman, P., Soetevent, A. & Kapteyn, A. The effects of lottery prizes on winners and their neighbors: Evidence from the Dutch postcode lottery. *Am. Econ. Rev.* **101**, 2226–2247 (2011).
 42. Wu, M. J., Zhao, K. & Fils-Aime, F. Response rates of online surveys in published research: A meta-analysis. *Comput. Hum. Behav. Reports* **7**, 100206 (2022).
 43. Dunning, T. Improving Causal Inference: Strengths and Limitations of Natural Experiments. *Polit. Res. Q.* **61**, 282–293 (2008).
 44. Duflo, E., Glennerster, R. & Kremer, M. *Using Randomization in Development Economics Research: A Toolkit. NBER working paper series* <https://www.nber.org/papers/t0333> (2006) doi:10.2139/ssrn.951841.
 45. Angrist, J. D. & Pischke, J.-S. *Mostly Harmless Econometrics*. (Princeton University Press, 2009). doi:10.2307/j.ctvc4j72.

Decision Letter, second revision:

20th May 2024

Dear Dr. Egami,

Thank you for your patience as we've prepared the guidelines for final submission of your Nature Human Behaviour manuscript, "Causal effect of video game play on mental well-being in Japan 2020-2022" (NATHUMBEHAV-23061961B). Please carefully follow the step-by-step instructions provided in the attached file, and add a response in each row of the table to indicate the changes that you have made. Please also address the additional marked-up edits we have proposed within the reporting summary. Ensuring that each point is addressed will help to ensure that your revised manuscript can be swiftly handed over to our production team.

We would hope to receive your revised paper, with all of the requested files and forms within two-three weeks. Please get in contact with us if you anticipate delays.

Nature Human Behaviour offers a Transparent Peer Review option for new original research manuscripts submitted after December 1st, 2019. As part of this initiative, we encourage our authors to support increased transparency into the peer review process by agreeing to have the reviewer comments, author rebuttal letters, and editorial decision letters published as a Supplementary item. When you submit your final files please clearly state in your cover letter whether or not you would like to participate in this initiative. Please note that failure to state your preference will result in delays in accepting your manuscript for publication.

In recognition of the time and expertise our reviewers provide to Nature Human Behaviour's editorial process, we would like to formally acknowledge their contribution to the external peer review of your manuscript entitled "Causal effect of video game play on mental well-being in Japan 2020-2022". For those reviewers who give their assent, we will be publishing their names alongside the published article.

Cover suggestions

We welcome submissions of artwork for consideration for our cover. For more information, please see our guide for cover artwork.

ORCID

Non-corresponding authors do not have to link their ORCIDs but are encouraged to do so. Please note that it will not be possible to add/modify ORCIDs at proof. Thus, please let your co-authors know that if they wish to have their ORCID added to the paper they must follow the procedure described in the following link prior to acceptance: <https://www.springernature.com/gp/researchers/orcid/orcid-for-nature-research>

Nature Human Behaviour has now transitioned to a unified Rights Collection system which will allow our Author Services team to quickly and easily collect the rights and permissions required to publish your work. Approximately 10 days after your paper is formally accepted, you will receive an email in providing you with a link to complete the grant of rights. If your paper is eligible for Open Access, our Author Services team will also be in touch regarding any additional information that may be required to arrange payment for your article.

Please note that *Nature Human Behaviour* is a Transformative Journal (TJ). Authors may publish their research with us through the traditional subscription access route or make their paper immediately open access through payment of an article-processing charge (APC). Authors will not be required to make a final decision about access to their article until it has been accepted. Find out more about Transformative Journals

[REDACTED]

Best regards,
[REDACTED]

On behalf of
[REDACTED]

Reviewer #6:
Remarks to the Author:
The authors have addressed my comments.

Final Decision Letter:

Dear Dr Egami,

We are pleased to inform you that your Article "Causal effect of video gaming on mental well-being in Japan 2020-2022", has now been accepted for publication in *Nature Human Behaviour*.

Please note that *Nature Human Behaviour* is a Transformative Journal (TJ). Authors may publish their research with us through the traditional subscription access route or make their paper immediately open access through payment of an article-processing charge (APC). Authors will not be required to make a final decision about access to their article until it has been accepted. Find out more about Transformative Journals

With best regards,
[REDACTED]